# MCM double hexamer loading visualized with human proteins

Florian Weissmann[1,4], Julia F. Greiwe[2,4], Thomas Pühringer[2], Evelyn L. Eastwood[1], Emma C. Couves[2], Thomas C. R. Miller[2,3], John F. X. Diffley[1,5 ✉] & Alessandro Costa[2,5 ✉]

Eukaryotic DNA replication begins with the loading of the MCM replicative DNA helicase as a head-to-head double hexamer at origins of DNA replication[1–3]. Our current understanding of how the double hexamer is assembled by the origin recognition complex (ORC), CDC6 and CDT1 comes mostly from budding yeast. Here we characterize human double hexamer (hDH) loading using biochemical reconstitution and cryo-electron microscopy with purified proteins. We show that the human double hexamer engages DNA differently from the yeast double hexamer (yDH), and generates approximately five base pairs of underwound DNA at the interface between hexamers, as seen in hDH isolated from cells[4]. We identify several differences from the yeast double hexamer in the order of factor recruitment and dependencies during hDH assembly. Unlike in yeast[5–8], the ORC6 subunit of the ORC is not essential for initial MCM recruitment or hDH loading, but contributes to an alternative hDH assembly pathway that requires an intrinsically disordered region in ORC1, which may work through a MCM–ORC intermediate. Our work presents a detailed view of how double hexamers are assembled in an organism that uses sequence-independent replication origins, provides further evidence for diversity in eukaryotic double hexamer assembly mechanisms[9], and represents a first step towards reconstitution of DNA replication initiation with purified human proteins.

We expressed the human ORC, CDC6, CDT1 and MCM2–7 (hereafter MCM) using the biGBac baculovirus expression system[10] (Fig. 1a). Consistent with previous work[11,12], we found that ORC6 did not co-purify with ORC1–5 when co-expressed, so we produced ORC1–5 and ORC6 separately. Similarly, unlike yeast, CDT1 did not co-purify with MCM, so CDT1 and MCM were expressed and purified separately. Negative-stain electron microscopy performed in solution for a human MCM loading reaction using reaction conditions similar to established yeast reactions (100 mM potassium glutamate) on a short synthetic yeast origin flanked by nucleosomes yielded 2D averages of a particle similar to the yeast double hexamer (Extended Data Fig. 1a), suggesting that hDH was being assembled. We aimed to develop a facile biochemical assay to follow double hexamer assembly. Although assays based on bead-bound DNA have been useful for budding yeast proteins[7,8], N-terminal intrinsically disordered regions (IDRs) of the *Drosophila melanogaster* orthologues of ORC1, CDC6 and CDT1 drive liquid–liquid phase separation[13,14] and the human proteins have similar IDRs, which generate high background on beads. We therefore developed a nuclease protection assay (Fig. 1b) to examine DNA-bound protein complexes. In this assay, complexes are assembled on soluble plasmid DNA and DNA is digested extensively with benzonase; all that remains after digestion is the DNA that is protected by proteins, which is resolved on polyacrylamide gels. With the yeast ORC, Cdc6 and MCM–Cdt1 in the presence of ATP, reaction conditions

that are known to generate double hexamer, a band approximately 55 base pairs (bp) in size appeared with time, beginning at 2–4 min (Fig. 1c, top). This is slightly shorter than the length of DNA contained within the central channel of the yeast double hexamer[15]. We initially expressed and purified the human ORC, CDC6 and CDT1 lacking the N-terminal IDRs of ORC1, CDC6 and CDT1 (ΔN; Fig. 1a) to avoid complications from liquid–liquid phase separation. Using these truncated proteins in the footprinting assay, we observed a band of slightly larger size (around 75 bp) that appeared with similar kinetics (Fig. 1c, bottom). Upon longer benzonase digestion, this 75-bp product was converted to a product very similar to the size of the yeast footprint (around 55 bp) (Extended Data Fig. 1b) and very similar in size to the length of DNA contained within double hexamer central channel isolated from human cells[4] (54–56 bp). The 75-bp intermediate suggests additional contacts with DNA at the C-terminal exit from the MCM central channel. This assay therefore detects the hDH in a very similar manner to detection of the yDH.

DNA footprinting (Fig. 1d) and negative-stain electron microscopy (Supplementary Fig. 1) showed that the appearance of the hDH under these conditions required ORC1–5, CDC6, MCM and CDT1, but did not require ORC6; in fact, the presence of ORC6 consistently reduced the amount of hDH (Supplementary Fig. 2a). Comparison of protein concentrations used for electron microscopy experiments and footprinting

[1]Chromosome Replication Laboratory, The Francis Crick Institute, London, UK. [2]Macromolecular Machines Laboratory, The Francis Crick Institute, London, UK. [3]Present address: DNRF Center for Chromosome Stability, Department of Cellular and Molecular Medicine, University of Copenhagen, Copenhagen, Denmark. [4]These authors contributed equally: Florian Weissmann, Julia F. Greiwe. [5]These authors jointly supervised this work: John F. X. Diffley, Alessandro Costa. ✉e-mail: john.diffley@crick.ac.uk; alessandro.costa@crick.ac.uk

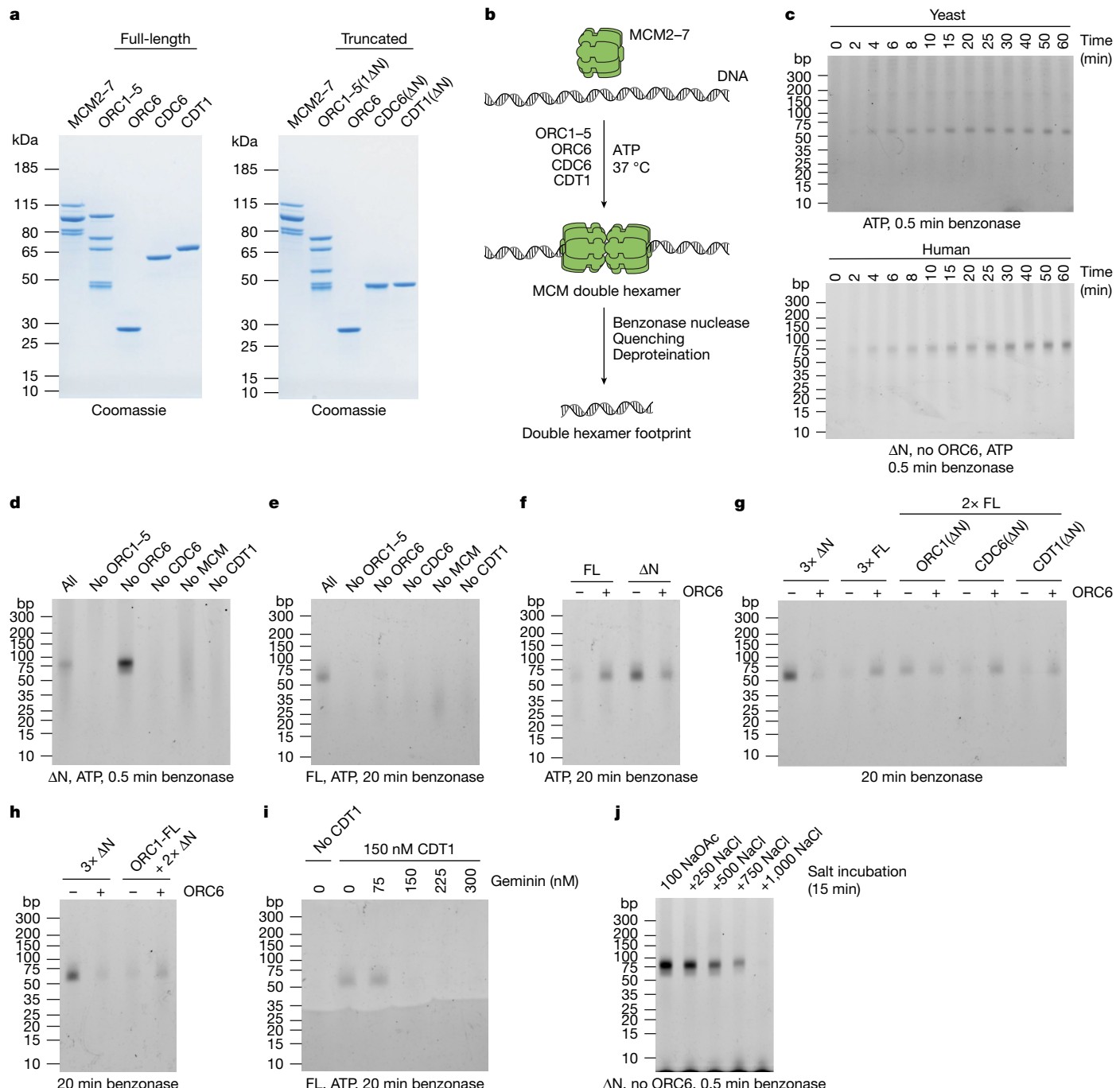

**Fig. 1 | Reconstitution of hDH loading. a**, Purified human MCM loading proteins analysed by SDS–PAGE and Coomassie staining. Full-length proteins (left) and truncated proteins (right): ORC1–5 complex containing ORC1(ΔN) (ORC1–5(1ΔN)), CDC6(ΔN) and CDT1(ΔN). **b**, Outline of the nuclease footprinting assay. MCM loading reactions are treated with benzonase followed by quenching with EDTA, SDS and proteinase K. Then DNA is purified by phenol–chloroform–isoamyl alcohol extraction and ethanol precipitation, resolved on a TBE polyacrylamide gel, and stained with SYBR Gold. **c**, Timecourse of yeast (top) and human (bottom) MCM loading reactions. ΔN indicates that the assay was carried out with truncated proteins. **d**, Nuclease footprinting assay with truncated human proteins. Proteins were omitted as indicated. **e**, As **d**, with full-length (FL) proteins. **f**, Side-by-side comparison of the ORC6 dependency

with full-length and truncated proteins. **g**, As **f**, testing conditions in which either ORC1, CDC6 or CDT1 was truncated and other proteins remained full-length. **h**, As **f**, with full-length ORC1 and truncated CDC6 and CDT1. **i**, Effect of geminin on full-length MCM loading reactions. CDT1 and geminin were pre-mixed before reactions were started. **j**, Salt stability of the double hexamer. MCM was loaded for 30 min and then incubated in buffers containing the indicated concentrations of sodium chloride (mM) for 15 min, followed by dilution to lower salt and benzonase treatment. The band at the bottom of the gel appears because of incomplete digestion owing to the higher salt concentration used. Gel source data for all figures can be found in Supplementary Fig. 4.

experiments showed the same inhibitory effect of ORC6 (Extended Data Fig. 1c). With the full-length proteins, we saw a band of the same size as the hDH band that also required ORC1–5, CDC6 and CDT1; in this case, however, only a faint band was seen when ORC6 was omitted

(Fig. 1e). Figure 1f shows MCM loading by the truncated and full-length proteins side-by-side, emphasizing that ORC6 stimulates hDH assembly with full-length proteins, but inhibits assembly with the truncated proteins. Consistently, titration of ORC6 from sub- to super-stoichiometric

amounts showed stimulation of hDH assembly with the full-length proteins and inhibition with the truncated proteins (Extended Data Fig. 1d). From these results, we conclude that there is a pathway for MCM loading that does not require ORC6 and that this pathway is more efficient when truncated proteins are used. Stimulation of hDH assembly by ORC6 with the full-length proteins suggests there is an alternative ORC6-dependent MCM loading pathway that requires one or more of the IDRs. To determine which IDR is responsible for this effect of ORC6, we tested each truncation individually. Figure 1g shows that ORC6 stimulated loading in the presence of either CDC6(ΔN) or CDT1(ΔN); however, with ORC1(ΔN) (or especially when all three N-terminal IDRs are truncated), ORC6 no longer stimulated loading, but instead inhibited it. The inclusion of full-length ORC1 with CDC6(ΔN) and CDT1(ΔN) conferred ORC6 dependence (Fig. 1h), indicating that the ORC1 IDR is necessary and sufficient for the ORC6-dependent pathway. In this context, ORC6 may stimulate loading by counteracting an inhibitory effect of the ORC1 IDR.

To characterize MCM loading further, we next tested the effect of geminin, which inhibits hDH assembly in vivo by inhibiting CDT1 (refs. 16–18). MCM loading in vitro by both full-length and truncated proteins was inhibited by geminin, and inhibition occurred when the amount of geminin was equal to or greater than the amount of CDT1 (Fig. 1i). To examine salt stability of hDH, reactions were challenged with different salt concentrations before benzonase treatment. In these reactions, hDH was stable in the presence of up to 500 mM NaCl, but unstable at higher NaCl concentrations. Notably, hDH was stable in 500 mM NaCl (Fig. 1j) for up to 60 min (Extended Data Fig. 1e); hDH could also be detected with short periods of exposure to 750 mM NaCl, but diminished in a time-dependent manner (Extended Data Fig. 1f) indicating that hDH is less salt-stable than yDH, which is stable in up to 2 M NaCl[19]. hDH footprints generated with full-length or truncated proteins in the presence or absence of ORC6 showed very similar salt stabilities in 500 mM NaCl (Extended Data Fig. 1g). The human ORC is not a sequence-specific DNA-binding protein[20–22], and consistently, MCM loading in vitro was similarly efficient with DNA containing or lacking human replication origin sequences (Extended Data Fig. 1h,i).

## The human MCM double hexamer

To characterize hDH and to understand its interactions with DNA, we used cryo-electron microscopy (cryo-EM) to image the human MCM loading reaction that we previously visualized by negative-stain electron microscopy. We solved a 3.1 Å resolution cryo-EM structure of the hDH (Extended Data Fig. 2). The two MCM hexamers interact via two N-terminal tiers forming a tilted homodimerization interface, similar to the yDH structure[23] (Fig. 2a). The catalytic site at the interface between MCM6 and MCM2 is ATP-bound, the MCM4–7 site is empty and all other sites are ADP-bound (Fig. 2b). Our structure of the reconstituted hDH matches that of hDH isolated from cells, although four out of the six catalytic sites were bound with ATP or a mixture of ATP and ADP in the structure of hDH purified from cells[4]. This discrepancy might reflect differences in sample preparation. Whereas in our reconstitution experiment cryo-EM grids were flash-frozen within 45 min of establishing the reaction, hDHs from cells were subjected to purification steps in ATP-containing buffer, which might have favoured nucleotide exchange. Differences in nucleotide occupancy between the two structures, however, exhibit no effect on the protein–DNA contacts. Collectively, our analysis of hDH agrees with observations in yeast that double hexamer loading requires ATP hydrolysis by MCM[15,24,25].

Forty-five base pairs of DNA could be clearly resolved inside the two MCM rings, although the length of the central channel is compatible with protection of 75 bp of DNA, as shown by DNA footprinting. The pre-sensor 1 (PS1) pore loops, which emanate from the ATPase

modules, are arranged in a staircase configuration that follows the DNA double helix. PS1 pore loops of MCM7, MCM4 and MCM6 engage in minor groove contacts with residues that will form the translocation strand upon replisome activation (Fig. 2c) and the h2i pore loops more loosely follow a staircase pattern and engage both strands (Supplementary Fig. 3).

The zinc fingers of the N-terminal B domain form a 13 Å passage that is too narrow to accommodate B-form DNA where the two MCM rings dimerize. For the two DNA strands to traverse this tight pore, 5 bp of the double helix becomes underwound and one base pair is broken (Fig. 2a). The resulting orphan bases are each stabilized symmetrically by R195 and L209 in MCM5. The neighbouring base is also stabilized by R195 through a cation–π interaction (Fig. 2d). The double hexamer structure and the degree of DNA untwisting inside the central channel had the same appearance irrespective of whether ORC6 was included or omitted in the loading reaction (Extended Data Fig. 3a–c). The close match between the structure of hDH from cells[4] and our hDH assembled in vitro indicates that hDH loading alone is sufficient to untwist DNA and break base pairing, with no additional activation steps required.

To assess the role of R195 and L209 of MCM5 in hDH assembly and nucleation of DNA melting, we generated MCM hexamers containing single and double MCM5 mutants. The single mutants and the double mutant loaded MCM almost as well as the wild-type proteins (Fig. 2e and Extended Data Fig. 3d). The double hexamer assembled with the R195A/L209G double mutant (AG mutant) is structurally very similar to the wild-type hDH (Extended Data Fig. 3e). Notably, DNA engagement by the MCM ATPase domain is identical to the wild type and the DNA between hexamers is distorted and untwisted to the same degree, although no breakage of DNA base pairing could be resolved (Fig. 2f). Thus, R195 and L209 are not required for DNA untwisting, but, rather, they stabilize the orphan DNA bases formed upon untwisting.

We observed that the DNA protected after 0.5 min benzonase digestion was slightly less intense and favoured the 55-bp product over the 75-bp intermediate with the AG mutant hDH compared with the wild-type complex despite the structural similarity of wild-type and mutant hDH and DNA within the central channels of hDH (Fig. 2g). We considered that the smaller size might indicate that the mutant hDH is more mobile on DNA than the wild type: movement of hDH during benzonase digestion might expose DNA that was within the central channel at the start of the digestion, resulting in a shorter product. Consistent with this, the footprint generated by hDH was stable with increasing time of digestion up to 20 min, whereas the footprint of the AG mutant hDH disappeared with time and was absent by 20 min. This is not because the AG hDH is less stably bound to DNA than wild-type hDH, because the footprint is still detected when benzoase is added for 0.5 min following 20 min preincubation. Therefore, AG mutant MCM5 can efficiently assemble a hDH containing underwound DNA, but this assembled hDH is more mobile on the DNA than the wild-type hDH.

## Human OCCM-related complexes

To begin to examine the mechanism of hDH assembly, we used the poorly hydrolysable ATPγS analogue of ATP. In budding yeast, ATPγS supports the recruitment of the ORC–CDC6–CDT1–MCM (OCCM) complex[6,26,27], an early intermediate in double hexamer assembly[28]. In the benzonase footprinting assay, yeast OCCM (yOCCM) generates a protection of around 50 bp, consistent with the length of DNA within the ORC–CDC6 and MCM central channels[29] (Extended Data Fig. 4a); a footprint of a similar size was seen with human proteins in ATPγS (Fig. 3a), with an additional, fainter band of approximately 120 bp. Formation of these products with truncated human proteins required ORC1–5, CDT1 and MCM, but did not require ORC6; surprisingly, in the

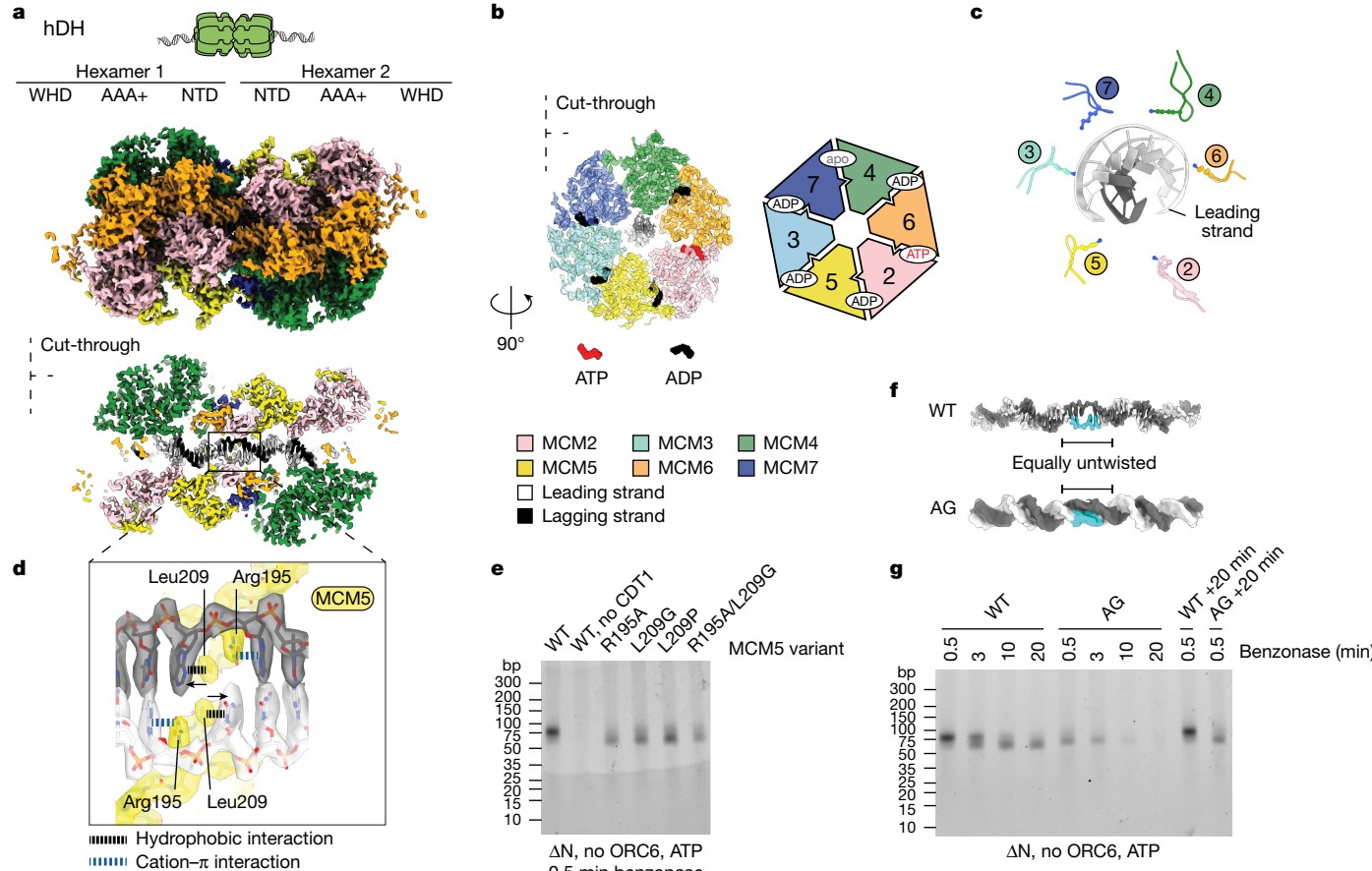

**Fig. 2 | hDH untwists and melts DNA. a**, Surface rendering and cut-through view of the hDH loaded onto duplex DNA. The double helix is untwisted and one base pair is broken at the dimerization interface. This distortion enables DNA to traverse a narrow passage created by two MCM5 zinc-finger domains. NTD, N-terminal domain; WHD, WH domain. **b**, View of C-terminal ATPases of one MCM ring. Nucleotides are shown in the ATPase sites between MCM subunits. **c**, PS1 pore loops form a staircase arrangement that follows the leading strand template. **d**, Zoomed-in view of the broken base pair stabilized by R195 and L209 of MCM5. **e**, Nuclease footprinting assay with wild-type (WT) and indicated mutants of MCM5. **f**, Comparison between duplex DNA in the wild-type MCM5 and MCM5(R195A/L209G) mutant (AG) double hexamer reveals a similar degree of DNA untwisting. **g**, Wild-type and AG mutant MCM were loaded on DNA and subjected to benzonase treatment lasting between 0.5 and 20 min. Controls were incubated for an additional 20 min followed by 0.5 min benzonase treatment.

absence of CDC6, both bands remained visible, although at markedly reduced intensities.

To characterize the structure that is stabilized by ATPγS in reconstituted human MCM recruitment onto nucleosome-capped ARS1 DNA, we analysed the reaction by cryo-EM. We solved a 3.8 Å resolution structure of a complex in which ORC1–5, CDC6, MCM and CDT1 could be recognized (hOCCM; Fig. 3b and Extended Data Fig. 4b–f and Extended Data Fig. 5). As in yOCCM, the ATPase domain of MCM is oriented towards the C-terminal face of a hexameric ORC1–5–CDC6 ring. The C-terminal winged-helix (WH) domains of MCM3, MCM6 and MCM7 contact the ORC5–ORC4, ORC1–CDC6 and CDC6–ORC2 interfaces, respectively. Notably, the WH domain of MCM4 that engages the ORC in yOCCM[29] is invisible in the human structure (Extended Data Fig. 1j–l). Conversely, similar to yOCCM, the human ORC–CDC6 and MCM rings are aligned so that duplex DNA can traverse the entire length of the complex. Although the cryo-EM density is of sufficient quality to only build 39 bp, we note that OCCM central channel is 185 Å long, which is enough to protect the approximately 50 bp measured with benzonase footprinting. Inside the channel, ORC2, ORC4 and CDC6 contact both DNA strands, and MCM4, MCM6 and MCM2 track along the leading strand via PS1 pore loops arranged in a staircase configuration (Supplementary Fig. 3).

ORC6 is absent in the hOCCM structure, which is not surprising given that omitting ORC6 has no effect on OCCM formation in ATPγS according to DNA footprinting (Fig. 3a), and 2D averages obtained in these conditions appear indistinguishable from reactions in which ORC6 is included (Supplementary Fig. 1b,d). Human CDT1 is shorter than the yeast counterpart, as it lacks the N-terminal dioxygenase domain that forms a handle-like feature in the yeast complex[29] (Fig. 3c and Extended Data Fig. 1j). Two tandem WH domains of CDT1 interdigitate between the MCM6–MCM2 and MCM4–MCM6 α-helical bundle (A) domains. The two WH domains of CDT1 are bridged by an α-helical element that packs against the WH domain of MCM6, connecting the N- and C-terminal tiers of the MCM ring along its outer surface (Fig. 3b). Together, CDT1 incorporation into in the hOCCM complex matches observations made for the yOCCM[29]. This is unexpected, given that previous co-expression and reconstitution attempts did not detect any interaction between CDT1 and MCM, which instead form a stable intermediate in the yeast MCM loading reaction. As for the yeast complex[29], the MCM3 and MCM5 subunits are poorly resolved in the hOCCM structure and in the N-terminal side of MCM in particular (Fig. 3b). The MCM2–MCM5 ATPase gate is cracked, yet topologically closed, with a recognizable α-helix from MCM5 packed against the MCM2 ATPase domain, keeping the subunit interface sealed (Fig. 3d).

We solved the structure of a subcomplex of OCCM containing the ORC, CDT1 and MCM but lacking CDC6 to 4.1 Å resolution (hOC₁M; Fig. 3e). Compared with the full hOCCM complex, the ORC and MCM

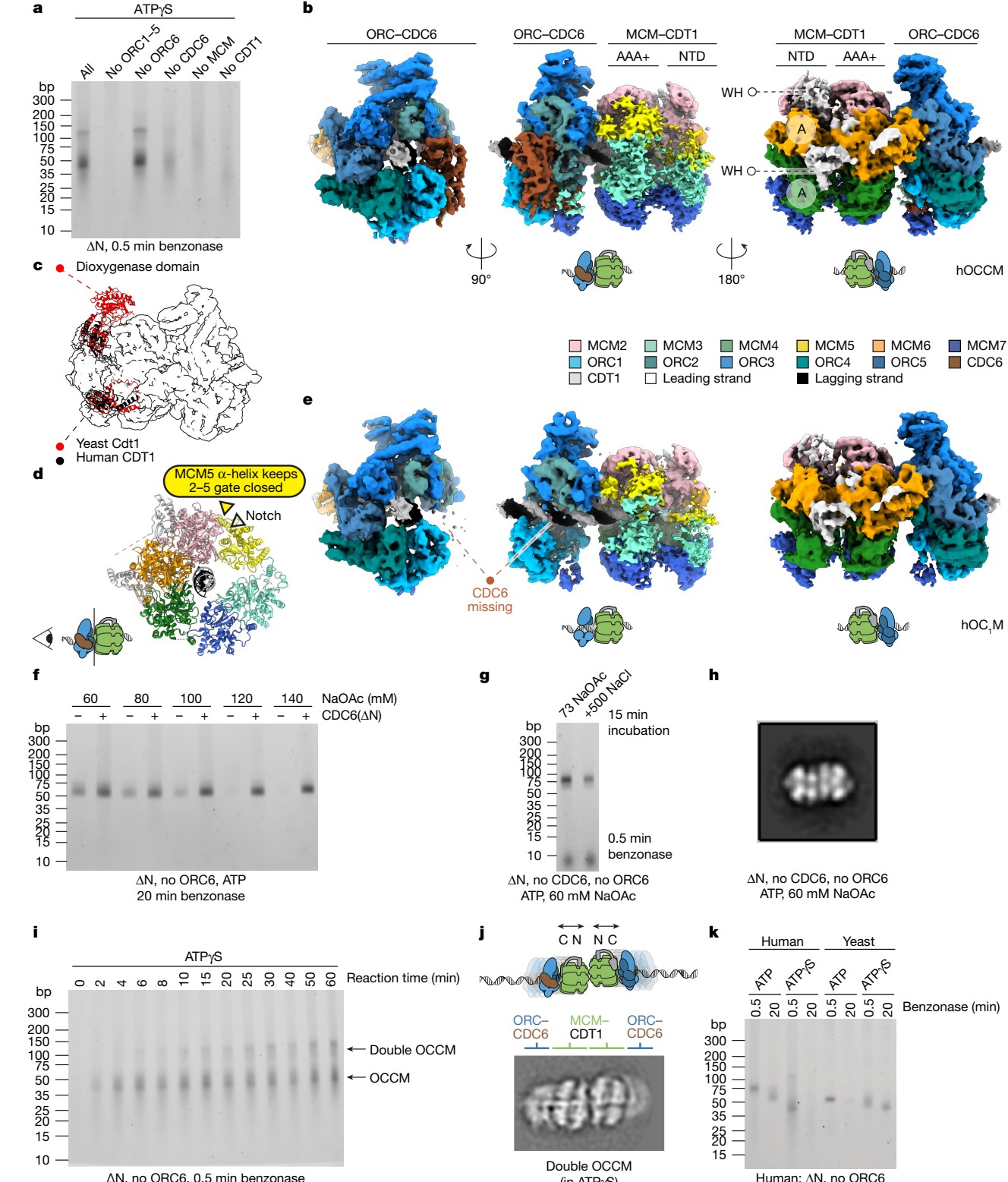

**Fig. 3 | hOCCM complexes. a**, Nuclease footprinting assay performed with truncated proteins and ATPγS. Proteins were omitted as indicated. **b**, Three views of the hOCCM structure. The WH domains of CDT1 (white) interdigitate between the N-terminal A domain of MCM6 and MCM4. **c**, Yeast Cdt1 (red) and human CDT1 (black) establish the same interactions with the MCM complex in OCCM. **d**, The MCM2–MCM5 gate is notched but topologically closed by an MCM5 α-helix packing against the MCM2 ATPase. **e**, Three views of the hOC₁M structure. **f**, Footprinting assay of MCM loading reactions (truncated proteins, no ORC6, ATP) in the presence or absence of CDC6(ΔN) using assay buffers containing the indicated sodium acetate (NaOAc) concentrations. **g**, Salt

stability of double hexamer generated in the absence of CDC6 and ORC6 (truncated proteins, ATP). MCM was loaded at 60 mM sodium acetate for 30 min, incubated in buffers containing the indicated salt concentrations for 15 min, then diluted to low salt and treated with benzonase. **h**, Negative-stain 2D average of hDH loaded in the absence of CDC6 and ORC6 (truncated proteins, 60 mM NaOAc, ATP). **i**, Footprinting assay with a reaction timecourse in ATPγS (truncated proteins, no ORC6). **j**, Cryo-EM 2D average of a double OCCM. **k**, Nuclease footprinting assay with human and yeast MCM loading reactions using either ATP or ATPγS. The experiment was performed at 30 °C.

are bridged by WH domains of MCM6 and MCM7 domains but not that of MCM3, which becomes too flexible to be seen in the absence of CDC6 (Extended Data Fig. 1k,l). The ORC2 WH domain also becomes invisible in the absence of CDC6, exposing a segment of B-form DNA clamped by ORC1–5 within the core of the protein complex (Fig. 3e). This hOC$_1$M complex could either be a product of OCCM disassembly or form independently of CDC6 recruitment. Despite the weaker signal, the OCCM-like benzonase footprint observed when CDC6 is omitted support the latter scenario, as do the OCCM-like 2D class averages obtained by negative-stain electron microscopy (Fig. 3a and Supplementary Fig. 1f). This led us to re-evaluate the requirements for CDC6 in MCM loading. Although CDC6 is required for efficient double hexamer loading in standard conditions (Supplementary Fig. 1), we found that at lower salt concentrations, a hDH footprint is observed in the absence of CDC6 when ATP is used (Fig. 3f). This footprint was still present after incubation with 500 mM NaCl (Fig. 3g) and negative-stain electron microscopy confirmed the formation of hDH (Fig. 3h and Supplementary Fig. 2b). Therefore, CDC6 is not essential for hDH assembly under all conditions.

The OCCM-related structures described so far explain the 50-bp but not the 120-bp benzonase footprint band. We noted that the 50-bp band in OCCM appeared within 2–4 min; however, the larger 120-bp band accumulated much more slowly and did not reach a maximum until 30–60 min of incubation (Fig. 3i). Geminin, when added after 9 min, led to the disruption of the 50-bp band but not the 120-bp band (Extended Data Fig. 1m). The larger footprint was roughly double the size of the OCCM, and we considered that it might be two adjoined OCCM complexes. A more loosely interacting MCM dimerization interface compared with the double hexamer could account for the slightly extended footprint of a double OCCM compared with two adjoined single OCCMs. Consistent with this, head-to-head double-OCCM class averages were seen in ATPγS reactions after 45 min (Fig. 3j). In these double-OCCM 2D averages, MCM features are well-resolved, indicating a uniform subunit register. No other nucleoprotein assembly observed in the cryo-EM experiment could account for such an extended footprint. One possible explanation for how the double OCCM forms is that it arises from the encounter of mobile single OCCM assemblies. Consistent with the idea that the OCCM is mobile, Fig. 3k shows that, whereas the hDH footprint was still seen after 20 min digestion, the hOCCM footprint disappeared by 20 min digestion, and this was not owing to instability of OCCM (Extended Data Fig. 1n). By contrast, the yDH footprint had disappeared after 20 min of benzonase digestion, whereas the yOCCM footprint remained after 20 min digestion. Together, these results indicate that hDH is immobile and hOCCM is mobile; by contrast, yDH is mobile and yOCCM is immobile. Extensive negative-stain electron microscopy analysis of MCM loading reactions detected double OCCM in the presence of ATP (Supplementary Fig. 2a). This species was much less frequently observed compared with the ATPγS condition, suggesting that it might be a transient intermediate that matures to fully loaded double hexamer upon ATP hydrolysis (Supplementary Fig. 2a). A second notable assembly was an OCCM-like particle mapping next to a human single hexamer (hSH), similar to observations made when the yDH loading reaction was crosslinked with glutaraldehyde[30] (Supplementary Fig. 2a).

## Fully loaded single MCM hexamers

Two fully loaded hSH species were identified in both the recruitment (ATPγS) and loading (ATP) datasets. One species is the isolated hSH topologically bound to duplex DNA, which we solved to 3.2 Å resolution from the ATP dataset, yielding a map of sufficient quality to distinguish ADP from ATP within the catalytic centres (Fig. 4a, Extended Data Figs. 6a–c and 7a). The 3.4 Å structure obtained from ATPγS data (Extended Data Fig. 5) presents no obvious difference

and is not discussed further. Unlike for the OCCM complex, both the N- and the C-terminal tiers are well-resolved and the MCM2–MCM5 gate is completely shut. The DNA grip of the PS1 pore loops matches that observed for the hDH, with MCM7, MCM4 and MCM6 engaging the leading strand template (Fig. 4b). The path of the double helix is less distorted compared with the hDH structure, which positions L209 too far to contact the double helix and R195 retracted and engaged to a DNA-backbone phosphate. As a result, no DNA untwisting or melting can be observed in the hSH structure (Fig. 4c). Despite this difference, nucleotide occupancy matches that of the hDH, with one apo site (MCM7–MCM4), one site with ATP bound and all other sites bound to ADP (Fig. 4a).

A second assembly contains a fully loaded hSH bound to the ORC (Fig. 4d, Extended Data Figs. 5 and 6d–f). In this structure, ORC1–5 (solved to 4 Å resolution by local refinement; Extended Data Fig. 6g,h) binds and bends DNA. Within the MCM ring (solved to 3.5 Å resolution as a separate body; Extended Data Fig. 6i,j), the DNA grip appears indistinguishable from that of the isolated hSH. ORC6 is essential for the formation of this intermediate (Supplementary Fig. 1) and bridges between ORC1–5 and the N-terminal side of MCM. Here, the N-terminal cyclin box domain of ORC6 is sandwiched between ORC4 and the N-terminal zinc-finger domains of MCM2 and MCM6. The C-terminal cyclin box domain of ORC6 instead connects ORC5 with the zinc-finger domains of MCM4 and MCM6 (Extended Data Fig. 7b). A human-specific IDR in ORC6 would make it physically possible for a conserved C-terminal α-helix of ORC6 to contact ORC3, as previously observed in yeast[28,29] and fly[31]. Owing to limited local resolution, however, we were unable to determine whether this interaction exists in the human complex (Extended Data Fig. 7b). Given that the C-terminal α-helix is essential for ORC6 recruitment to the ORC in yeast[28,29] and metazoans[31], we consider it likely that this interaction exists in the ORC6-dependent human MCM–ORC.

Time-resolved cryo-EM and single-molecule work with yeast proteins showed that the ORC sequentially binds first the C-terminal side (in OCCM) and then the N-terminal side (in the yeast MCM–ORC loading intermediate (yMO); Fig. 4e) of MCM. In yMO, Orc6 straddles across the closed Mcm2–Mcm5 gate (Extended Data Fig. 7b) of a yeast single hexamer (ySH) and orients the ORC so that it can load a ySH to form a yDH (Fig. 4f). In the human structure, the N-terminal cyclin box of ORC6 contacts MCM2, as seen in the yMO complex. The engagement of the N terminus of ORC6 with ORC4 is seen in the human complex only, and not in yMO. Also consolidated by other human-specific interactions (Extended Data Fig. 7b), this ORC6–ORC4 contact collapses the structure in a folded configuration in which recruitment of a second hSH is sterically impeded (Fig. 4f). To mark the structural differences with yMO, we refer to the human MCM–ORC complex as hMO*.

## Discussion

Although both yeast and humans use the ORC, CDC6 and CDT1 to load an MCM double hexamer around double stranded DNA, our work shows that there are clear differences in how the two organisms accomplish this feat. For example, CDT1 is pre-bound to MCM in yeast where it stabilizes an open form of the MCM hexamer, whereas human CDT1 does not associate with free MCM but instead joins the complex during MCM recruitment. Despite this difference, the interactions of CDT1 with MCM are very similar in hOCCM and yOCCM. In yeast, the initial recruitment of MCM to origins is dependent on interaction between CDC6 and a WH domain at the C terminus of MCM3[5]; this interaction is conserved in hOCCM, but CDC6 is not essential for MCM recruitment or loading, and interactions of ORC with WH domains in MCM6 and MCM7 stabilize the hOC$_1$M structure in the absence of CDC6.

ORC1–5 is required for loading, consistent with genetic analysis indicating that it is essential for human cell viability[32], although we cannot

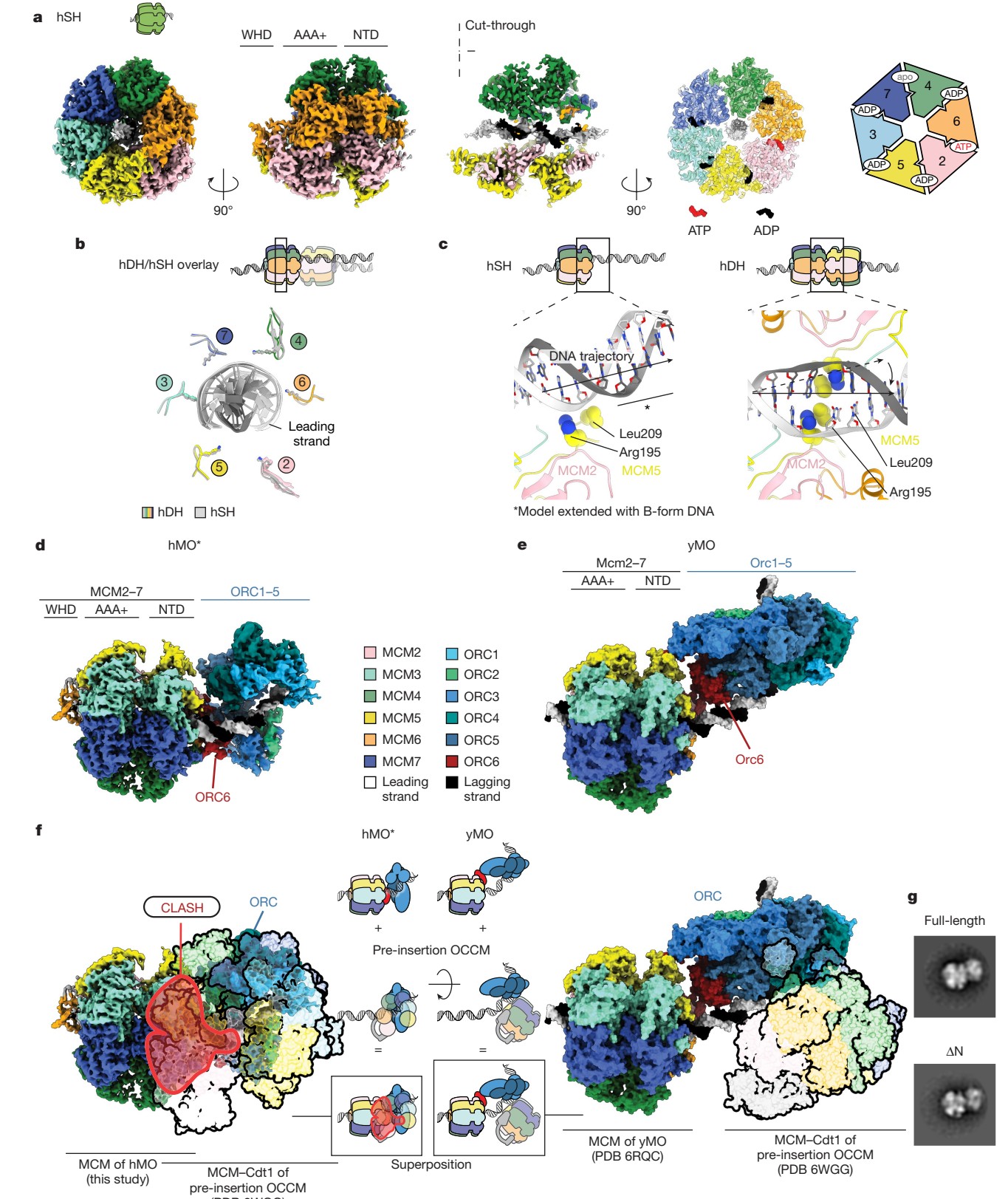

**Fig. 4 | Single-loaded MCM complexes. a**, Surface rendering and cut-through view of a hSH loaded onto duplex DNA. The C-terminal ATPase site is shown with nucleotides bound at subunit interfaces. **b**, A superposition of PS1 pore loops and DNA extracted from hDH and hSH show the same configuration. **c**, MCM hexamer dimerization changes the path of duplex DNA, leading to untwisting and opening of the double helix. **d**, Surface rendering of the hMO* structure. **e**, Surface rendering of the yMO structure highlights a different

configuration of the ORC compared with hMO*. ORC6 bridges N-terminal MCM and ORC1–5 in both structures. **f**, A steric clash obtained by modelling the pre-insertion yOCCM (PDB entry 6WGG) demonstrates that hMO* cannot support recruitment of a second MCM, unlike observations for the yMO complex. **g**, Truncated and full-length ORC1, CDC6 and CDT1 yield the same hMO* complex, as observed by negative-stain 2D averaging.

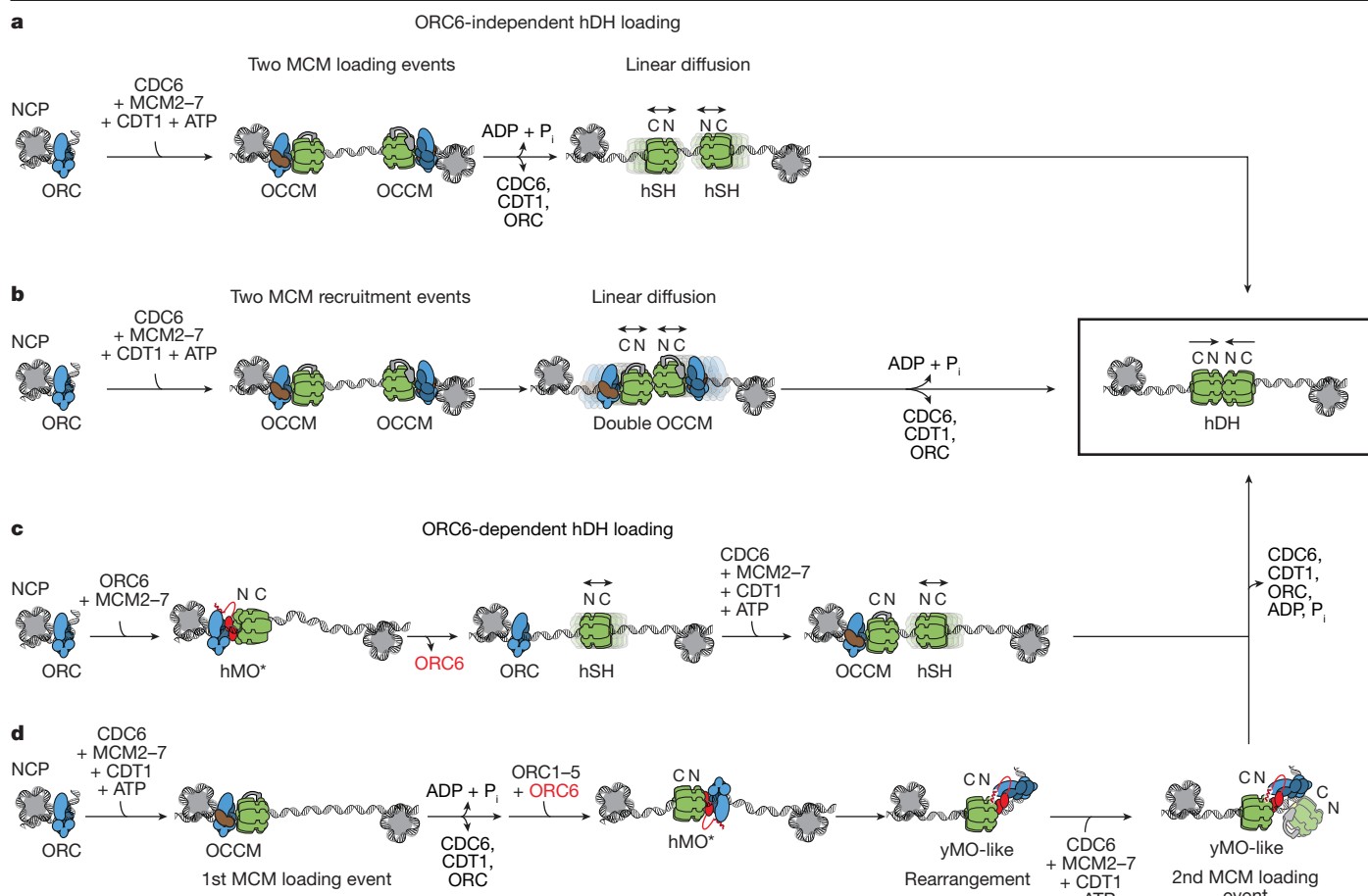

**Fig. 5 | Mechanisms of hDH loading.** Model of proposed ORC6-independent and ORC6-dependent MCM loading mechanisms. **a**, MCM can be loaded via an ORC6-independent pathway, via two inverted hOCCM complexes that load two hSHs in a process that requires ATP hydrolysis. Free diffusion along duplex DNA would then lead to hDH formation. **b**, In a variation of the same mechanism, two hOCCMs assembled around duplex DNA are free to diffuse and form a double OCCM. ATP hydrolysis then promotes hDH formation. **c**, MCM loading with full-length ORC1 can occur in an ORC6-dependent manner and might go through the hMO* intermediate. hMO* might recruit a hSH through an ORC6 interaction with N-terminal MCM. Following hSH release, the same ORC could recruit a second hSH via C-terminal MCM interaction (hOCCM intermediate), resulting in two N-terminally facing hSHs that can assemble a hDH. **d**, Alternatively, a first hSH could be loaded via OCCM. ORC6 would then interact with the N-terminal domain of hSH forming hMO*. A structural change would then occur, which causes hMO* to transition to a yMO-like state, in which the human ORC is competent for the recruitment of a second hSH, eventually leading to hDH formation. However, we did not observe formation of a yMO-like complex with the human proteins.

yet rule out the possibility that individual subunits within ORC1–5 may be dispensable for hDH formation[33,34]. ORC6 is required for yDH loading, but our results show that hDH is loaded in vitro via at least two distinct pathways, one that is ORC6-dependent and another that is ORC6-independent (Fig. 5). It is unclear whether both pathways are used in vivo: a Meier–Gorlin syndrome mutant that maps to the C-terminal α-helix of ORC6 that interacts with ORC3 and inhibits ORC6 interaction with ORC1–5 also inhibits MCM loading in *D. melanogaster*[35]; however, depletion of ORC6 from human U2OS cells was shown to have no effect on MCM loading levels in a single cell cycle[36], and an ORC6 knockout in glioma cells was observed to support cell viability and proliferation[37]. In yeast, CDK regulation of MCM loading works in part through phosphorylation of Cdc6 and Orc6[1–3]; however, these proteins are not essential for MCM loading with human proteins. This might explain why regulation of MCM loading in human cells works primarily through degradation and inhibition of CDT1[38], which is essential for all MCM loading pathways.

ORC6-dependent loading may occur through formation of the ORC6-dependent hMO* complex. hMO* contains a closed-ring, loaded hSH with its N-terminal domain bound to the ORC. hMO* was also seen in the absence of CDT1 and CDC6, suggesting that the hSH in hMO* may not arise from OCCM but may instead be loaded directly by the ORC alone.

Alternatively, if hSHs are formed via OCCM, hMO* is likely to form via direct N-terminal single-hexamer engagement by the ORC and not by flipping from the C-terminal side to the N-terminal side. This is because ORC6, which enables flipping from OCCM to yMO in the yeast system, is not seen in hOCCM. ORC6 in hMO* engages the N-terminal domains of MCM2, MCM4 and MCM6; this is the same side of MCM that is bound by CDT1 in the yeast MCM–CDT1 loading intermediate (Extended Data Fig. 7b), which promotes loading of a single MCM hexamer in yeast. Loading of hDH, however, still requires CDT1, indicating that at least one hSH in the hDH must be loaded via OCCM. It remains to be established whether hSHs loaded via the ORC alone and those loaded via the OCCM pathway differ in some way, for example nucleotide binding. In fact, hMO* and hSH can be observed in the presence of ATPγS with the human proteins, whereas OCCM maturation requires ATP hydrolysis, at least in yeast[25,39].

ySH has a short lifetime on DNA[9], so it is possible that the role of the ORC in hMO* is to stabilize the hSH on DNA. The position of the ORC at the N terminus of the hSH in hMO*, however, would block the recruitment of a second hexamer as seen in yMO (Fig. 4f) and block hDH assembly via a second hSH. Therefore, for hMO* to be a fruitful intermediate, the ORC must somehow release its grip on MCM. Because of how the N- and C-terminal cyclin boxes are arranged in ORC6, the human

MCM and ORC do not recapitulate the exact same architecture seen in yMO. The hMO* structure could however transition from the collapsed state described here to a more open configuration similar to yMO, although we did not observe such a structure. The ORC1 IDR is critical for ORC6-dependent loading but is not required for hMO* formation, and could have a role in this hMO* rearrangement, creating the space for ORC-mediated recruitment of a second hSH. Alternatively, the IDR of ORC1 could act on hMO* by disengaging the ORC from MCM, perhaps prompted by interaction with a second, correctly oriented hSH loaded from OCCM. Such a role for the ORC1 IDR could also explain why ORC6 inhibits MCM loading with truncated proteins: in these reactions hMO* is a dead-end complex that cannot be converted to hDH because the ORC cannot be released from hMO*.

ORC6-independent hDH loading may occur via the locking of two oppositely oriented and independently loaded hSH (which may or may not require ATP hydrolysis). This is fundamentally similar to assembly of yDH from two high-affinity binding sites when yMO formation is blocked—for example, by CDK phosphorylation of ORC2 (ref. 9)—except that there are no specific sequences to correctly orient hexamer loading. This might appear wasteful, as only 25% of randomly loaded MCM pairs would be predicted to be in the correct orientation to assemble the double hexamer. Nonetheless, hDH assembly in vitro by the ORC containing truncated ORC1 and lacking ORC6 is quite efficient; amounts of hDH assembled are similar to amounts of yDH assembled from specific DNA sequences under similar conditions (for example, in Fig. 1c). A second possible mechanism capitalizes on the fact that the human ORC is not a sequence-specific DNA-binding protein: because the human ORC is not anchored at a specific sequence, intermediates such as OCCM are not static, in contrast to yeast, but rather can diffuse on DNA. Loading of hDH via a double OCCM could, for example, guarantee correct orientation of two hexamers before loading. The ORC6-independent pathway is inefficient with wild-type, full-length ORC1. This may indicate that the role of the ORC1 IDR is to prevent ORC6-independent hDH loading, although we do not rule out the possibility that the other IDRs contribute to what pathway is chosen. Alternatively, some post-translational modification may counteract ORC1 IDR inhibition, allowing ORC6-independent loading to occur in a regulated manner.

Although the structures of hDH and yDH are very similar, their engagement with DNA exhibits two major differences. Firstly, hDH has a 5-bp stretch of underwound DNA between the hexamers, with one broken base pair stabilized by R195 and L209 of MCM5, which pin the double helix and are not required for untwisting. hDH containing two hexamers with the R195A/L209G mutations could not be isolated from chromatin when expressed in cells, suggesting that bubble formation may be required for hDH assembly[4]. Our results indicate that this mutant can form hDH, but the hDH formed is mobile on DNA. Thus, the inability to detect the mutant in cells may reflect the loss of bound DNA after benzonase treatment during purification, which might render the mutant hDH unstable. The second major difference is that DNA engagement by yDH within the central channel is displaced by approximately 5 bp relative to hDH. In comparing the DNA grip of various MCM structures, yDH emerges as an outlier: ySH, hSH and hMO* all grip the DNA in a very similar manner to hDH. DNA slippage within the central channel during yDH loading might dissipate torsional strain. DNA in the yMO is already in the same configuration as yDH, indicating that this slippage can happen before the final steps in yDH formation. In yeast, origin melting begins within the AAA+ domains of each hexamer during helicase activation, not at the interface between hexamers[40]. It will be interesting to identify where melting begins during human helicase activation.

Our results show that hDH can be assembled efficiently in the absence of both human origin sequences and nucleosomes. Thus, any nucleosome-free stretch of DNA of appropriate length in the genome may be sufficient for hDH assembly and origin function in vivo. Efficient assembly of a hDH that is nearly identical to the hDH isolated from cells represents a first step toward reconstitution of DNA replication with purified human proteins.

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

# Methods

## Protein sequences

Human cDNAs were cloned from RPE-1 and U2OS cells by total RNA isolation, oligo-dT reverse transcription, followed by PCR with gene-specific oligonucleotides and cloning into expression vectors. Sanger sequencing showed that the cloned cDNA coded for the reference protein sequence for the following proteins: ORC1, ORC2, ORC3 (NCBI: iso2, Uniprot: canonical iso), ORC4, ORC5, ORC6, MCM2, MCM3, MCM5, MCM6, MCM7, CDC6 and GMNN. For MCM4 and CDT1, the cloned cDNA coded for a natural variant that differed from NCBI reference protein sequences, but according to the Genome Aggregation Database (https://gnomad.broadinstitute.org) represented the majority allele (allele frequency > 0.5): MCM4_L650M (allele frequency: 0.862), CDT1_C234R (allele frequency: 0.999). These alleles were used and considered wild type throughout this study.

## Expression and purification of human MCM2–7

MCM2–7 was expressed in insect cells using the biGBac baculovirus expression vector system[10]. Human cDNAs of MCM2, MCM3, MCM4, MCM5, MCM6 and MCM7 were cloned into pLIB. MCM3 was subcloned to contain an N-terminal TEV protease-cleavable Flag tag. pLIB-derived expression cassettes of MCM4, MCM5, MCM6, MCM7 were subcloned into pBIG1a (pBIG1a:MCM4, MCM5, MCM6, MCM7) and expression cassettes of MCM2 and Flag–MCM3 were subcloned into pBIG1b (pBIG1b:MCM2, Flag–MCM3). Expression cassettes of these two vectors were subcloned into pBIG2ab (pBIG2ab:MCM4, MCM5, MCM6, MCM7, MCM2, Flag–MCM3). Baculovirus was generated and amplified in Sf9 cells (Thermo Fisher, 12659017) using the EMBacY baculoviral genome[41]. For protein expression, Sf9 cells were infected and collected 52 h after infection, flash-frozen and stored at −80 °C.

Cell pellets were thawed on ice in MCM buffer (50 mM HEPES/KOH pH 7.6, 100 mM potassium glutamate, 5 mM magnesium acetate, 10% glycerol, 0.02% NP-40, 1 mM dithiothreitol (DTT)) + protease inhibitors (1 tablet per 50 ml Roche Complete Ultra EDTA-free, 10 µg ml⁻¹ leupeptin, 10 µg ml⁻¹ pepstatin A, 1 mM AEBSF, 1 µg ml⁻¹ aprotinin, 2 mM benzamidine). Cells were resuspended and lysed by Dounce homogenization using a tight-fitting pestle. The lysate was cleared by centrifugation (158,000$g$, 4 °C, 1 h) and incubated with anti-Flag M2 affinity gel (Sigma-Aldrich, A2220) for 1 h at 4 °C with rotation. The beads were transferred to a Econo-Pac gravity flow column (Bio-Rad, 732-1010) and washed with 2× 20 column volumes MCM buffer, 1× 20 column volumes MCM buffer + 5 mM ATP (Sigma-Aldrich, A2383). For dephosphorylation, the beads were resuspended in MCM buffer + 0.2 mg ml⁻¹ lambda protein phosphatase, 1 mM manganese (II) chloride and incubated for 1 h at 4 °C. The beads were washed with 2× 20 column volumes MCM buffer. For elution and proteolytic tag removal the beads were resuspended in 5 column volumes MCM buffer + 80 µg ml⁻¹ TEV protease, 0.1 mg ml⁻¹ 3×Flag peptide for 2 h at 4 °C. The eluate was concentrated using an Amicon Ultra-15 concentrator (Merck, UFC903024) and further purified by gel filtration using a HiLoad 16/600 Superdex 200 pg column equilibrated in MCM buffer. Fractions containing stoichiometric MCM2–7 were concentrated, flash-frozen in liquid nitrogen, and stored at −80 °C.

## Expression and purification of human ORC1–5

ORC1–5 was expressed in baculovirus-infected insect cells. The coding sequences of human ORC1, ORC2, ORC3, ORC4 and ORC5 were codon-optimized for *Spodoptera frugiperda*, synthesized (GeneArt, Thermo Fisher Scientific) and subcloned into modified pBIG1 vectors that contain a pLIB-derived polyhedrin expression cassette. ORC1 was subcloned into pBIG1a with a TEV protease-cleavable N-terminal 3×Flag tag. ORC2 was subcloned into pBIG1b, ORC3 into pBIG1c, ORC4 into pBIG1d, ORC5 into pBIG1e. Expression cassettes from these five vectors were subcloned into pBIG2abcde (pBIG2abcde:Flag-ORC1,2,3,4,5).

Baculoviruses were generated using EMBacY and Sf9 cells. To express ORC1–5, Sf9 cells were infected, collected 52 h after infection, flash-frozen and stored at −80 °C.

Cell pellets were thawed in ORC1–5 lysis buffer (50 mM HEPES/KOH 7.6, 650 mM potassium chloride, 5 mM magnesium acetate, 1 mM ATP, 10% glycerol, 0.02% NP-40, 1 mM DTT, 2 mM benzamidine) + protease inhibitors (1 tablet per 50 ml Roche Complete Ultra EDTA-free, 10 µg ml⁻¹ leupeptin, 10 µg ml⁻¹ pepstatin A, 1 mM AEBSF, 1 µg ml⁻¹ aprotinin). Cells were lysed by Dounce homogenization, and the lysate cleared by centrifugation (158,000$g$, 4 °C, 1 h). ORC1–5 was bound to anti-Flag M2 affinity gel for 1 h at 4 °C. The column was washed with 2× 20 column volumes ORC1–5 lysis buffer + 4 mM ATP. For dephosphorylation, the beads were resuspended in 1 column volumes ORC1–5 lysis buffer + 0.2 mg ml⁻¹ lambda protein phosphatase, 1 mM manganese (II) chloride, potassium chloride adjusted to 650 mM, and incubated at 4 °C for 1 h. The column was washed with ORC1–5 lysis buffer. For elution and proteolytic tag removal, the beads were resuspended in 5 column volumes ORC1–5 lysis buffer + 80 µg ml⁻¹ TEV protease, 0.1 mg ml⁻¹ 3×Flag peptide, potassium chloride adjusted to 650 mM and were incubated at 4 °C for 2 h. To remove TEV protease, the eluate was supplemented with 35 mM imidazole pH 8.0 and incubated with Ni-NTA Agarose (Invitrogen, R90115) for 1 h at 4 °C. The flowthrough was concentrated and further purified by gel filtration using a HiLoad 16/600 Superdex 200 pg column (Cytiva) equilibrated in ORC1–5 SEC buffer (50 mM HEPES/KOH pH 7.6, 650 mM potassium chloride, 5 mM magnesium acetate, 10% glycerol, 0.02% NP-40, 1 mM DTT). Fractions containing stoichiometric ORC1–5 were concentrated, flash-frozen, and stored at −80 °C.

## Expression and purification of human CDC6

CDC6 was expressed in insect cells. Human cDNA of CDC6 was cloned into pLIB to contain an N-terminal TEV protease-cleavable Flag tag. The baculovirus was generated in Sf9 cells using the EMBacY genome. For expression, Sf9 cells were infected and the culture collected 52 h after infection. The cell pellet was flash-frozen in liquid nitrogen and stored at −80 °C.

The cell pellet was thawed on ice in CDC6 lysis buffer (50 mM HEPES/KOH pH 7.6, 650 mM potassium chloride, 5 mM magnesium acetate, 4 mM benzamidine, 1 mM ATP, 10% glycerol, 0.02% NP-40, 1 mM DTT) + protease inhibitors (1 tablet per 50 ml Roche Complete Ultra EDTA-free, 10 µg ml⁻¹ leupeptin, 10 µg ml⁻¹ pepstatin A, 1 mM AEBSF, 1 µg ml⁻¹ aprotinin). Cells were lysed with a Dounce homogenizer and the lysates centrifuged (158,000$g$, 4 °C, 1 h). The cleared lysate was incubated with anti-Flag M2 affinity gel for 1 h at 4 °C. The beads were washed with 2× 20 column volumes CDC6 lysis buffer + 4 mM ATP. For dephosphorylation, the beads were resuspended in CDC6 lysis buffer + 0.2 mg ml⁻¹ lambda protein phosphatase, 1 mM manganese (II) chloride and incubated for 1 h at 4 °C. The column was washed with CDC6 lysis buffer, followed by CDC6 HTP-wash buffer (50 mM potassium phosphate pH 7.6, 75 mM potassium acetate, 5 mM magnesium acetate, 0.1% Triton X-100, 1 mM DTT, 2 mM ATP). For elution, the beads were resuspended in CDC6 HTP-wash buffer + 80 µg ml⁻¹ TEV protease, 0.1 mg ml⁻¹ 3×Flag peptide and incubated for 2 h at 4 °C. To remove TEV protease, the eluate was supplemented with 30 mM imidazole pH 8.0 and incubated with Ni-NTA agarose for 1 h at 4 °C. The CDC6-containing flowthrough was collected. A hydroxyapatite column was prepared by resuspending 2 g Bio-Gel HTP Hydroxyapatite (Bio-Rad, 130-0420) in 12 ml CDC6 HTP-wash buffer. To remove fine particles, the beads were allowed to settle for 2 min and the supernatant was removed. Two more times, the beads were resuspended in 10 ml CDC6 HTP-wash buffer, allowed to settle for 2 min, and the supernatant removed. Then, 4 ml of a 50% slurry were incubated with CDC6 for 15 min at 4 °C. The beads were transferred to a gravity flow column and washed with 2 ml CDC6 HTP-wash buffer, followed by 5 ml CDC6 HTP-rinse buffer (50 mM potassium phosphate pH 7.6, 150 mM potassium acetate, 5 mM magnesium acetate, 0.1% Triton X-100, 15% glycerol, 1 mM DTT). CDC6 was

eluted by applying 10 ml CDC6 HTP-elution buffer (50 mM potassium phosphate pH 7.6, 400 mM potassium acetate, 5 mM magnesium acetate, 0.1% Triton X-100, 15% glycerol, 1 mM DTT). The eluate was dialysed 2×1 h at 4 °C against CDC6 dialysis buffer (50 mM HEPES/KOH pH 7.6, 650 mM potassium chloride, 5 mM magnesium acetate, 10% glycerol, 0.02% NP-40, 1 mM DTT). CDC6 was concentrated, flash-frozen, and stored at −80 °C.

## Expression and purification of human CDT1

CDT1 was expressed in insect cells. The coding sequence of human CDT1 was codon-optimized for *S. frugiperda*, synthesized (GeneArt, Thermo Fisher Scientific) and subcloned into pLIB as a fusion protein with an N-terminal Flag–His–SumoEu1 fusion[42]. The baculovirus was generated in Sf9 cells using the EMBacY genome. Expression cultures were collected 52 h after infection, snap-frozen in liquid nitrogen and stored at −80 °C.

The cell pellet was thawed in CDT1 buffer (50 mM HEPES pH 7.6, 650 mM potassium chloride, 5 mM magnesium acetate, 10% glycerol, 0.02% NP-40, 1 mM DTT) + protease inhibitors (1 tablet per 50 ml Roche Complete Ultra EDTA-free, 10 µg ml$^{-1}$ leupeptin, 10 µg ml$^{-1}$ pepstatin A, 1 mM AEBSF, 1 µg ml$^{-1}$ aprotinin). Cells were lysed using a Dounce homogenizer. The lysate was cleared by centrifugation (158,000$g$, 4 °C, 1 h) and incubated with anti-Flag M2 affinity gel for 1 h at 4 °C. The beads were transferred to a gravity flow column and washed twice with CDT1 buffer + 5 mM ATP. The beads were resuspended in CDT1 buffer + 0.2 mg ml$^{-1}$ lambda protein phosphatase, 1 mM manganese (II) chloride and incubated for 1 h at 4 °C. The beads were washed with CDT1 buffer. For proteolytic elution the beads were resuspended in CDT1 buffer + 80 µg ml$^{-1}$ His-SENP1_EuH protease[42] and incubated for 2 h at 4 °C. To remove His-SENP1_EuH protease, 35 mM imidazole pH 8.0 was added and the eluate incubated with Ni-NTA agarose for 1 h at 4 °C. The CDT1-containing flowthrough was concentrated and further purified by gel filtration on a Superdex 200 Increase 10/300GL column (Cytiva) using CDT1 buffer. CDT1-containing fractions were concentrated, snap-frozen, and stored at −80 °C.

## Expression and purification of human ORC6

ORC6 was expressed in *Escherichia coli*. The coding sequence of human ORC6 was codon-optimized for *E. coli*, synthesized (GeneArt, Thermo Fisher Scientific) and subcloned into pK27Sumo to encode the fusion protein His–Sumo–ORC6. The protein was expressed using the strain T7express lysY (NEB, C3010I). Expression was induced with 0.4 mM IPTG at OD600 ~ 0.6 at 16 °C. The culture was collected after 16 h, flash-frozen, and stored at −80 °C.

The cell pellet was thawed in ORC6 lysis buffer (50 mM HEPES pH 7.6, 500 mM potassium chloride, 5 mM magnesium acetate, 10% glycerol, 0.02% NP-40, 1 mM DTT, 35 mM imidazole) + protease inhibitors (1 tablet per 50 ml Roche Complete EDTA-free, 1 mM AEBSF), lysozyme. The cells were resuspended and lysed by sonication. The lysate was centrifuged (158,000$g$, 4 °C, 1 h) and the cleared lysate incubated with Ni-NTA agarose (Invitrogen) for 1 h at 4 °C. The beads were transferred to a gravity flow column and washed twice with ORC6 lysis buffer + 5 mM ATP and twice with ORC6 lysis buffer. The protein was eluted with ORC6 elution buffer (50 mM HEPES/KOH pH 7.6, 200 mM potassium chloride, 5 mM magnesium acetate, 10% glycerol, 0.02% NP-40, 1 mM DTT, 300 mM imidazole pH 8.0). The Sumo-specific protease His-Ulp1 was added at 0.08 mg ml$^{-1}$ and the protein was dialysed overnight against ORC6 SEC buffer (50 mM HEPES/KOH pH 7.6, 200 mM potassium chloride, 5 mM magnesium acetate, 10% glycerol, 0.02% NP-40, 1 mM DTT). To remove His-Ulp1 protease and His–Sumo, the imidazole concentration was adjusted to 35 mM and the protein incubated with Ni-NTA agarose (Invitrogen) for 1 h at 4 °C. The ORC6 containing flowthrough was concentrated and further purified by gel filtration using a HiLoad 16/600 Superdex 200 pg column (Cytiva) equilibrated in ORC6 SEC buffer. ORC6 containing fractions were pooled, snap-frozen, and stored at −80 °C.

## Expression and purification of truncated loading factors

The ORC1–5 complex with ORC1(ΔN) (ORC1–5(1ΔN)), and CDC6(ΔN) and CDT1(ΔN) with N-terminal IDR truncations were expressed in insect cells using a Flag–His–SumoEu1 fusion system[42] and sequences that were codon-optimized for *S. frugiperda* and synthesized (GeneArt, Thermo Fisher Scientific). The SumoEu1 fusions were only partially stable in Sf9 cells giving relatively low yields. Flag–His–SumoEu1–ORC1(391–861) was cloned into pBIG1a containing a pLIB-derived polyhedrin expression cassette. The expression cassette was subcloned together with ORC2–5 expression cassettes into pBIG2abcde (pBIG2abcde:Flag–His–SumoEu1–ORC1(ΔN), ORC2–5). Flag–His–SumoEu1–CDC6(ΔN) (CDC6 residues 143–560) and Flag–His–SumoEu1–CDT1(ΔN) (CDT1 residues 167–546) were cloned into pLIB. Baculoviruses were generated in Sf9 cells using EMBacY. Sf9 expression cultures were collected 52 h after infection, flash-frozen and stored at −80 °C.

The three proteins were purified with the same purification protocol with only the gel filtration column differing. Cell pellets were thawed in Wash-300 buffer (50 mM HEPES/KOH pH 7.6, 300 mM potassium chloride, 5 mM magnesium acetate, 1 mM ATP, 10% glycerol, 0.02% NP-40, 1 mM DTT) + protease inhibitors (1 tablet per 50 ml Roche Complete Ultra EDTA-free, 10 µg ml$^{-1}$ leupeptin, 10 µg ml$^{-1}$ pepstatin A, 1 mM AEBSF, 1 µg ml$^{-1}$ aprotinin, 4 mM benzamidine) and lysed using a Dounce homogenizer. The lysate was centrifuged (158,000$g$, 4 °C, 1 h) and the cleared lysate incubated with anti-Flag M2 affinity gel for 2 h at 4 °C. The beads were washed twice with Wash-300 buffer + 4 mM ATP, and then resuspended in Wash-300 buffer + 0.2 mg ml$^{-1}$ lambda protein phosphatase, 1 mM manganese (II) chloride and incubated for 1 h at 4 °C. The beads were washed with Wash-300 buffer. For proteolytic elution, the beads were resuspended in Wash-300 buffer + 80 µg ml$^{-1}$ His-SENP1_EuH protease[42] and incubated for 2 h at 4 °C. To remove His-SENP1_EuH protease, the eluate was supplemented with 35 mM imidazole pH 8.0 and incubated with Ni-NTA agarose (Invitrogen) for 1 h at 4 °C. The flowthrough was concentrated and further purified by gel filtration using SEC-ΔN buffer (50 mM HEPES/KOH pH 7.6, 500 mM potassium glutamate, 5 mM magnesium acetate, 10% glycerol, 0.02% NP-40, 1 mM DTT). ORC1–5(1ΔN) was purified using a HiLoad 16/600 Superdex 200 pg column (Cytiva). CDC6(ΔN) and CDT1(ΔN) were purified using a Superdex 200 Increase 10/300GL column (Cytiva). Fractions containing the respective protein were concentrated, snap-frozen, and stored at −80 °C.

## Expression and purification of human geminin

Geminin was expressed in insect cells. Human cDNA of geminin was cloned into pLIB with an N-terminal Flag tag. The baculovirus was generated in Sf9 cells using EMBacY. Geminin was expressed in Sf9 cells and the culture collected 52 h after infection. The cell pellet was stored at −80 °C.

The cell pellet was thawed in GMNN lysis buffer (50 mM HEPES/KOH pH 7.6, 300 mM potassium chloride, 5 mM magnesium acetate, 10% glycerol, 0.02% NP-40, 1 mM DTT) + protease inhibitors (1 tablet per 50 ml Roche Complete Ultra EDTA-free, 10 µg ml$^{-1}$ leupeptin, 10 µg ml$^{-1}$ pepstatin A, 1 mM AEBSF, 1 µg ml$^{-1}$ aprotinin) and lysed using a Dounce homogenizer. The lysate was cleared by centrifugation (158,000$g$, 60 min, 4 °C) and incubated with anti-Flag M2 affinity gel for 1 h. The beads were washed with GMNN lysis buffer + 5 mM ATP, followed by GMNN lysis buffer. Protein was eluted in GMNN lysis buffer + 0.1 mg ml$^{-1}$ 3×Flag peptide. The protein was concentrated and further purified by gel filtration on a Superdex 200 Increase 10/300GL column equilibrated in GMNN SEC buffer (50 mM HEPES/KOH pH 7.6, 200 mM potassium chloride, 5 mM magnesium acetate, 10% glycerol, 0.02% NP-40, 1 mM DTT). Geminin containing fractions were concentrated and the protein concentration determined for the homodimer. The protein was flash-frozen, and stored at −80 °C.

## Mass spectrometry

Protein preparations of human MCM2–7, ORC1–5 (FL), ORC1–5(1ΔN), CDC6 (FL), CDC6(ΔN), CDT1 (FL) and CDT1(ΔN) were subjected to mass spectrometry. Insect cell ORC6 and CDC6 were not detected in any purification.

## Nuclease footprinting assay with human proteins

MCM loading reactions were performed in assay buffer (25 mM HEPES/KOH pH 7.6, 100 mM sodium acetate, 10 mM magnesium acetate, 1 mM DTT). MCM was typically loaded onto a ARS1-containing 10.6 kb plasmid that was purified by caesium chloride density gradient centrifugation (pJY22[43]).

A 20 µl reaction with truncated proteins typically contained 4 nM (27.5 ng µl$^{-1}$) plasmid DNA (10.6 kb), 2 mM ATP, 60 nM MCM2–7, 150 nM ORC6, 150 nM CDC6(ΔN), 150 nM CDT1(ΔN), and 90 nM ORC1–5(1ΔN). Stocks of MCM2–7, CDC6(ΔN), CDT1(ΔN), and ORC1–5(1ΔN) were diluted to a 10× working concentration in assay buffer. ORC6 was diluted to a 20× working concentration in assay buffer + 200 mM sodium chloride. Reactions were started by adding MCM2–7, ORC6, CDC6(ΔN), CDT1(ΔN) and ORC1–5(1ΔN) to a mix of DNA and ATP in assay buffer.

A 20 µl reaction with full-length proteins typically contained 4 nM (27.5 ng µl$^{-1}$) plasmid DNA (10.6 kb), 2 mM ATP, 60 nM MCM2–7, 150 nM ORC6, 120 nM ORC1–5, 150 nM CDC6, and 150 nM CDT1. MCM2–7 was diluted to 10× working concentration in assay buffer. ORC6 was diluted to 20× working concentration in assay buffer + 200 mM sodium chloride. A 20× OCC mix of ORC1–5, CDC6 and CDT1 was prepared in assay buffer + 650 mM sodium chloride. Reactions were started by adding MCM2–7, ORC6, and the OCC mix to a mix of DNA and ATP in assay buffer.

In experiments that contained truncated proteins as well as their full-length counterparts, ORC1–5 and ORC1–5(1ΔN) were both used at 120 nM and the 20× OCC mix (650 mM sodium chloride) method was used. When proteins were omitted, a salt-containing buffer was added instead to maintain the same salt concentration throughout the experiment.

Reactions were started in 2-min intervals and incubated at 37 °C in a thermomixer with shaking at 1,250 rpm. After 30 min of MCM loading, 2 µl of benzonase nuclease (Sigma-Aldrich, E1014) was added and the mix incubated at 37 °C with shaking for either 0.5 min or 20 min depending on the experiment. Then, 20 µl of the mix was transferred to a tube containing 10 µl 3× Stop buffer (assay buffer + 100 mM EDTA, 500 µM proteinase K (Sigma-Aldrich, 107393), 1% SDS) and the mix incubated at 37 °C for 20 min with shaking.

The sample was diluted with TE (10 mM Tris/HCl pH 8.0, 1 mM EDTA) to 200 µl and an equal volume of phenol/chloroform/isoamyl alcohol (25/24/1; Invitrogen UltraPure, 15593031) was added. The sample was vortexed for 1 min, transferred to a 5PRIME Phase Lock Gel Heavy spin column (VWR, 733-2478), and spun for 5 min at 20,000g. The aqueous phase was transferred to a new tube, 20 µl of 3 M sodium acetate pH 5.2, 1 µl of 20 mg ml$^{-1}$ glycogen (Thermo Scientific, R0561), and 550 µl ethanol were added and DNA precipitated overnight at −20 °C. The DNA was pelleted (20,000g, 4 °C, 40 min), washed with 80% ethanol (20,000g, 4 °C, 10 min), and air-dried. The pellet was resuspended in 3 µl TE. Then, 1.5 µl 20% Ficoll 400 (Sigma-Aldrich, F2637) was added, and samples were loaded on a pre-run 4–20% Novex TBE gel (Invitrogen, EC62252BOX). The gel was run using TBE running buffer (Invitrogen, LC6675) at 150 V for 50 min, stained with SYBR Gold (Invitrogen, S11494) for 30 min, and imaged using an Amersham ImageQuant 800 imager.

## Salt stability experiments

For the salt stability experiment in Fig. 1j, the nuclease footprinting assay was performed with the following modifications. MCM loading reactions (20 µl) were set up in assay buffer and were incubated (37 °C,

30 min, 1,250 rpm). Then, 10 µl of 3× sodium chloride in 1× assay buffer solutions were added to achieve the indicated sodium chloride concentrations, and the reaction was incubated (37 °C, 15 min, 1,250 rpm). The reactions were diluted to a volume of 200 µl with solutions that adjusted all reaction buffers to 25 mM HEPES/KOH pH 7.6, 20 mM sodium acetate, 150 mM sodium chloride, 10 mM magnesium acetate, 1 mM DTT. Immediately, 10 µl of benzonase nuclease (Sigma-Aldrich, E1014) was added and the mix incubated (37 °C, 0.5 min, 1,250 rpm). Then, 200 µl were transferred into 100 µl 3× Stop buffer and the mix incubated (37 °C, 20 min, 1,250 rpm). The samples were extracted with an equal volume of phenol−chloroform−isoamyl alcohol. The aqueous phase was diluted 3-fold with TE, and DNA precipitated with 50% isopropanol, 200 mM sodium chloride, 20 µg glycogen, followed by an 80% ethanol wash. The salt stability experiments in Extended Data Fig. 1e,f were performed in a similar way with the following modifications. MCM was loaded for 30 min in a single large reaction and split into two tubes before adding assay buffer or a 3× sodium chloride in 1× assay buffer solution to achieve the indicated salt concentration. Samples were taken at the indicated timepoints, diluted, treated with benzonase, and transferred into Stop buffer. For the salt stability experiments in Fig. 3g and Extended Data Fig. 1g, MCM was loaded using the indicated reaction conditions for 30 min. Then, assay buffer or NaCl-containing assay buffer was added to achieve the indicated salt concentration. After 15 min incubation, samples were diluted, treated with benzonase for 0.5 min and transferred into Stop buffer.

## Geminin inhibition experiments

For the geminin inhibition experiment shown in Fig. 1i, geminin and CDT1 were mixed on ice prior to setting up the MCM loading reactions. Geminin/CDT1 mixes were prepared at 20× concentration in assay buffer + 200 mM sodium chloride. Reactions were then started by adding MCM (10×, assay buffer), ORC6 (20×, 200 mM NaCl), geminin-CDT1 mix (20×, 200 mM NaCl), and ORC1–5/CDC6 mix (40×, 650 mM NaCl) to a mix of DNA and ATP in assay buffer. For the ATPγS timecourse experiment in Extended Data Fig. 1m geminin was added at minute 9 at 300 nM (twofold excess over 150 nM CDT1(ΔN)).

## Nuclease footprinting assay with yeast proteins

Nuclease footprinting experiments with yeast proteins were performed at 30 °C instead of 37 °C. Reactions with yeast proteins contained 4 nM (27.5 ng µl$^{-1}$) pJY22 plasmid DNA (10.6 kb, caesium chloride purified), 2 mM ATP, 100 nM Mcm2–7/Cdt1, 40 nM yCdc6 and 40 nM Orc1–6. In experiments containing human and yeast reactions, both were carried out at 30 °C.

## Experiments with human origin sequences

A 2,398 bp fragment of the origin at the human lamin B2 (*LMNB2*) locus, and a 2,398 bp fragment of the origin at the human *MYC* locus were cloned into the vector pBIG1c. The *LMNB2* origin fragment was amplified from human genomic DNA by PCR using oligonucleotides hOri-LamB2-2.4_for (AACGCTCTATGGTCTAAAGATTTACTCAGCAGCC CGGTG) and hOri-LamB2-2.4_rev (AACCCCGATTGAGATATAGATTTTG AGAATTGAGTCTTTGGAAACACTAAG). The *MYC* origin fragment was amplified using oligonucleotides hOri-Myc_for (AACGCTCTATGG TCTAAAGATTTAAGCTTGTTTGGCCGTTTTAGGG) and hOri-Myc_rev (AACCCCGATTGAGATATAGATTTCTCGAGGCAGGAGGGGAG). The fragments were inserted into pBIG1c that was linearized with the restriction enzyme SwaI using Gibson assembly, and the constructs were sequence verified. Doubly biotinylated DNA fragments of 2,398 bp size were generated by PCR using 5′-biotinylated oligonucleotides (Integrated DNA Technologies): LamB2 ori using Bio-LamB2-2.4_for ([5′-biotin] ACTCAGCAGCCCGGTG) and Bio-LamB2-2.4_rev ([5′-biotin]TGAGAATTG AGTCTTTGGAAACACTAAG), Myc ori using Bio-Myc-2.4_for ([5′-biotin] AAGCTTGTTTGGCCGTTTTAGGG) and Bio-Myc-2.4_rev ([5′-biotin]CT CGAGGCAGGAGGGGAG), yeast Ars1 using Bio-Ars1-2.4_for ([5′-biotin]

GGTGGAGATATTCCTTATGGCATG) and Bio-Ars1-2.4_rev ([5′-biotin] GTAATTCGACCATTCCGACACAG) on pJY22, no origin (pET21a backbone) using Bio-pET21a-2.4_for ([5′-biotin]CCACAGGTGCGGTTGC) and Bio-pET21a-2.4_rev ([5′-biotin]TTCACCGTCATCACCGAAAC) on pET21a. The PCR products were column purified (QIAquick PCR purification kit, Qiagen) and purity confirmed by agarose gel electrophoresis. For the experiments shown in Extended Data Fig. 1h,i nuclease footprinting assays were performed using 4 nM doubly biotinylated PCR products and 400 nM streptavidin (Thermo Fisher, 434301).

## DNA templates for electron microscopy experiments

The DNA substrate used for electron microscopy imaging of wild-type MCM loading was modified from the pGC209 (ref. 44) construct, containing two inverted ACS sequences spaced by 70 bp, by adding Widom601 and Widom603 strong positioning sequences at both ends. The Widom sequences map 7 and 5 base pairs away from the inverted ACS sites, making the nucleosome-free region 148 base pairs long. For the experiment shown in Extended Data Fig. 1a, the pGC211 (ref. 44) construct, which has two inverted ACS sequences spaced by 90 bp, was modified to include Widom601 and Widom603 strong positioning sequences on both ends. The nucleosome-free region is 168 base pairs long in this construct. For MCM5(AG) mutant, Widom sequences were swapped for a suicide substrate for covalent M.HpaII methyltransferase binding (equally efficient at blocking double hexamer sliding[28]). The plasmids were synthesized by Eurofins and used for PCR amplification with the primer pairs NCP F/NCP R (for nucleosome reconstitution) or Gid70-MTRB F/Gid70-MTRB R (for methyltransferase capping).

Reconstitution of nucleosomes with yeast histones and preparation of HpaII-flanked origins was carried out as described[28]. In brief, amplified templates were purified by anion exchange chromatography on a 1 ml RESOURCE Q column (Cytiva), followed by ethanol precipitation. DNA pellets were resuspended in TE buffer, mixed with purified yeast histone octamers and subjected to dialysis with decreasing NaCl concentration to reconstitute nucleosomes[28,45]. The chromatinized construct was purified by size exclusion chromatography using a Superose 6 Increase 3.2/300 column (Cytiva). For the methyltransferase construct, DNA was incubated with M.HpaII in 1:6 molar ratio at 30 °C overnight in buffer 1 (50 mM potassium acetate, 25 mM Tris pH 7.5, 10 mM magnesium acetate, 1 mg ml⁻¹ bovine serum albumin (BSA), 150 μM S-adenosyl-methionine (NEB)). The M.HpaII–70bp–M.HpaII construct was isolated by anion exchange chromatography using a 1 ml RESOURCE Q column (Cytiva).

NCP F, 5′-(Des)-CGATAGAACTCGGGCCGCCCTGGAGAATCGCGG TGCCG-3′; NCP R, 5′-CCTGCACCCCAGGGACTTGAAGTAATAAGGAC-3′; Gid70-MTRB F, atatatCC*GGcctgtATCTCGATTTTTTTATGTTTAGTTT CGC; Gid70-MTRB R -TGGGCGCC*GGAACTGGGTGCTGTaTTTTTATG TTTAGTTCG; (Des), Desthiobiotin TEG; C*, 5-fluoro-2′-deoxycytosine.

## Human MCM loading for cryo-EM

For the hDH loading reaction, 45 nM of chromatinized DNA (nucleosome–Gid70–nucleosome) were incubated with 120 nM ORC1–5(1ΔN), 120 nM ORC6, 150 nM CDC6(ΔN), 150 nM CDT1(ΔN) and 60 nM MCM2–7 in EM buffer (25 mM HEPES-KOH pH 7.6, 100 mM potassium glutamate, 10 mM magnesium acetate, 1 mM DTT, 2 mM ATP) resulting in a final volume of 35 μl. Incubation was carried out for 30 min at 37 °C and 1,250 rpm constant mixing.

The MCM recruitment reaction was established by substituting ATP with ATPγS and a chromatinized origin concentration of 70 nM.

## Negative-stain electron microscopy reactions and imaging

The MCM loading reaction for the negative-stain experiment shown in Extended Data Fig. 1a was performed similarly as described above, but using 20 nM ORC1–5(1ΔN), 20 nM ORC6, 20 nM CDC6(ΔN), 40 nM MCM2–7, 40 nM CDT1(ΔN), 7.5 nM of chromatinized DNA (nucleosome–Gid90–nucleosome) in a total volume of 20 μl. Dropout experiments

shown in Supplementary Fig. 1 were carried out with the same sample concentrations as described for cryo-EM, but using a final volume of 20 μl per reaction, and omitting one factor at a time. Reactions were diluted twofold (Extended Data Fig. 1a) and 4-fold (Supplementary Fig. 1) in EM buffer where nucleotide was omitted. 300-mesh copper grids coated with a layer of continuous carbon (EM Resolutions, C300Cu100) were glow-discharged at 25 mA for 1 min using a GloQube Plus Glow Discharge System (Quorum), before applying 4 μl of the sample for 2 min. Grids were stained with two successive applications of 4 μl of 2% (w/v) uranyl acetate solution. Excess stain was blotted after 40 s using filter paper. Micrographs were collected using a FEI Tecnai G2 Spirit transmission electron microscope operated at 120 keV, equipped with a 2 K x 2 K GATAN UltraScan 1000 CCD camera. Data collection was carried out at a nominal magnification of 30,000×, yielding a pixel size of 3.45 Å at the specimen level, and a defocus range of −0.6 to −1.4 μm.

Further electron microscopy investigation of MCM loading with and without ORC6 was performed as follows. Reactions were set up by mixing 45 nM of a M.HpaII-Gid70-M.HpaII DNA template with 90 nM ORC1–5(1ΔN), 150 nM CDC6(ΔN), 150 nM CDT1(ΔN), 60 nM MCM2–7 in EM buffer containing either 2 mM ATP or 2 mM ATPγS, with or without 150 nM ORC6. Samples were incubated at 37 °C for 30 min under agitation, diluted 1:4 in EM buffer and immediately used for negative staining as described above. Grids were imaged on a FEI Tecnai G2 Spirit microscope using a RIO16 camera at a pixel size of 3.1 Å per pixel. Particles were picked using crYOLO[46] and extracted with a box size of 144 pixels in Relion 4[47]. After 2 rounds of 2D classification, well-averaging particles were counted, class populations were visualized in a 10 × 10 dot plot in Prism10.

## Negative-stain image processing

Negative-stain images were processed using Relion 3.1[47]. Particles were picked using Topaz v0.2.5[48]. Contrast transfer function (CTF) parameters were estimated using Gctf v1.06[49]. Extracted particles were then subjected to reference-free 2D classification.

## Cryo-EM reactions and imaging with ORC6

UltrAuFoil R1.2/1.3 300-mesh grids (Quantifoil) were glow-discharged at 40 mA for 5 min using a GloQube Plus Glow Discharge System (Quorum), before applying 3 μl graphene oxide dispersion (10 ml graphene oxide flake dispersion (Sigma) diluted in 80 ml water; aggregates removed by centrifugation at 500g for 1 min). Incubation was carried out for three minutes, followed by blotting of excess liquid and three successive washes with 20 μl droplets of water. After 1–2 h drying at room temperature, 4 μl of the undiluted (ATP) reaction or the 3:1 diluted (ATPγS) reaction were applied to grids for 60 s at room temperature and 90% humidity in a Vitrobot Mark IV (Thermo Fisher). Grids were double-side plotted with force 0 for 5 s and immediately plunge frozen in liquid ethane. Micrographs were collected in counting mode using a pixel size of 1.08 Å on a Titan Krios transmission electron microscope with a K2 Summit direct electron detector and BioQuantum energy filter. A total electron dose of 49.28 e⁻ Å⁻² was used over 32 dose-fractioned movie frames and a total exposure time of 9.4 s. The defocus ranged from −1.0 to −2.5 μm. 3,589 movies were collected for the ATP reaction and a total of 31,569 movies for the ATPγS reaction.

## Cryo-EM reactions and imaging without ORC6

Gid70 DNA template (45 nM), carrying a TwinStrep-tagged M.HpaII roadblock at each end, were mixed with 90 nM ORC1–5(1ΔN), 150 nM CDC6(ΔN), 150 nM CDT1(ΔN) and 60 nM MCM2–7 in EM buffer to a total volume of 40 μl, and incubated for 30 min under agitation at 37 °C. Four microlitres of 50% diluted loading reaction were applied onto UltrAuFoil R1.2/1.3 300-mesh grids, which were coated with a graphene oxide layer as described above. After one minute of on-grid incubation in a Vitrobot Mark IV, the grids were blotted from both sides

with Whatman Filter paper (blotting strength 0, blot time 3.5 s) and plunged into liquified ethane. Grids were clipped and stored in liquid nitrogen prior to data collection.

### hDH loading reactions without CDC6 imaged

A hDH loading reaction was prepared by co-incubating 45 nM of a M.HpaII-Gid70-M.HpaII DNA with 90 nM ORC1–5(1ΔN), 150 nM CDT1(ΔN) and 60 nM MCM2–7 in assay buffer containing 60 mM sodium acetate (supplemented with 2 mM ATP) for 30 min at 37 °C under agitation. The sample was diluted 1:4 and subsequently negative-stained as described above. One hundred and fourteen micrographs were acquired using a RIO16 detector on a FEI TECNAI G2 Spirit Microscope at 3.1 Å per pixel. 69,785 particles were extracted with a 144-pixel box after crYOLO[46] picking and submitted to multiple rounds of 2D classification to identify MCM hDH particles in Relion 4[47].

### Cryo-image processing of hDH loading

Image processing was performed using Relion 4.0b-GPU and cryoSPARC v3.3.2[50] at different stages of the processing pipeline as indicated in Extended Data Fig. 2. Beam-induced motion was accounted for by the Relion implementation with 5 × 5 patches and CTF parameters were estimated using CTFFIND v4.1.10[51]. Particle picking was carried out using Topaz v0.2.4[48], followed by particle extraction with a 440-pixel box and rescaling to 110 pixels in cryoSPARC. Three thousand, five hundred and eighty-nine micrographs with 970,326 particles were selected based on the CTF fit resolution of 2.57–4.50 Å, CTF fit cross-correlation of 0.07–0.27 and median pick score of 20.18–43.55. Three rounds of reference-free 2D classification, ab initio reconstruction and heterogeneous refinement were conducted to identify 49,485 hDH particles. After particle extraction without downscaling and using a 400-pixel box, the particle stack was further cleaned up by 2D classification, which yielded 19,049 particles used to compute high-resolution hDH 2D class averages. Ab initio reconstruction, followed by homogeneous, non-uniform and local refinement with $C_2$ symmetry resulted in a map with 3.1 Å resolution. The particle stack was re-extracted, re-grouped, and cleaned from duplicate particles using Relion. Fifteen thousand, eight hundred and seventy-four particles of the highest quality were isolated by 3D classification without alignment using a 320 Å mask and a regularization parameter $T$ of 4. Particles were polished[47] subjected to 2D classification without alignment and 3D refinement imposing $C_2$ symmetry. CTF parameters were optimized[47] in three rounds (first, per-particle defocus, per-micrograph astigmatism; second, per-particle defocus, per-particle astigmatism, beamtilt; third, per-particle defocus, per-particle astigmatism, beamtilt, trefoil, 4th order aberrations) to yield a 3.3 Å resolution hDH structure. Homogeneous and non-uniform refinement with $C_2$ symmetry in cryoSPARC resulted in the final map at 3.1 Å resolution.

To determine the structure of hSH from the same dataset, a new Topaz model was trained. One hundred and seventy-eight thousand, eight hundred and fifty three particles were extracted with a 440-pixel box, downscaled to 110 pixels. Smaller particles and contaminations were removed by 2D classification in cryoSPARC. Initial volumes were obtained by ab initio reconstruction with six classes. Thirty-seven thousand, three hundred and ninety-six hSH particles were then isolated in two rounds of heterogeneous refinement (Extended Data Fig. 7a) and re-extracted with a 400-pixel box. Homogeneous, non-uniform and local refinement with a mask encompassing the entire hSH yielded a 3.4 Å resolution structure. The same particles were re-extracted and 3D refined in Relion. CTF refinement (per-particle defocus, per-micrograph astigmatism), Bayesian polishing and one additional round of CTF refinement (per-particle defocus, per-particle astigmatism, beamtilt) were carried out. Twenty-five thousand and sixty-nine high-resolution hSH particles were isolated by 3D classification. Homogeneous, non-uniform and local refinement with a mask around the hSH yielded 3.2 Å resolution in cryoSPARC.

### Cryo-image processing of MCM recruitment

Movies were corrected for beam-induced motion using the Relion implementation with 5 × 5 patches in Relion 4.0b-GPU[52] and CTF parameters were estimated using CTFFIND v4.1.13[51]. A Topaz model[48] was trained and 1,334,277 particles were picked from 31,569 micrographs. Particles were extracted with a 416-pixel box, rescaled to 104 pixels. Six hundred and twenty-nine thousand, two hundred and forty-one MCM-containing particles (that is, hOCCM, hSH and hMO*) were isolated using reference-free 2D classification. Initial volumes were generated using ab initio reconstruction in cryoSPARC. Low-pass filtered volumes were used for multi-reference 3D classification in Relion (Extended Data Fig. 5). Particles contributing to the different complexes (114,995 hSH, 170,792 hOCCM and 203,088 hMO* particles) were re-extracted using a 400-pixel box without rescaling. Homogeneous, non-uniform and local refinements with masks around the entire respective complex resulted in 3.6 Å resolution structure of the hSH, 4.0 Å hOCCM and 3.7 Å hMO* in cryoSPARC. Bayesian polishing was carried out for each particle stack individually in Relion[47].

The hOCCM was then subjected to three rounds of CTF refinement (first, per-particle defocus, per-micrograph astigmatism; second, anisotropic magnification; third, per-particle defocus, per-particle astigmatism, beamtilt) followed by another round of Bayesian polishing in Relion. A mask around CDC6 and portions of ORC1–5 was generated to carry out focused 3D classification without alignment. This yielded 100,567 particles with good density for ORC–CDC6. 3D refinement in Relion was followed by homogeneous refinement in cryoSPARC. Local refinement with a mask around ORC–CDC6 was used to improve alignment on this part of the complex. Thirty-four thousand, one hundred and sixteen hOCCM particles with well-resolved CDC6 density and 49,771 hOC₁M particles that lacked CDC6 were isolated by 3D classification without alignment in cryoSPARC. Both hOCCM and hOC$_1$M structures were locally refined to 3.8 and 4.1 Å, respectively, using a mask encompassing the entire complex.

Complexes containing single-loaded hexamers, hSH and hMO* particles were initially processed together (Extended Data Fig. 5). Three-dimensional refinement in Relion using a mask around the MCM yielded a 3.8 Å hSH structure. Two rounds of CTF refinement (first, per-particle defocus, per-micrograph astigmatism; second, anisotropic magnification) and another round of Bayesian polishing was carried out. One hundred and thirty-five thousand, seven hundred and forty-two hSH particles and 182,341 hMO* particles were subsequently separated by multi-reference 3D classification using the initial maps of hSH and hMO*, low-pass filtered to 30 Å. The hSH was refined in cryoSPARC (homogeneous, non-uniform, followed by local refinement) to 3.4 Å. Homogeneous and non-uniform refinement of the hMO* resulted in a consensus map solved to 3.6 Å resolution. The MCM portion of the complex was locally refined to 3.5 Å, while the ORC was refined to 4.0 Å.

**Cryo-EM of hDH loading without ORC6.** A total of 5,158 movies were collected in counting mode on a 200 kV Talos transmission electron microscope using a pixel size of 1.61 Å per pixel with a total dose of 48 e⁻ Å⁻² and a defocus range of −2 to −3.5 μM (step size 0.25 μM). Movies were corrected for beam-induced motion with the Relion implementation with 5 × 5 patches in Relion 5.0 and CTF parameters estimated with CTFFIND v4.1.13[51]. Subsequent micrograph curation reduced the number of micrographs to 2,168. Four hundred and fifty-six thousand, six hundred and eighty-six particles were picked using template matching in cryoSPARC v4.4.1[50] and extracted at a pixel size of 3.22 Å per pixel (2× binned) and subjected to two rounds of 2D classification. A total of 77,002 particles were used to generate an ab initio reconstruction in $C_1$. Particles were re-extracted at full resolution, refined and then symmetry expanded in $C_2$. Three-dimensional classification was performed using 4 classes. The class with the highest quality DNA density

containing 74,880 particles was subject to a local refinement resulting in a 4.1 Å structure according to gold-standard FSC at 0.143 criterion.

## MCM5(AG) DH loading for cryo-EM

A 50-µl MCM DH loading reaction was assembled using the protocol described in 'Human MCM loading for cryo-EM'. In short, 45 nM M.HpaII–Gid70–M.HpaII DNA capped at each end with TwinStrep-tagged M.HpaII, was co-incubated with 90 nM ORC1–5(1ΔN), 150 nM CDC6(ΔN), 150 nM CDT1(ΔN) and 60 nM MCM5(AG)–MCM2–7 in EM buffer at 37 °C under agitation. After 30 min, the loading reaction was diluted either by 40% or 75% with EM buffer and immediately used for cryo-EM.

UltrAuFoil R1.2/1.3 300-mesh grids were coated with a graphene oxide support as described above. Four microlitres of diluted loading reaction were applied onto each grid in a Vitrobot Mark IV set to 22 °C and 90% humidity. After one minute on-grid incubation, grids were double-side blotted for 4.5 s with blot force 0 and plunged into liquid ethane. Grids were subsequently clipped and stored in liquid nitrogen until data collection.

## Cryo-EM of MCM5(AG) DH

In total, 13,203 movies were acquired from two grids at 92,000× magnification (1.58 Å per pixel) on a Glacios microscope equipped with a Falcon 3 direct electron detector operated in linear mode. A total dose of 50 e$^-$ Å$^{-2}$ (exposure time 1.12 s) and a defocus range of −1 to −2.5 µm (0.3 µm step size) were applied. Movie frames were aligned using Relion 4.0 (ref. 52) and CTF was estimated using GCTF v1.06 (ref. 49). Two million, eight hundred and seventy-five thousand, eight hundred and twenty-five particles were picked using a pre-trained Topaz network[48], extracted with a box size of 64 pixels (4× binned to 6.32 Å per pixel) and imported in cryoSPARC v4.0[50] for 2 rounds of 2D classification. After 2D cleaning, two ab initio models were generated with $C_1$ symmetry using a subset of 50,000 particles. All 379,664 particles were subjected to one round of heterogenous refinement against the two ab initio reconstructions, followed by homogenous refinement in $C_1$. Particles were re-extracted in Relion with a box size of 300 pixels (unbinned at 1.58 Å per pixel) and 2D-classified in cryoSPARC. The resulting 322,548 particles were used to generate an unbinned ab initio reconstruction in $C_1$, yielding a 5.76 Å map after non-uniform refinement in $C_1$. Three-dimensional classification in Relion without alignment using a regularization parameter $T$ of 200 distributed the particles into four classes with roughly equal populations, differing by presence of both, either or no MCM6 WH domain. The best-resolved class (showing density for both MCM6 WH domains) was non-uniform and locally refined in cryoSPARC with $C_2$ symmetry, yielding a final resolution of 5.6 Å according to gold-standard FSC and the 0.143 criterion. Local resolution estimation was carried out in cryoSPARC. Refinement statistics are reported in Extended Data Table 1.

## hDH model building

AlphaFold-Multimer[53] was used to generate models of the ATPase tier (including the WH domains) of the hexameric human MCM2–7 assembly as well as the amino-terminal tier. Each model was rigid-body docked into one hexamer of the 3.1-Å resolution hDH map using UCSF ChimeraX v1.6.1 (ref. 54). Each chain was refined in Coot v0.9.8.1 EL[55] and sections that could not be confidently built were deleted[54]. The models of the ATPase and amino-terminal tiers were then combined. ATP, ADP, magnesium and zinc ions were added in the pertinent sites. Idealized B-form DNA was first docked into the density of the double helix and then manually modified to account for the stretch of underwound and melted DNA within the N-terminal dimerization interface. The model consisting of the MCM2–7 hexamer, ligands and DNA was adjusted using ISOLDE 1.6.0 (ref. 56), imposing ligand, secondary and base pairing restraints. This was followed by an iterative process of real space refinement with restraints on geometry, secondary structure,

metal coordination and nucleic acid planarity in Phenix v1.21 (ref. 57) and manual adjustments in Coot and ISOLDE. To generate the hDH, a copy of the refined MCM hexamer was generated and rigid-body docked into the second hexamer of the hDH map. Clashes at the interface of the two hexamers were addressed using ISOLDE. The resulting model was subjected to real space refinement in Phenix. Refinement statistics are reported in Extended Data Table 1.

## hSH model building

The atomic model of the hDH was rigid-body docked into the 3.2 Å cryo-EM map of the hSH using UCSF ChimeraX v1.6.1 (ref. 58). The second hexamer was deleted, nucleotides were inspected and the DNA was replaced by an idealized B-form duplex DNA using Coot v0.9.8.1 EL[59]. The model was refined with ISOLDE 1.6.0 (ref. 56) and Phenix real space refinement with restraints on geometry, secondary structure, metal coordination and nucleic acid planarity[57]. Refinement statistics are reported in Extended Data Table 1.

## hMO* model building

To generate the model of the hMO* complex, the atomic structure of the hSH described above was docked into the map of the globally refined hMO* map using UCSF ChimeraX v1.6.1 (ref. 54). The atomic model of *H. sapiens* ORC6 was retrieved from the AlphaFold Protein Structure Database (accession code AF-Q9Y5N6-F1). To guide the positioning of ORC6, the yMO (PDB entry 6RQC)[28] was aligned with the human MCM in the map. Based on the alignment with the *S. cerevisiae* Orc6, the N-terminal cyclin box domain (residues 1–94) of human ORC6 was positioned next to the N-terminal domains of MCM2 and MCM6. The domain was then rigid-body docked into the density. The second cyclin box domain (residues 95–190) was docked into the adjacent density of the map that is positioned between MCM6 and ORC5. Docking solutions with the highest cross-correlation scores were chosen. The starting model of ORC1–5 was the open conformation of the human ORC1–5 complex (PDB code 7JPR)[60]. The model was docked into the locally refined map of the ORC at 4.0 Å resolution. The DNA model of the yMO complex (PDB entry 6RQC)[28] was fit into the DNA density in the globally refined hMO* map. The DNA model was refined in Coot v0.9.8.1 EL[59] applying all-molecule self-restraints 6.0 and DNA B-form restraints, before combining it with the DNA in the hSH. ORC6 was combined with the MCM and DNA into one PDB model. The two models of ORC6–MCM and ORC1–5 were iteratively refined with ISOLDE 1.6.0 (ref. 56) and Phenix real space refinement with restraints on geometry, secondary structure, metal coordination and nucleic acid planarity[57] against the globally refined map of hMO* and the locally refined map of the ORC, respectively. For illustration purposes, a composite map was generated using Phenix and the refined ORC6–MCM and ORC1–5 models were combined into one PDB model. Refinement statistics are reported in Extended Data Table 2.

## hOCCM model building

To assemble the atomic model of hOCCM, the human ORC1–5–DNA model (PDB entry 7JPS), the ORC2 WH domain of the ORC1–5 model (PDB entry 7JPR)[60] and the prediction of human CDC6 retrieved from AlphaFold Protein Structure Database (accession code AF-Q99741-F1) were rigid-body docked into the 3.8 Å cryo-EM map using UCSF ChimeraX v1.6.1 (ref. 54). Unstructured parts of the models, which were not visible in the hOCCM map, were deleted. The hSH model was docked into the map and each subunit was split and fitted independently into the density. The N-terminal tiers of MCM7, MCM3 and MCM5 were deleted from the atomic coordinates file as the local quality of region of the map was deemed insufficient for model building. The positioning of the ATPase tiers of the same MCM subunits was guided by the position in the hSH and no further adjustment of atomic positions was carried out due to the limited local resolution. The atomic models of the MCM3 and MCM7 WH domains were overlayed with equivalent

domains in yOCCM. The atomic model of CDT1 was retrieved from the AlphaFold database (accession code AF-Q9H211-F1) and the three structured domains (residues 167–387, 418–440 and 441–546) were extracted from the model. Each domain was overlayed with yeast Cdt1 in yOCCM (PDB entry 5V8F)[29]. All models were combined into one PDB model and adjusted using Coot v0.9.8.1 EL[59] and ISOLDE 1.6.0 (ref. 56). Models of ATPγS and magnesium were added in the ATPase active sites of MCM2–6, MCM6–4, ORC1–4, ORC4–5 and ORC5–3. A 39-mer idealized B-form duplex DNA was generated and fit into the density by applying all-molecule self-restraints 6.0 and DNA B-form restraints in Coot. The DNA coordinates were then merged with the hOCCM model before carrying out Phenix real space refinement with restraints on geometry, secondary structure, metal coordination and nucleic acid planarity[57]. Refinement statistics are reported in Extended Data Table 3.

## Analysis of protein–DNA contacts

Protein–DNA contacts were analysed using the DNAproDB web-based visualization tool[61,62].

## Statistics and reproducibility

Proteins were independently purified at least twice (Fig. 1a). The experiments in Figs. 1d,f,g,j, 2e,g and 3a,f, Extended Data Fig. 1a,n and Supplementary Fig. 1a–d were performed three times. The experiments in Figs. 1c,e,h,i and 3g,i,k, Extended Data Figs. 1b–i,m, 3d and 4a and Supplementary Figs. 1e–h and 2b were performed twice. The micrographs shown in Extended Data Figs. 2a and 4b are representative micrographs of cryo-EM datasets. Similar experiments have been analysed multiple times by negative-stain electron microscopy.

## Reporting summary

Further information on research design is available in the Nature Portfolio Reporting Summary linked to this article.

# Data availability

Data supporting the findings of this study are available within the paper and its Supplementary Information files. Cryo-EM density maps have been deposited in the Electron Microscopy Data Bank (EMDB) under the accession codes EMD-19618 (hDH), EMD-19625 (MCM5(AG) hDH), EMD-19619 (hSH), EMD-19620 (hMO*), EMD-19621 (the locally refined ORC of hMO*), EMD-19622 (composite map of the globally refined hMO* and the locally refined ORC map), EMD-19623 (hOCCM) and EMD-19624 (hOC₁M). Atomic coordinates have been deposited in the Protein Data Bank (PDB) with the accession codes 8S09 (hDH), 8S0A (hSH), 8S0B (MCM–ORC6 of hMO*, which was modelled using the globally refined hMO* map), 8S0C (ORC1–5 of hMO*, which was modelled using the locally refined map of the ORC), 8S0D (composite model of MCM–ORC6 and ORC1–5 docked into the composite map), 8S0E (hOCCM) and 8S0F (hOC₁M).

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

**Acknowledgements** The authors thank A. Alidoust, N. Patel and D. Patel for yeast protein expression; A. Nans, D. Benton, A. Purkiss and P. Walker for support with cryo-EM and computing; M. Gross for the gift of yeast proteins; and G. Lee for purified plasmid. This work was supported by the Francis Crick Institute, which receives its core funding from Cancer Research UK (CC2002 and CC2009), the UK Medical Research Council (CC2002 and CC2009), and the Wellcome Trust (CC2002 and CC2009). This work was also funded by Wellcome Trust Senior Investigator Awards (219527/Z/19/Z) to J.F.X.D. and European Research Council Advanced Grants (101020432-MeChroRep) to J.F.X.D. A.C. receives funding from the European Research Council (ERC) under the European Union's Horizon 2020 research and innovation programme (grant agreement 820102). F.W. has received funding from the European Union's Horizon 2020 research and innovation programme under the Marie Sklodowska-Curie grant agreement 844211. T.P. is the recipient of a Boehringer Ingelheim PhD fellowship. Work in the laboratory of T.C.R.M. is supported by a Novo Nordisk Fonden Hallas-Møller Emerging Investigator Grant (NNF22OC0073571), the Danish National Research Foundation (DNRF115) and the Carlsberg Foundation (CF21-0571). For the purpose of Open Access, the author has applied a CC BY public copyright licence to any Author Accepted Manuscript version arising from this submission.

**Author contributions** F.W., J.F.G., A.C. and J.F.X.D. conceived the study. F.W. cloned, expressed and purified human proteins and developed all biochemical assays. E.L.E. found Cdc6-independent MCM loading. J.F.G., F.W., T.P., E.L.E. and T.C.R.M. performed negative-stain work to develop and optimize the electron microscopy-based MCM loading assay. J.F.G. prepared cryo-EM grids, performed all cryo-EM data collection and image processing except for cryo-EM work on MCM5(AG), which was performed by T.P., and the ORC6-dropout hDH, which was performed by E.C.C. J.F.G. performed all atomic model building and refinement. J.F.X.D. supervised the biochemical work and A.C. supervised the structural work. F.W., J.F.G., A.C. and J.F.X.D. wrote the manuscript with input from the other authors.

**Funding** Open Access funding provided by The Francis Crick Institute.

**Competing interests** The authors declare no competing interests.

**Additional information**
**Correspondence and requests for materials** should be addressed to John F. X. Diffley or Alessandro Costa.

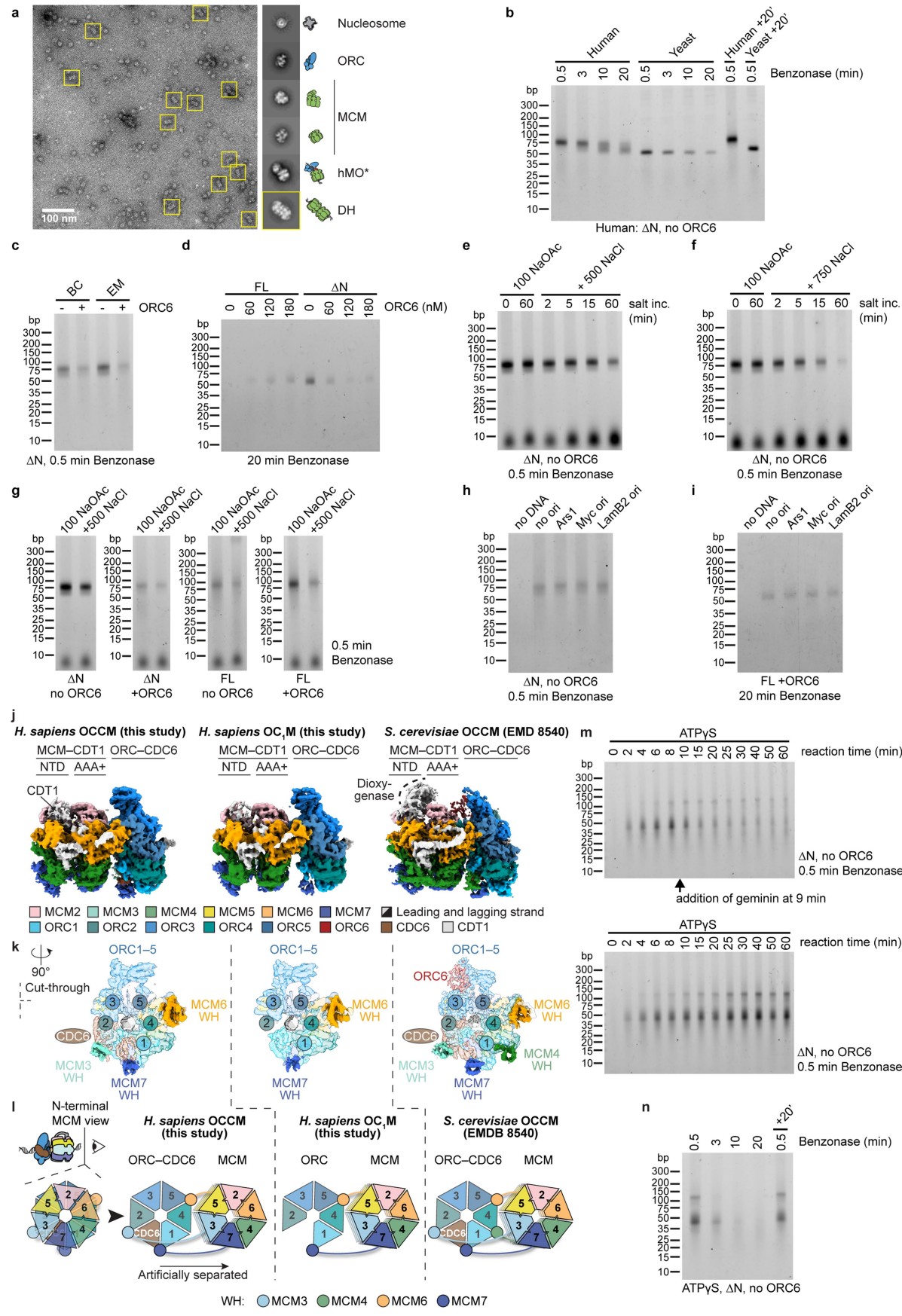

**Extended Data Fig. 1** | See next page for caption.

**Extended Data Fig. 1 | a**, Representative negative-stain EM micrograph and 2D averages of a human MCM loading reaction (ΔN proteins) on a short synthetic yeast origin flanked by nucleosomes. **b**, Benzonase footprinting assay with human (ΔN, no ORC6) and yeast MCM loading reactions that were performed at 30 °C for 30 min, and then incubated with benzonase as indicated for 0.5–20 min. Control reactions were incubated for an additional 20 min before addition of benzonase for 0.5 min. **c**, Comparison of MCM loading reactions (ΔN) using protein concentrations used in biochemistry experiments (BC: 90 nM ORC1-5(1ΔN), 150 nM ORC6) and in electron microscopy experiments (EM: 120 nM ORC1-5(1ΔN), 120 nM ORC6). Other components at standard concentrations (60 nM MCM2-7, 150 nM CDC6ΔN, 150 nM CDT1ΔN, 4 nM pJY22 plasmid DNA). **d**, MCM loading was performed using 3xFL or 3xΔN conditions and ORC6 was titrated. ORC1-5 FL and ORC1-5(1ΔN) were used at 120 nM. **e**, Salt stability of the hDH footprint in a buffer containing 500 mM NaCl. MCM was loaded for 30 min and incubated with salt-containing buffers for the indicated time. Reactions were then diluted and treated with benzonase. **f**, As **e**, with a buffer containing 750 mM NaCl. **g**, Salt stability of double hexamers generated using different reaction conditions. MCM was loaded as indicated for 30 min, and then incubated in a buffer containing 500 mM NaCl for 15 min. Reactions were diluted to low salt and treated with benzonase for 0.5 min. The different reaction conditions were tested in the same experiment, run on separate gels and the gels stained individually. **h**, MCM loading (ΔN, no ORC6) was performed on 2,398 bp doubly biotinylated PCR products in the presence of streptavidin. No ori: fragment of pET21a backbone, Ars1: yeast origin from pJY22, Myc ori: human origin at the Myc locus, LamB2 ori: human origin at the Lamin B2 locus. **i**, As **h**, using FL + ORC6 reaction conditions. **j**, Cryo-EM maps of the hOCCM, hOC$_1$M and yOCCM. Cdt1 is colored in white to highlight the similar binding mode to MCM in the human and yeast complexes. **k**, Cut-through view of the hOCCM, hOC$_1$M and yOCCM illustrating the interactions of the MCM WH domains with ORC–CDC6. MCM6 and MCM7 WH domains interact with ORC in all structures, while MCM3 is only visible when CDC6 is present. The MCM4 WH is visible only in yOCCM. **l**, Cartoon view of the WH interactions described in **k**. **m**, Nuclease footprinting assay showing reaction timecourses (ΔN, no ORC6) using ATPγS with (top) and without (bottom) addition of geminin at minute 9 after starting the reaction. **n**, Timecourse of benzonase treatment (0.5–20 min) after a 30-minute reaction using ATPγS (ΔN, no ORC6). A control was incubated for an additional 20 min followed by 0.5 min benzonase treatment.

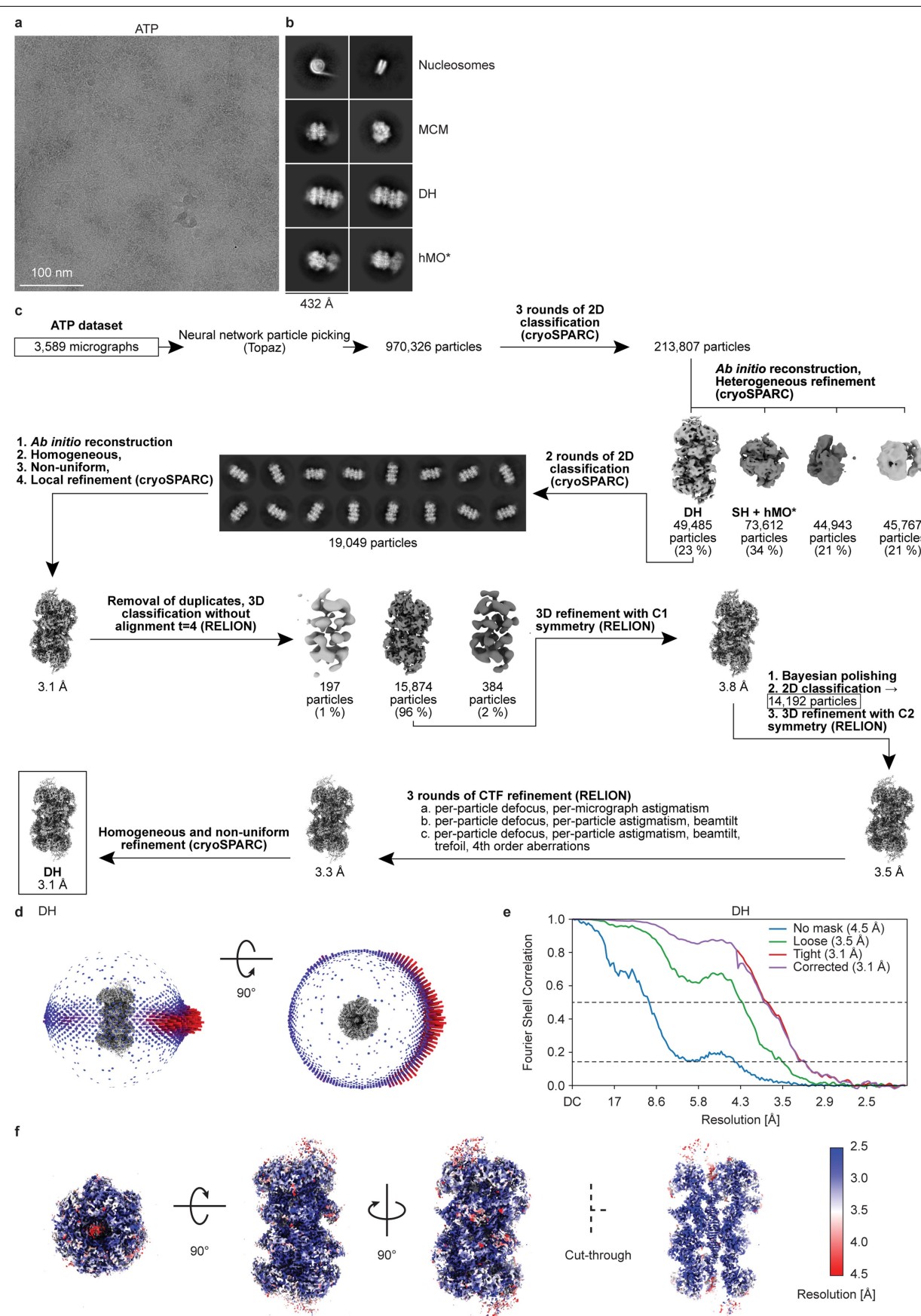

**Extended Data Fig. 2 | a, Representative cryo-electron micrograph of the human MCM loading reaction. b**, 2D averages of nucleosomes, MCM, hMO* and hDH were observed in the dataset. Box size 432 Å. **c**, Processing pipeline for the hDH. **d**, Angular distribution of the hDH structure. **e**, Fourier shell correlation plot for the hDH structure. **f**, Three rotated views and one cut-through view of the hDH structure, color-coded according to the local resolution as estimated in cryoSPARC.

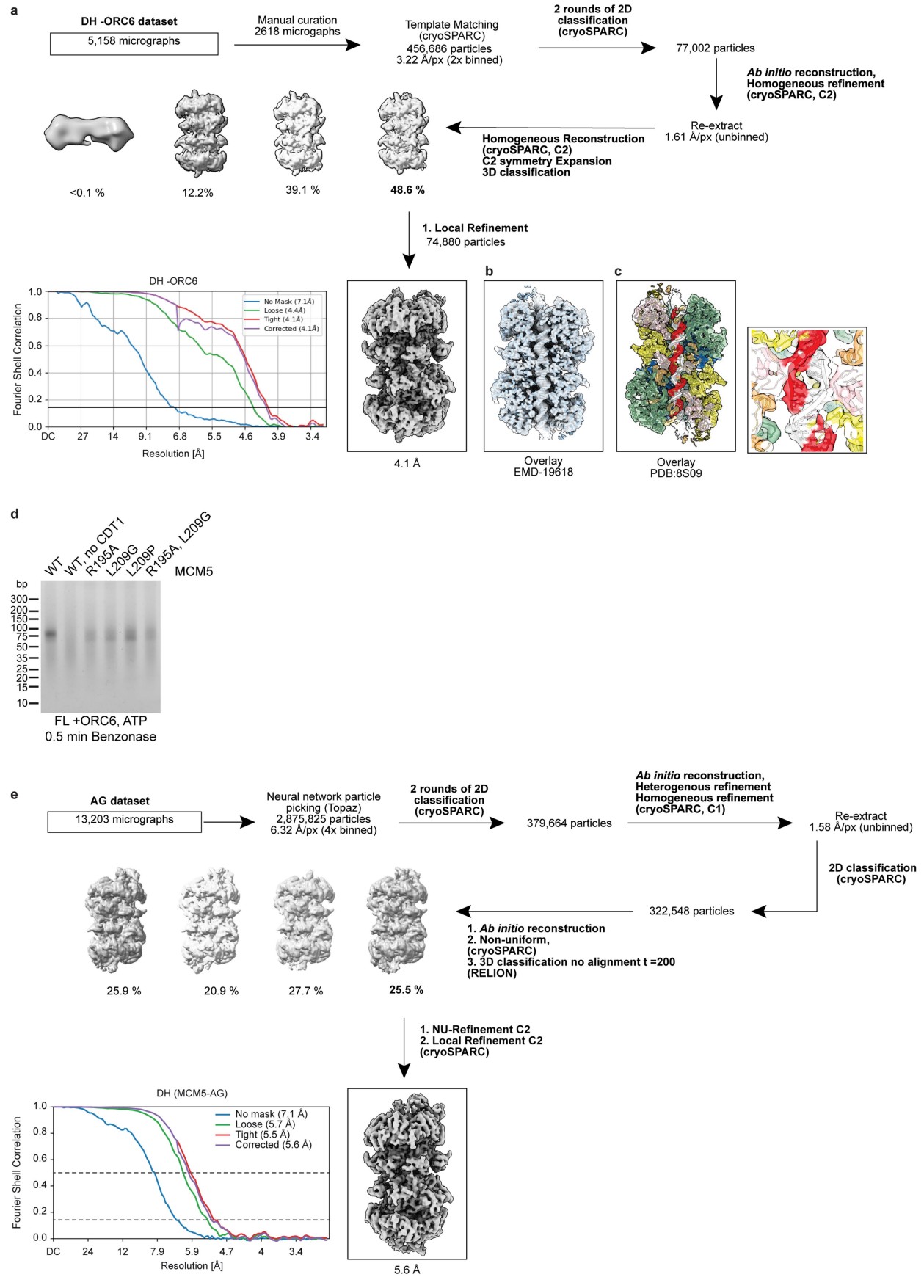

**Extended Data Fig. 3** | See next page for caption.

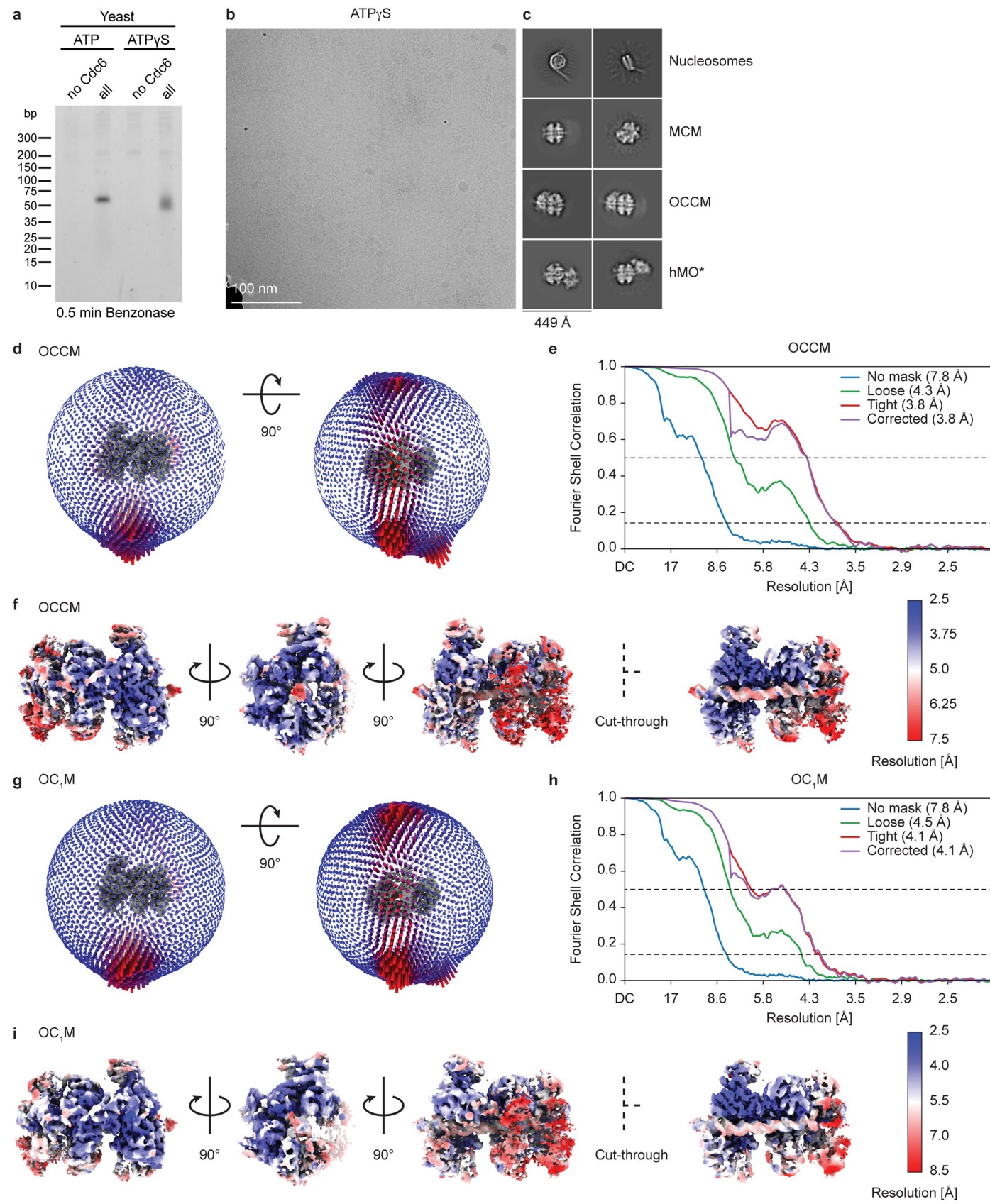

**Extended Data Fig. 4** | See next page for caption.

**Extended Data Fig. 4** | **a**, Nuclease footprinting assay with yeast proteins using ATP or ATPγS. **b**, Representative micrograph of the human MCM recruitment reaction using ATPγS. **c**, 2D averages of nucleosomes, MCM, hOCCM and hMO* were observed in the dataset. The box size 449 Å. **d**, Angular distribution of the hOCCM structure. **e**, Fourier shell correlation plot for the hOCCM structure. **f**, Three rotated views and one cut-through view of the hOCCM structure, color-coded according to the local resolution as estimated in cryoSPARC. **g**, Angular distribution of the hOC$_1$M structure. **h**, Fourier shell correlation plot for the hOC$_1$M structure. **i**, Three rotated views and one cut-through view of the hOC$_1$M structure, color-coded according to the local resolution as estimated in cryoSPARC.

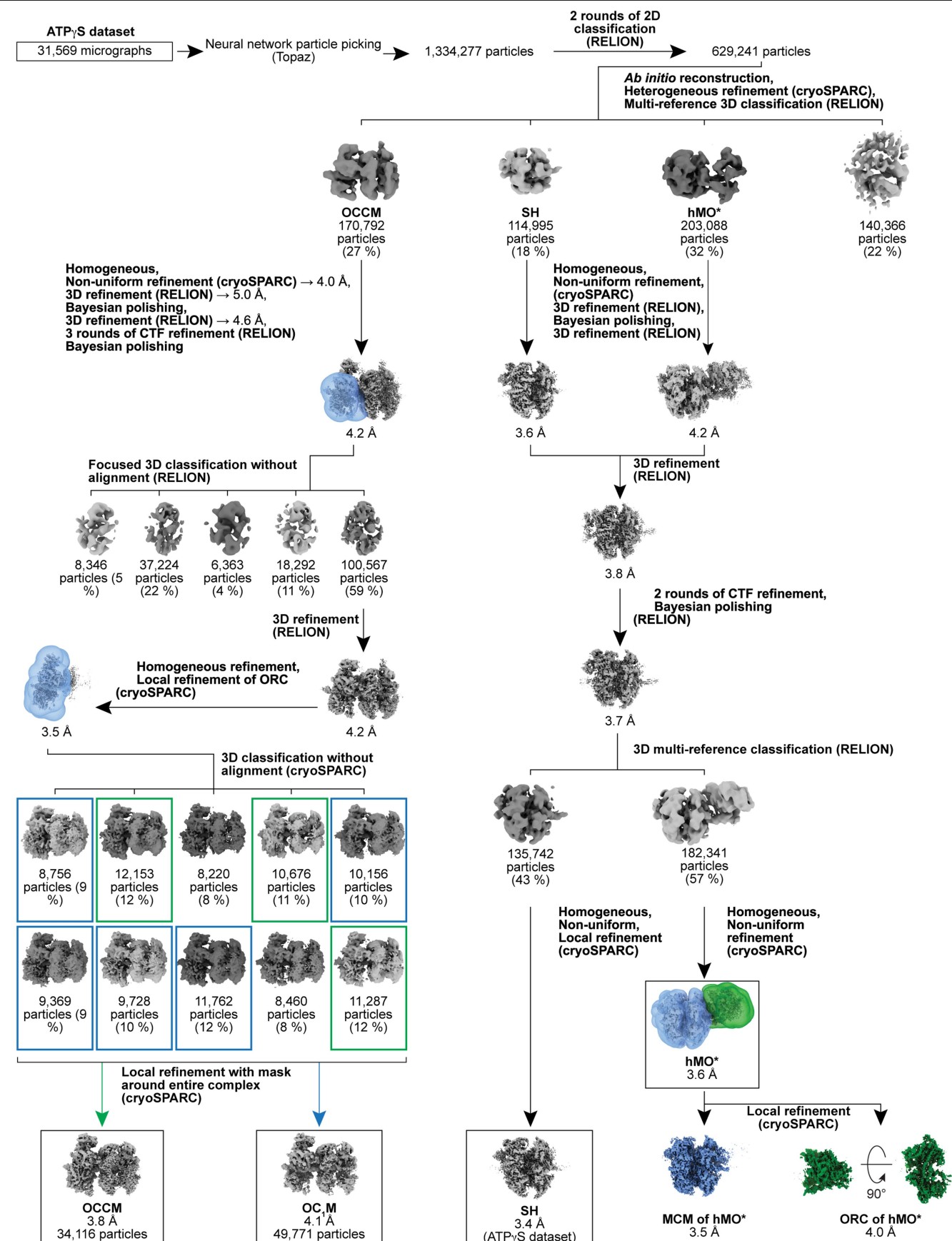

**Extended Data Fig. 5 | Processing pipeline for the structures obtained in the ATPγS reaction.**

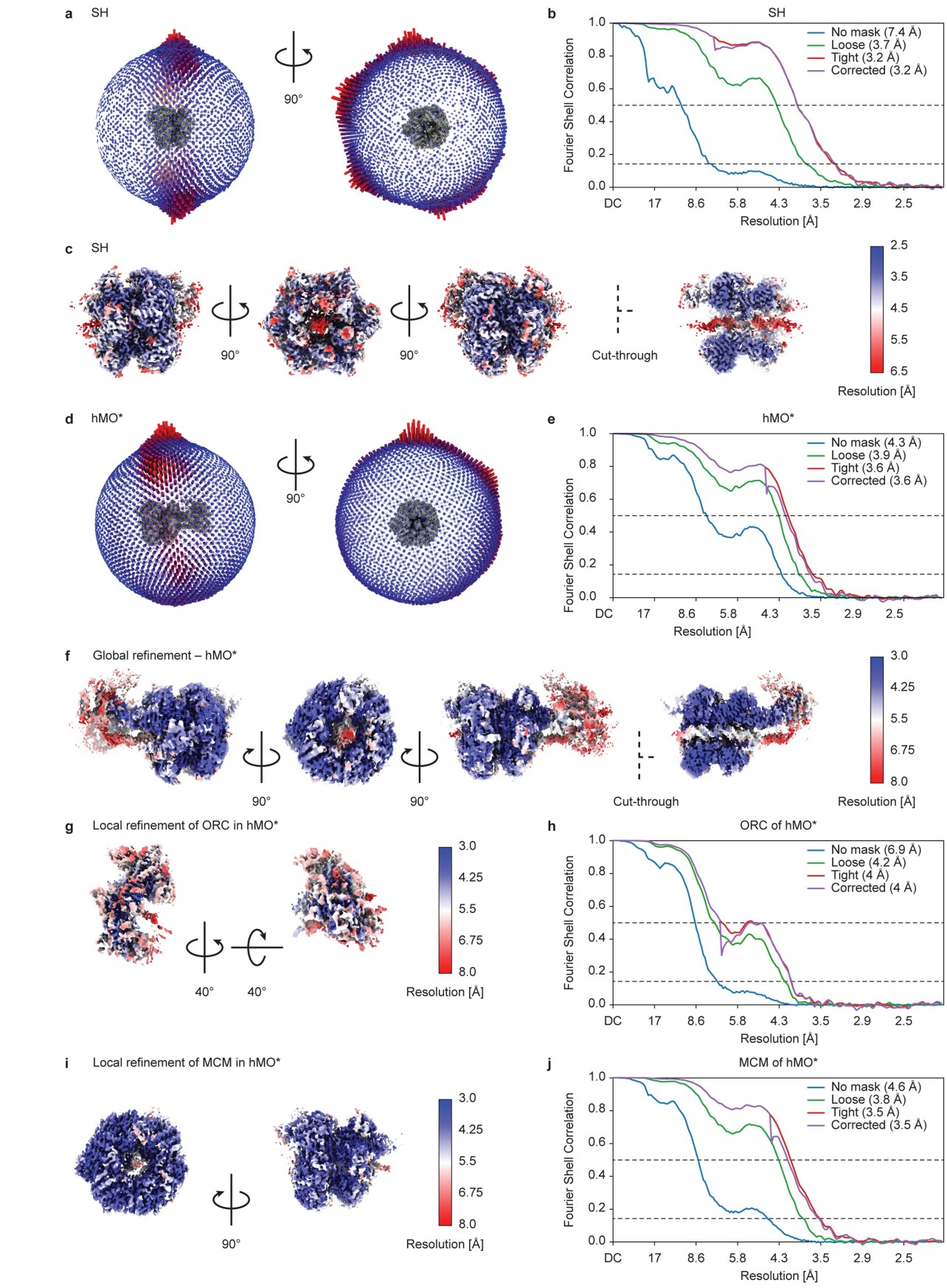

**Extended Data Fig. 6 |** See next page for caption.

**Extended Data Fig. 6 | a**, Angular distribution of the hSH structure that was obtained in the presence of ATP. **b**, Fourier shell correlation plot for the hSH structure. **c**, Three rotated views and one cut-through view of the hSH structure, color-coded according to the local resolution as estimated in cryoSPARC. **d**, Angular distribution of the globally refined hMO* structure. **e**, Fourier shell correlation plot for the globally refined hMO* structure. **f**, Three rotated views and one cut-through view of the globally refined hMO* structure, color-coded according to the local resolution as estimated in cryoSPARC. **g**, Locally refined ORC structure in the hMO*, color-coded according to the local resolution as estimated in cryoSPARC. **h**, Fourier shell correlation plot for the locally refined ORC structure. **i**, Locally refined MCM structure in the hMO*, color-coded according to the local resolution as estimated in cryoSPARC. **j**, Fourier shell correlation plot for the locally refined MCM structure.

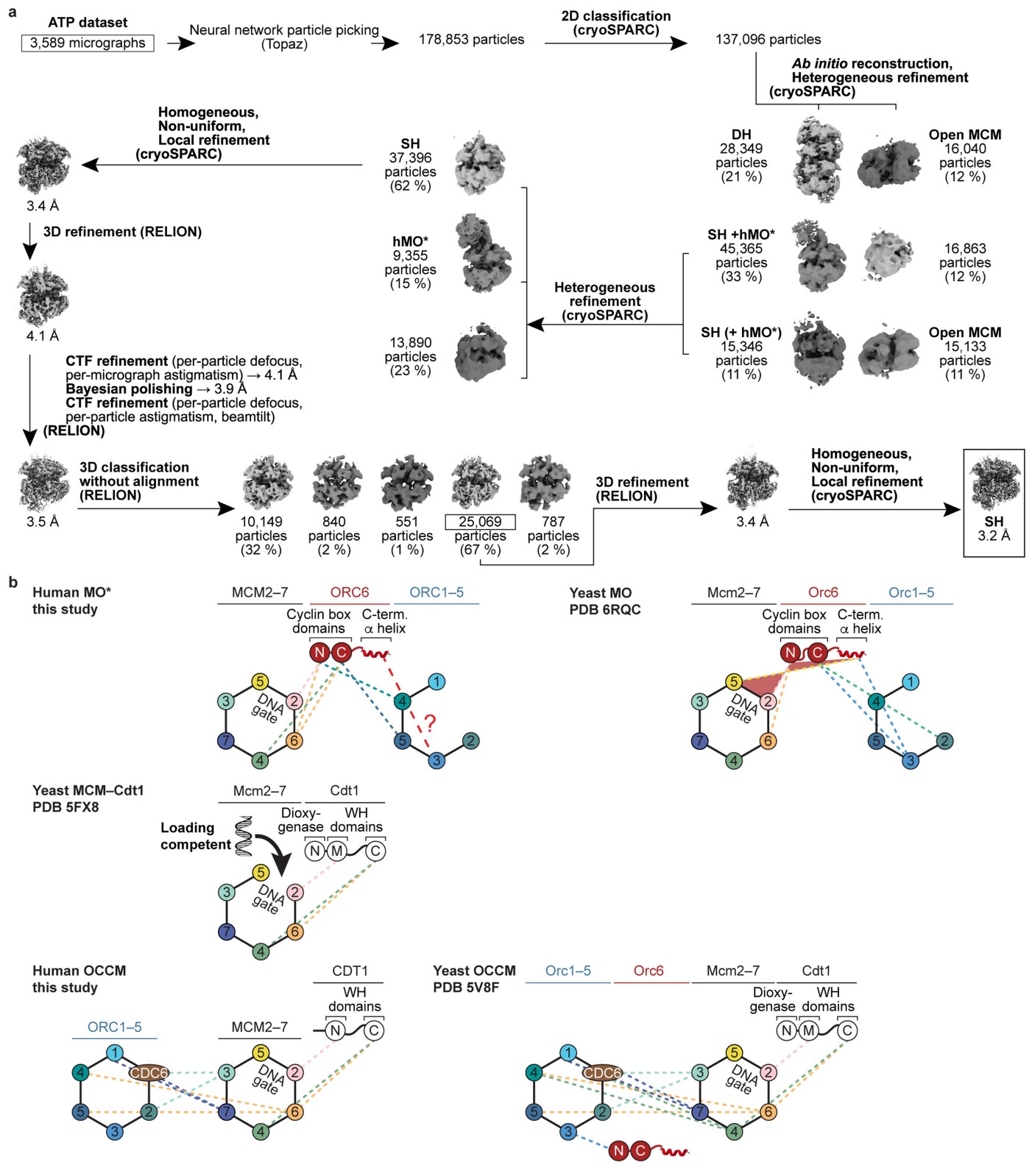

**Extended Data Fig. 7 | a**, Processing pipeline for the hSH structure that was obtained from the human MCM loading reaction in the presence of ATP. **b**, Schematic representation of protein–protein interactions in hMO*, yMO, yMCM–Cdt1, hOCCM and yOCCM. In hMO*, the N-terminal cyclin box domain interacts with MCM2, MCM6 and ORC4, while the C-terminal cyclin box binds MCM6, MCM4 and ORC5. The resolution of the ORC3 specific insertion is limited. Thus, we cannot see the C-terminal alpha helix of ORC6 binding ORC3, which has been previously seen in the yeast and *Drosophila* ORC structures.

The N-terminal cyclin box domain of yeast Orc6 binds similarly to Mcm2 and Mcm6, but it does not interact with Orc4. The C-terminal cyclin box domain of yeast Orc6 establishes contacts with Orc2 and Orc3. The C-terminal alpha helix of Orc6 also contacts Mcm5 and thereby bridges the Mcm2–Mcm5 DNA entry gate. In MCM–Cdt1, hOCCM and yOCCM, CDT1 establishes interactions with the same MCM subunits that are contacted by ORC6 in hMO*. In this context, the MCM2-MCM5 gate appears accessible.

**Extended Data Table 1 | Cryo-EM data collection, refinement and validation statistics – part 1**

| | hDH (EMD-19618) (PDB ID 8S09) | MCM5 AG DH (EMD-19625) | hSH (EMD-19619) (PDB ID 8S0A) |
|---|---|---|---|
| **Data collection and processing** | | | |
| Magnification | 130,000 | 92,000 | 130,000 |
| Voltage (kV) | 300 | 200 | 300 |
| Electron exposure (e$^-$/Å$^2$) | 49.28 | 50 | 49.28 |
| Defocus range (μm) | -1.0 to -2.5 | -1.0 to -2.5 | -1.0 to -2.5 |
| Pixel size (Å) | 1.08 | 1.61 | 1.08 |
| Symmetry imposed | C2 | C2 | C1 |
| Initial particle images (no.) | 970,326 | 2,875,825 | 178,853 |
| Final particle images (no.) | 15,874 | 72,117 | 25,069 |
| Map resolution (Å) | 3.1 | 5.6 | 3.2 |
| FSC threshold | 0.143 | 0.143 | 0.143 |
| Map resolution range (Å) | 2.4–6.6 | 4.8–7.1 | 3.1–8.7 |
| | | | |
| **Refinement** | | | |
| Initial model used (PDB code) | AlphaFold Multimer (ATPase-tiers + WHD, N-tiers) | | hDH (this study) |
| Model resolution (Å) | 3.1 | | 3.2 |
| FSC threshold | 0.5 | | 0.5 |
| Map sharpening $B$ factor (Å$^2$) | -44.9 | -482.8 | -46.0 |
| Model composition | | | |
| Non-hydrogen atoms | 64,149 | | 31,436 |
| Protein residues | 7,802 | | 3,821 |
| Ligands | 2 ATP, 8 ADP, 8 Mg$^{2+}$, 10 Zn$^{2+}$ | | 1 ATP, 4 ADP, 5 Mg$^{2+}$, 5 Zn$^{2+}$ |
| $B$ factors (Å$^2$) | | | |
| Protein | 9.96/128.22/49.15 | | 30.0/94.69/80.35 |
| Nucleotide | 5.50/54.29/23.15 | | 20.0/118.45/95.07 |
| Ligand | 23.63/63.91/41.18 | | 14.77/91.30/88.39 |
| R.m.s. deviations | | | |
| Bond lengths (Å) | 0.003 | | 0.002 |
| Bond angles (°) | 0.541 | | 0.486 |
| Validation | | | |
| MolProbity score | 1.28 | | 1.08 |
| Clash score | 3.11 | | 2.63 |
| Poor rotamers (%) | 0.00 | | 0.00 |
| CaBLAM outliers (%) | 1.29 | | 1.16 |
| Ramachandran plot | | | |
| Favored (%) | 96.94 | | 97.88 |
| Allowed (%) | 3.06 | | 2.12 |
| Disallowed (%) | 0.00 | | 0.00 |

**Extended Data Table 2 | Cryo-EM data collection, refinement and validation statistics – part 2**

| | MCM–ORC6 in hMO* (EMD-19620) (PDB ID 8S0B) | ORC1–5 in hMO* (EMD-19621) (PDB ID 8S0C) | Composite map & model of hMO* (EMD-19622) (PDB ID 8S0D) |
|---|---|---|---|
| **Data collection and processing** | | | |
| Magnification | 130,000 | | |
| Voltage (kV) | 300 | | |
| Electron exposure (e⁻/Å²) | 49.28 | | |
| Defocus range (μm) | -1.0 to -2.5 | | |
| Pixel size (Å) | 1.08 | | |
| Symmetry imposed | C1 | | |
| Initial particle images (no.) | 1,334,277 | | |
| Final particle images (no.) | 182,341 | | |
| Map resolution (Å) | 3.6 | 4.0 | Composite map of EMD-19620 and EMD-19621 |
| FSC threshold | 0.143 | 0.143 | |
| Map resolution range (Å) | 2.3–8.5 | 3.4–8.2 | |
| | | | |
| **Refinement** | | | |
| Initial model used (PDB code) | hSH (this study) AF-Q9Y5N6-F1 (human ORC6) | 7JPS (human ORC1–5) | MCM–ORC6 of hMO*, ORC1–5 of MO* (this study) |
| Model resolution (Å) | 3.6 | 4.0 | 3.6 |
| FSC threshold | 0.5 | 0.5 | 0.5 |
| Map sharpening $B$ factor (Å²) | -99.8 | -119.9 | |
| Model composition | | | |
| Non-hydrogen atoms | 32,427 | 14,133 | 46,027 |
| Protein residues | 3,834 | 1,591 | 5,424 |
| Ligands | 3 ATPγS, 3 ADP, 6 Mg²⁺, 5 Zn²⁺ | 2 ATPγS, 2 Mg²⁺ | 5 ATPγS, 3 ADP, 8 Mg²⁺, 5 Zn²⁺ |
| $B$ factors (Å²) | | | |
| Protein | 23.61/177.20/82.58 | 26.19/199.89/95.89 | 8.05/144.08/58.04 |
| Nucleotide | 210.23/296.91/249.44 | 103.41/384.46/302.06 | 173.65/279.27/220.61 |
| Ligand | 42.80/214.48/67.54 | 53.72/84.82/59.24 | 21.15/177.29/41.10 |
| R.m.s. deviations | | | |
| Bond lengths (Å) | 0.002 | 0.003 | 0.002 |
| Bond angles (°) | 0.497 | 0.531 | 0.431 |
| Validation | | | |
| MolProbity score | 1.30 | 1.74 | 1.35 |
| Clash score | 3.81 | 7.00 | 4.34 |
| Poor rotamers (%) | 0.00 | 0.00 | 0.00 |
| CaBLAM outliers (%) | 1.32 | 1.73 | 1.23 |
| Ramachandran plot | | | |
| Favored (%) | 97.29 | 95.93 | 97.31 |
| Allowed (%) | 2.71 | 4.07 | 2.69 |
| Disallowed (%) | 0.00 | 0.00 | 0.00 |

**Extended Data Table 3 | Cryo-EM data collection, refinement and validation statistics – part 3**

| | hOCCM (EMD-19623) (PDB ID 8S0E) | hOC$_1$M (EMD-19624) (PDB ID 8S0F) |
|---|---|---|
| **Data collection and processing** | | |
| Magnification | 130,000 | 130,000 |
| Voltage (kV) | 300 | 300 |
| Electron exposure (e$^-$/Å$^2$) | 49.28 | 49.28 |
| Defocus range (μm) | -1.0 to -2.5 | -1.0 to -2.5 |
| Pixel size (Å) | 1.08 | 1.08 |
| Symmetry imposed | C1 | C1 |
| Initial particle images (no.) | 1,334,277 | 1,334,277 |
| Final particle images (no.) | 34,116 | 49,771 |
| Map resolution (Å) | 3.8 | 4.1 |
|   FSC threshold | 0.143 | 0.143 |
| Map resolution range (Å) | 2.3–8.7 | 2.4–8.6 |
| | | |
| **Refinement** | | |
| Initial model used (PDB code) | AF-Q9H211-F1 (human CDT1) 7JPS, 7JPR (human ORC1–5) AF-Q99741-F1 (human CDC6) hSH (this study) | hOCCM (this study) |
| Model resolution (Å) | 3.8 | 4.1 |
|   FSC threshold | 0.5 | 0.5 |
| Map sharpening $B$ factor (Å$^2$) | -40.8 | -56.9 |
| Model composition | | |
|   Non-hydrogen atoms | 44,735 | 40,867 |
|   Protein residues | 5,347 | 4,863 |
|   Ligands | 5 ATPγS, 5 Mg$^{2+}$, 1 Zn$^{2+}$ | 5 ATPγS, 5 Mg$^{2+}$, 1 Zn$^{2+}$ |
| $B$ factors (Å$^2$) | | |
|   Protein | 115.10/687.12/278.32 | 113.54/921.49/293.58 |
|   Nucleotide | 231.37/360.13/283.56 | 228.91/395.14/300.87 |
|   Ligand | 132.81/366.95/186.89 | 126.79/379.37/199.07 |
| R.m.s. deviations | | |
|   Bond lengths (Å) | 0.002 | 0.002 |
|   Bond angles (°) | 0.517 | 0.484 |
| Validation | | |
|   MolProbity score | 1.38 | 1.36 |
|   Clash score | 6.95 | 6.07 |
|   Poor rotamers (%) | 0.00 | 0.00 |
|   CaBLAM outliers (%) | 0.82 | 0.77 |
| Ramachandran plot | | |
|   Favored (%) | 98.05 | 97.90 |
|   Allowed (%) | 1.95 | 2.10 |
|   Disallowed (%) | 0.00 | 0.00 |

# Reporting Summary

## Statistics

For all statistical analyses, confirm that the following items are present in the figure legend, table legend, main text, or Methods section.

| n/a | Confirmed | |
|---|---|---|
| ☒ | ☐ | The exact sample size (*n*) for each experimental group/condition, given as a discrete number and unit of measurement |
| ☒ | ☐ | A statement on whether measurements were taken from distinct samples or whether the same sample was measured repeatedly |
| ☒ | ☐ | The statistical test(s) used AND whether they are one- or two-sided<br>*Only common tests should be described solely by name; describe more complex techniques in the Methods section.* |
| ☒ | ☐ | A description of all covariates tested |
| ☒ | ☐ | A description of any assumptions or corrections, such as tests of normality and adjustment for multiple comparisons |
| ☒ | ☐ | A full description of the statistical parameters including central tendency (e.g. means) or other basic estimates (e.g. regression coefficient) AND variation (e.g. standard deviation) or associated estimates of uncertainty (e.g. confidence intervals) |
| ☒ | ☐ | For null hypothesis testing, the test statistic (e.g. *F*, *t*, *r*) with confidence intervals, effect sizes, degrees of freedom and *P* value noted<br>*Give P values as exact values whenever suitable.* |
| ☒ | ☐ | For Bayesian analysis, information on the choice of priors and Markov chain Monte Carlo settings |
| ☒ | ☐ | For hierarchical and complex designs, identification of the appropriate level for tests and full reporting of outcomes |
| ☒ | ☐ | Estimates of effect sizes (e.g. Cohen's *d*, Pearson's *r*), indicating how they were calculated |

*Our web collection on statistics for biologists contains articles on many of the points above.*

## Software and code

Policy information about availability of computer code

| Data collection | Gatan DigitalMicrograph and ThermoFisher EPU v3.2. |
|---|---|
| Data analysis | Topaz v0.2.5, crYOLO 1.9.3, MotionCor2, Gctf v1.06, CTFFIND v4.1.10 and v4.1.13, RELION v3.1 and v4.0, cryoSPARC 3.3.2 and 4.4.1, UCSF ChimeraX v1.6.9, ISOLDE v1.6.0, Coot v0.9.8.1 EL, Phenix v1.21, DNAproDB web sever. |

For manuscripts utilizing custom algorithms or software that are central to the research but not yet described in published literature, software must be made available to editors and reviewers. We strongly encourage code deposition in a community repository (e.g. GitHub). See the Nature Portfolio guidelines for submitting code & software for further information.

## Data

Policy information about availability of data

All manuscripts must include a data availability statement. This statement should provide the following information, where applicable:

- Accession codes, unique identifiers, or web links for publicly available datasets
- A description of any restrictions on data availability
- For clinical datasets or third party data, please ensure that the statement adheres to our policy

Data supporting the findings of this study are available within the paper and its Supplementary Information files. Cryo-EM density maps have been deposited in the Electron Microscopy Data Bank (EMDB) under the accession codes EMD-19618 (hDH), EMD-19625 (MCM5 AG DH), EMD-19619 (hSH), EMD-19620 (hMO*), EMD-19621 (locally refined ORC of hMO*), EMD-19622 (composite map of the globally refined hMO* and the locally refined ORC map), EMD-19623 (hOCCM), EMD-19624 (hOC1M). Atomic coordinates have been deposited in the Protein Data Bank (PDB) with the accession codes 8SO9 (hDH), 8SOA (hSH), 8SOB (MCM–ORC6 of hMO*, which was modelled using the globally refined hMO* map), 8SOC (ORC1–5 of hMO*, which was modelled using the locally refined map of ORC), 8SOD (composite model of MCM–ORC6 and ORC1–5 docked into the composite map), 8SOE (hOCCM), 8SOF (hOC1M).

# Field-specific reporting

Please select the one below that is the best fit for your research. If you are not sure, read the appropriate sections before making your selection.

☒ Life sciences ☐ Behavioural & social sciences ☐ Ecological, evolutionary & environmental sciences

For a reference copy of the document with all sections, see nature.com/documents/nr-reporting-summary-flat.pdf

# Life sciences study design

All studies must disclose on these points even when the disclosure is negative.

| | |
|---|---|
| Sample size | Negative stain EM experiments were used to determine the nucleotide and protein dependency of the human MCM loading reaction. To obtain 2D classes of different reaction intermediates, we usually collected 60-150 micrographs per condition.<br><br>To yield high-resolution structures of hDH and hSH in the cryo-EM experiment, ~3.6 K micrographs were collected from a single grid. High-resolution structures of hOCCM, hOC1M and hMO* were obtained from ~31.6 K micrographs, which were collected from a single grid.<br><br>No statistical methods were used to predetermine sample size. Sample size was chosen to obtain interpretable negative stain 2D averages and cryo-EM volumes of resolution appropriate for atomic model building. |
| Data exclusions | Negative stain and cryo-EM micrographs with poor staining, ice contamination or without graphene oxide, respectively, were excluded. Picked particles that did not align to a distinct class in 2D and 3D (cryo-EM only) were excluded from further analysis. |
| Replication | The different complexes identified in this study were visualised in multiple experiments (both negative stain and cryo-EM). All findings from nuclease footprinting assays were reproducible. Please refer to statistics and reproducibility section of the manuscript. |
| Randomization | For 3D reconstructions independent halves of the dataset were used to compute Fourier Shell Correlation. |
| Blinding | Blinding is not relevant for single particle reconstruction, because reference free models are used. |

# Reporting for specific materials, systems and methods

We require information from authors about some types of materials, experimental systems and methods used in many studies. Here, indicate whether each material, system or method listed is relevant to your study. If you are not sure if a list item applies to your research, read the appropriate section before selecting a response.

### Materials & experimental systems

| n/a | Involved in the study |
|---|---|
| ☒ | ☐ Antibodies |
| ☐ | ☒ Eukaryotic cell lines |
| ☒ | ☐ Palaeontology and archaeology |
| ☒ | ☐ Animals and other organisms |
| ☒ | ☐ Human research participants |
| ☒ | ☐ Clinical data |
| ☒ | ☐ Dual use research of concern |

### Methods

| n/a | Involved in the study |
|---|---|
| ☒ | ☐ ChIP-seq |
| ☒ | ☐ Flow cytometry |
| ☒ | ☐ MRI-based neuroimaging |

# Eukaryotic cell lines

Policy information about cell lines

| | |
|---|---|
| Cell line source(s) | Sf9 cells were purchased from Thermo Fisher (12659017). RPE-1 and U2-OS cells were obtained from the Crick Cell Services science technology platform. |
| Authentication | None of the cell lines were authenticated. |
| Mycoplasma contamination | All cell lines were tested negative for mycoplasma contamination. |
| Commonly misidentified lines<br>(See ICLAC register) | No commonly misidentified cell lines were used in this study. |

