## [Peer Review File · Nature]

Manuscript Title: MCM Double Hexamer Loading Visualised with Human Proteins

Reviewer Comments & Author Rebuttals

Reviewer Reports on the Initial Version:

Referee #1:

Here, Weissmann and Greiwe et al. report the first in vitro reconstitution of MCM double hexamer (DH) assembly with human proteins, which is the first step in replisome assembly. Using benzonase footprinting assay the authors define the requirements for DH formation and find that ORC6 protein is dispensable. ORC6 stimulates loading with full-length protein but inhibits loading with truncated proteins, indicating that there are ORC6-dependent and -independent MCM loading pathways. Cryo-EM structures of DH and single hexamers (SH) loaded with truncated proteins (likely ORC6-independent) are presented. The DH is essentially the same as the DH that was purified from human cells, aside from different ATPase site occupancy. DNA is melted in the DH but not the SH and residues seen to be stabilizing melted DNA are mutated. Mutant proteins can still form DH but they are more mobile, which probably explains why DH was not isolated from cells with these mutants. Cryo-EM of complexes formed in presence of ATP- γ -S, which doesn't support efficient DH formation, are also determined. An OCCM complex similar to the budding yeast complex and an OCM. The structure of an ORC6-dependent MO* complex is also presented. This complex is different from yeast MO identified by the authors previously and would require rearrangement before ORC could recruit a second MCM hexamer. Curiously, fully loaded SH and MO* can also be formed in the absence of ATP hydrolysis (ATP- γ -S condition), CDT1 or CDC6.

The reconstitution of human MCM loading is a major advance. The data are generally of high quality and important structural and mechanistic insights are provided. Most of the conclusions seem sound based on data. The clarity of the manuscript and organization of some data could be improved. Sometimes the narrative is not so intuitive and, in several places, additional experimental detail should be provided. Some of the structural descriptions are longer than necessary. We appreciate that a lot of data is presented and developing a model consistent with all observations is challenging. However, we think the manuscript would benefit from greater discussion of the model and certain aspects of the data (see below). Although the model is mostly consistent with the data, no direct evidence is provided showing that DH can be formed from SH or how MO* leads to DH. There are also areas of the model that we think need to be reconsidered. Overall, due to this being the first reported reconstitution of human MCM loading and the high quality of the biochemical and structural data, we support publication in Nature subject to the following comments being addressed:

- We think the order of results at the beginning of the manuscript should be reconsidered. The first section describes attempts to perform MCM loading on bead-bound DNA and is shown in ED Fig. 1. These experiments do not find evidence for salt stable MCM. The authors then write: "Nonetheless, negative-stain EM imaging of a human MCM loading reaction on a short synthetic yeast origin flanked by nucleosomes yielded 2D averages similar to yeast DH, suggesting that hDH was being

assembled". We presume these EM experiments were done in solution and not on beads? The description in the results makes this ambiguous. If they were done in solution, this should be made clear. We find it strange that these EM experiments are included as an ED figure, rather than in Fig. 1 because they demonstrate the first reconstitution of human DH with purified factors.

- Some elements of the model in Fig. 5 seem contradictory. In 5a two loaded SH diffuse along DNA and meet to form DH. In 5c, MO* that is formed without CDT1 and CDC6, releases SH along DNA. If this can happen as depicted, two SH loaded in this manner should be able to form DH, just like in 5a. However, CDT1 and CDC6 are shown to be essential for DH formation. The model therefore invokes that the second SH must be loaded via OCCM but I can't see any reason for this. No explanation is provided for why the SH that is released from MO* in 5c is any different to the SH that is released from OCCM in 5a. They are depicted as being exactly the same thing in the model.
- The ATPase site occupancy for SH and DH is the same and DH but not SH has unwound DNA. It would be helpful if the authors could discuss how they envisage unwinding/melting occurring when two loaded SH meet in pathways a, b and c of the model? Do they envisage ATP hydrolysis being required? If so it wouldn't seem entirely compatible with the model that two SH undergo linear diffusion to form a DH. If ATP hydrolysis is not required, why wouldn't two single hexamers loaded in ATP- γ -S form a DH if the SH from ATP/ATP- γ -S are said to look indistinguishable (line 229/230) (apart from presumably ATPase site occupancy)?
- The size of the hDH footprint changes depending on the length of the benzonase digestion. It's larger than yeast DH with a 0.5 min incubation but more similar to yeast DH with a 20 min incubation. This is not commented on. Related, why are different length benzonase treatments used in Fig. 1?
- In Figs. 2g and 3h the footprint for WT gets smaller over time and resembles the smaller footprint seen with the mutants in 2f. What might be the reason for this?
- The MCM5 mutants that are analyzed in Fig. 2e are done with no ORC6 and truncated proteins. Because a key discovery of this work is the multiple MCM loading pathways the mutants should also be tested under loading conditions where ORC6 is required.
- Some of the structural descriptions are unnecessarily detailed. Apart from ATPase site occupancy differences the DH structure seems to be identical to the published hDH structure so the text could be condensed.
- More detail should be provided for the plasmid used to make DNA for EM experiments. It is not clear from the methods how close the Widom sequences are to the ACS so it is difficult to gauge exactly how big the nucleosome free region is.

Minor points

Presumably the nucleosomes used on the DNA for EM are yeast? This should be made clear to the reader.

Fig. 1a, right: MCM2-7 are not truncated.

What is the band at the bottom of Fig. 1j?

Fig. 2d legend: It should be MCM5 not Mcm5.

Fig. 2 legend: Some of the marker annotations overlap each other. This is also seen in some ED figures.

Line 198: OC1M can't be described as a subcomplex of OCCM because it doesn't contain CDC6.

Line 315: Fig. 1x is referenced.

Why was a different DNA construct used for imaging MCM AG?

In Fig. 5 why is one of the MCM subunits in the MO states colored yellow? This is not mentioned in the legend.

Referee #2:

The paper by Weissmann et al. reports the biochemical reconstitution with purified proteins of the human DNA helicase MCM2-7 loading on DNA, a key step in the initiation of DNA replication, and the cryoEM analysis of loading intermediates.

To restrict DNA replication to once per cell cycle, cells load MCM onto origin DNA in G1 as an inactive double hexamer, a key step known as origin licensing. MCM loading is dependent on the Origin Recognition Complex (ORC), a hetero-hexameric complex that together with Cdc6 engages MCM - Cdt1 to load sequentially two copies of MCM onto dsDNA. Our current knowledge of the biochemical mechanism of origin licensing relies mainly on experiments in the model system budding yeast, that have yielded structures of the ORC - MCM intermediates ORC-Cdc6-Cdt1-MCM (OCCM), MCM - ORC (MO) and MCM double hexamer (MCM-DH), as well as the biochemical reconstitution of MCM loading on origin DNA with purified components. Recently, a high-resolution cryoEM structure of a MCM-DH loaded on dsDNA recovered from human cells was also reported.

The sequence conservation of all protein components of the origin licensing apparatus indicates that the mechanism will be broadly conserved in eukaryotes; however, little is known about how widely applicable the mechanistic model of origin licensing gleaned from yeast is and specifically how much it will differ in human cells. The paper by Weissmann et al. therefore addresses an important research question in a timely fashion.

MAIN FINDINGS

Weissman et al. report the biochemical reconstitution of human MCM-DH formation on DNA using purified proteins. They find that MCM-DH can be loaded on DNA in the absence of Orc6 (essential for yeast MCM-DH loading), albeit at reduced levels. Interestingly, absence of ORC6 stimulates loading with truncated ORC1, CDC6, CDT1 lacking the IDRs. The ORC1 IDR mediates at least partially the effect of ORC6 on loading. Loading of human MCM-DH is salt sensitive, unlike yeast MCM-DH, more so in assays that use FL proteins (see specific point below). The authors also report high-resolution cryoEM structures for MCM-DH as well as for OCCM, MO and MCM-SH. The structure of MCM-DH confirms the published structure of MCM-DH obtained from cells. The OCCM structure resembles that of the yeast OCCM. The structure of the MO complex shows binding of ORC to the N-tier of MCM mediated by ORC6 as seen in the yeast MO; however, the ORC is positioned close to the MCM in a conformation that is not compatible with engagement of the second MCM.

MAIN COMMENTS

The authors bring an impressive amount of high-quality biochemical and structural data to bear on the question of the molecular mechanism of origin licensing in human cells. The biochemical reconstitution and characterisation of human MCM-DH loading with purified components is an important step forward in the quest to understand the activation mechanism of the CMG helicase. My main critique is that none of the novel observations reported shifts decisively the paradigm that we have learnt from yeast, thanks also to major earlier contributions from the authors' laboratories. In fact, what impresses this reviewer most is the degree of evolutionary conservation of these

mechanisms, something we have learned already from the structures of yeast and human replisomes.

The observations that some MCM-DH loading can be obtained in the absence of ORC6 is interesting, but it is difficult to assign it its proper biological significance in isolation. Given that human cells perform origin licensing with full-length proteins and in the presence of ORC6, when would this pathway become relevant? Can we be reasonably sure that in the chromatin environment of the cell ORC6 might not be required? I appreciate that this is a biochemical/structural study but for publication in Nature some further proof should be warranted.

The authors have completed an impressive cryoEM tour-de-force and obtained high-resolution information for all of the loading intermediates that have been observed in yeast. The most interesting is the MO*, a putative human yeast-like MO intermediate that does not support a straightforward application of the yeast model for loading of the second MCM copy due to steric clash. The interpretation of the role of this intermediate is uncertain; the authors postulate an alternative route whereby the MO* loads single MCM-SH hexamers that can slide towards an OCCM intermediate (or another MCM-SH?) to yield an MCM-DH. Is it possible that other, more open MO intermediates exist that might be compatible with the 2-step ORC-flip model proposed earlier by the authors? Can the authors confirm that they were unable to observe any MO + MCM species? Is it possible that the 2-step ORC flip model evolved uniquely in yeast to exploit the presence of juxtaposed binding sites at the origin?

SPECIFIC COMMENTS

The authors state that CDC6 is not required in the MCM loading reaction of ED Fig. 1b. However, the reaction lacking CDC6 shows no MCM loading after 150 mM NaCl wash, unlike the loading reaction with all components. Please clarify.

I agree that the ORC1 IDR seems to mediate the ORC6-dependent stimulation. However, removal of the ORC1 IDR stimulates MCM loading in the absence of ORC6 (Fig. 1g, compare 3xFL and truncated ORC1 in the absence of Orc6). So the ORC1 IDR seems to have an intrinsic inhibitory effect on loading. Please comment.

Removal of all three IDRs leads to much higher levels of MCM-DH loading relative to truncating ORC1 IDR only, in the absence of ORC6 (Fig. 1g, compare 3xDN). This would suggest further IDR-dependent effects, beyond the effect mediated by ORC1 IDR alone. Please comment.

The authors should make clear in the text that the salt-resistance experiments of Fig. 1j were performed with truncated proteins. The loading reactions in the DNA beads assay of ED Fig. 1a, performed with FL proteins, are sensitive to salt above 150 mM NaCl, whereas the loading reactions in the nuclease footprinting assay of Fig. 1j with truncated proteins are salt-resistant up to 0.5 M NaCl. What is the salt resistance when the assay of Fig. 1j is performed with FL proteins?

In Fig. 2g, two bands are clearly distinguishable after benzonase digestion of the WT MCM loading reaction, the shorter of which looks similar in size to the shorter product seen with the AG mutant. Further reduction of the MCM footprint on DNA is also noticeable in Fig. 1, comparing panels reporting 0.5 min incubation with benzonase (DNA runs as the 75 marker size) or 20 min incubation (DNA runs just above 50). The authors should interpret this observation.

A simple explanation for the increased mobility of the AG MCM-DH double mutant is that R195 and L209 act as pins by protruding into the double helix and reducing the any residual sliding between protein and DNA.

This reviewer is unclear about how it is possible to obtain a fully loaded MCM-SH with ATP γ S, as the authors observe by cryoEM and propose in Fig. 6c. It would be helpful if they commented on it in the Discussion, perhaps at the expense of the text dedicated to the MCM-DH, which can be usefully reduced in size.

MINOR POINTS

Line 187: Please define the MCM6-2 and MCM4-6 “A” domains, for MCM-naive readers.

Line 155: Full stop missing.

Line 256: (ED Fig. 10a) should be ED Fig. 10b?

Referee #3:

This manuscript reports the reconstitution of origin licensing using human proteins and the structural characterization of the double hexamer (DH) and intermediates in this process. The structural studies show that the resulting DH has the same characteristics as a previously solved structure of the human DH purified from licensed origins in human cells. This correspondence provides strong evidence that the *in vitro* process is recapitulating human origin licensing. A key conclusion of the paper is that there are at least two pathways for human origin licensing. One that requires and one that does not require Orc6, the smallest subunit of ORC. Because Orc6 is not as tightly associated with the remaining human ORC subunits, there has been debate about its involvement in human origin licensing, and these studies suggest that it does play a role. They perform extensive structural studies of assemblies present in the loading reaction in the presence of Orc6, identifying two complexes similar to but different from the yeast ORC-Cdc6-Cdt1-Mcm2-7 (OCCM) and MO complexes as well as both double and single Mcm hexamer structures. The authors characterize loading reactions in which Orc6 is left out using a novel assay for helicase loading, protection of plasmid DNA from digestion with a non-specific nuclease. Based on these studies the authors suggest that there is a second mechanism for human helicase loading that does not use Orc6 and instead involves independently loaded Mcm2-7 complexes sliding into one another.

The strengths of the paper are several fold. First, reconstitution of human origin licensing with purified proteins that creates a cryoEM solvable DH (and potential intermediates) is a major advance for the field. The strongest aspect of the paper are the structural studies of a variety of assemblies observed in reactions that lead to DH formation. The DNA protection assay for helicase loading is also a potential step forward in the assessment of human origin licensing, but requires further characterization.

The structural studies are well executed and focus on the reaction in the presence of Orc6. Overall, these studies suggest that a similar reaction as seen in yeast is also occurring in human cells in the presence of Orc6 with some notable differences. As mentioned above, the structural characterization of the human DH shows the same differences from yeast as was observed in the previously published structure of DHs purified from human cells: the DNA in the central channel is partially unwound with one base pair completely lost. An important contribution of this study is that it demonstrates that the partially unwound DNA observed when DHs are purified from cells is a consequence of loading and not a downstream activation event. One difference from the human cell-derived DH is that the nucleotide occupancy is distinct, with most of the sites occupied by ADP. This occupancy is similar to what is observed in yeast, and this observation suggests that Mcm2-7 ATP hydrolysis is involved in human helicase loading as is known for the yeast reaction. A second significant difference from the yeast studies comes from the analysis of the structure of a "MO-like" structure (hMO*). Unlike its potential yeast counterpart, the ORC molecule in the determined structure of the hMO* is not in a position to recruit a second Mcm2-7 as it is in the yeast MO (yMO). This finding raises the possibility that hMO* is not an intermediate in human helicase loading but instead prevents a non-productive pathway from occurring. The latter viewpoint is consistent with the finding that the hMO* can be observed in the absence of Cdc6 and Cdt1. On the other hand, it does not explain why Orc6 stimulates the DNA protection assay in the presence of the full-length ORC, Cdc6 and Cdt1 proteins but does the opposite with mutant versions of the proteins missing

their IDR (see below).

The authors also determine the structures of two other Mcm2-7 containing complexes. First, they determine a structure of single Mcm2-7 hexamers around DNA and show that the Mcm2-5 gate through which DNA enters the Mcm ring is in a closed state. Such an intermediate has not been characterized in yeast previously, suggesting that this could be an intermediate that is unique to the human reaction. In addition, they determine the structure of a human ORC-Cdc6-Cdt1-Mcm2-7 (OCCM) complex. The ORC-Mcm interface in this complex has some differences relative to the yeast structure although the significance of these are not investigated. The Cdt1-Mcm2-7 interface is also analogous to the yeast OCCM despite the finding that the two proteins don't interact in solution in human cells (but do in yeast). One significant difference is that Orc6 is not observed in the human OCCM. This is not unexpected given the relatively lower affinity of human Orc6 for Orc1-5. Overall, the structural studies provide strong evidence that the reconstituted reaction used in the cryoEM experiments leads to the formation of a relevant Mcm2-7 DH. Although these structural studies do not identify a clear pathway for DH formation in human cells, they do provide structures that could facilitate further studies to reveal such mechanism(s).

A major concern with the paper is the discrepancies between the benzonase protection assay and the EM studies. While the benzonase-protection studies focus on reactions without Orc6, the EM studies used to solve structures are performed with Orc6. A particularly confusing aspect of the study arises when the studies are compared. For example, all the EM studies are done with the deleted Orc1, Cdc6, and Cdt1 and mostly with Orc6 (all structural studies included Orc6, but some 2D experiments did not). According to the benzonase protection assay this should lead to very poor DH formation, yet the authors get enough DHs and other potential intermediates using this combination of proteins to determine their structures. This raises questions about the validity of either the benzonase assay or the EM assay to detect DHs. It is possible that the EM assay detects unstable versions of a DH structure. Alternatively, the benzonase assay may not be an accurate measure of DHs. Another important area of discrepancy is that cryo-EM structural studies are not performed on the DHs formed in the absence of Orc6. Because the DH structure is the strongest evidence that human helicase loading can be reconstituted with purified proteins and is forming a correct (and hopefully active) DH, this absence greatly reduces the relevance of the studies without Orc6. Finally, that they chose to study the structures from a condition that is non-productive in their DNA protection assay reduces the value of the structural studies (see below specific point 6 about potential differences between the EM and benzonase assays that could explain such discrepancies).

The idea of a "double OCCM" as an intermediate in loading becomes a theme for a proposed Orc6-independent alternative mechanism for human origin licensing. A significant weakness of these studies is the lack of evidence that the benzonase-protection interpreted to be a double-OCCM is in fact due to this complex. For example, instead of being double the size of the OCCM protection (50 bp), it is 2.5 times larger. This suggests a role for other complexes in this larger protection (e.g. an hMO?). In favor of the double OCCM hypothesis, the authors provide evidence that the "OCCM" has increased benzonase sensitivity which they interpret as sliding on DNA (note: this experiment is missing a critical control, see below). On the other hand, both the lengthy time required for appearance of the "double OCCM" protection (Fig. 3f) relative to DH footprint formation (Fig. 1c) and the apparent lack of prevalence of the "double OCCM" in the EM studies of reactions with ATP

suggests that this is not a common mechanism/intermediate.

Overall, the data presented shows that the authors have developed biochemical assays to assemble human Mcm2-7 into double hexamers on DNA and they have used this to analyze the structure of potential intermediates in this process. These structures will be valuable in the subsequent determination of the mechanism of this event. The data presented does not allow the authors to strongly support any of the models presented in Fig. 5 and instead they are mostly derived from analogy with the yeast system. The strongest conclusion of the paper is that there is an Orc6-dependent and a possible Orc6-independent mechanism (but see specific point 2 below). The finding that they only see the Orc6-independent mechanism with mutant forms of Orc1, Cdc6, and Cdt1 that would not be present in cells makes significantly reduces the biological interest of this potential pathway.

Specific points:

1. One significant concern with the manuscript is the extensive use of a benzonase protection assay to identify protein complexes associated with DNA. Although an interesting assay, the authors cannot use this assay to demonstrate what proteins are responsible for the observed protections. While they perform a number of experiments that suggest the DH is responsible for the protected DNA, there are important discrepancies with the EM studies (see above). The inability to identify the associated proteins responsible is a significant concern for the double OCCM. Without more direct evidence, the authors need to be more careful in their statements concerning the components of the assemblies responsible for DNA protection in these assays. It would greatly improve the paper to include some assay that identifies the proteins in these complexes. Although the authors frequently refer to the EM studies to support their conclusions, there are many different assemblies observed in these EM grids, any one of which could be responsible.

2. In ED Fig. 1b, the “no Orc6” lane shows an easily detectable band at the position of Orc6. Could this be insect cell Orc6? This is a critical issue that needs to be addressed given that the major take home message of the paper relies on this reaction not having Orc6. While the gels of the purified proteins show no bands at this point in the Orc1-5, Cdc6, or Cdt1 proteins, small amounts of contaminating Orc6 could be present and become concentrated on the DNA during the reaction. It would be much more compelling if the authors included analysis of a mutant in Orc1-5 that could no longer interact with Orc6.

3. The EM grids that are presented in Ext. Data Fig. 2 are frequently referred to in the paper with a focus on what assemblies are or are not present. The authors must present this data in a more quantitative manner to allow evaluation of their significance. A good way to do this would be to report the percentage of Mcm complexes observed in each type of assembly under each condition. This would be particularly helpful with regard to the “double OCCM”. An averaged EM image of this is shown (Fig. 3g) and is apparently derived from an ATP γ S experiment, but there is no evidence as to its frequency in any of the EM grids. Without a sense of frequencies, these data are of limited use. Given that the authors indicate that they are using computational analysis to identify the different assemblies this should be easy to do. The authors should also indicate what percentage of particles were left unassigned/uncharacterizable.

4. If the authors want to propose that the double OCCM is an important intermediate in the absence of Orc6, it should be observed in experiments with ATP and it should be inhibited by addition of Orc6. Tests of both of these predictions could and should be included in the manuscript.

5. In the discussion, the authors suggest that the hMO* could act to stabilize single hexamers (SHs) on the DNA; however, earlier in the paper they suggest that SHs observed are stably closed around the DNA. The authors should address this discrepancy.

6. One concerning aspect of the manuscript is the use of different protein concentrations in different assays without explanation. For example, they use a two-fold molar ratio of Orc1-5 and Orc6 in the experiments with the deleted proteins but a 1:1 ratio of Orc1-5 and Orc6 in the full-length experiments and cryoEM experiments (using the deleted proteins). This raises the possibility that equimolar Orc6 is beneficial and excess Orc6 is inhibitory. For an accurate comparison, these experiments should be repeated using the same protein concentrations. Similarly, they use lower concentrations of most of the proteins in the bead-based assays than they use in the benzonase assay. Could this explain the lack of signal in the bead-based assays? Finally, although the authors show that lack of the Orc1-IDR reduces the extent of Orc6 stimulation, comparison of Fig. 1g lanes 1&2 vs lanes 5&6 shows that loss of the Cdc6 and Cdt1 IDRs further reduces Orc6 stimulation. It is concerning that there are no repetitions and quantification of any of the benzonase experiments to demonstrate that such differences are significant. It would also improve Fig. 1 if the authors stuck with one time of benzonase treatment (either 0.5 or 20 min).

7. One oddity of the paper is that the authors are not able to identify protein associations in the bead pull-down assay that has been a reliable measure of helicase loading protein-DNA association in the yeast assay. Does this mean that the complexes they are observing by benzonase footprinting are short-lived? Is the reaction very inefficient? Why are so many intermediates observed by EM but not by other assays? Could these differences have to do with the different protein concentrations used in the different assays mentioned above?

8. The comparison of an ARS1-containing plasmid and a human origin-containing plasmid is not useful since the experiments were not done in the same plasmid backbone. Also, there is a notable difference in the benzonase footprint on different DNA templates suggesting that the ARS1 sequence is enhancing (with the caveat that the plasmids have different backbones and the authors have not established the quantitative nature of this assay). Does this mean that human origins inhibit the reaction they are studying? If the authors want to explore the role of DNA sequence in this reaction, they need to do a better experiment. If not, the current experiment should be removed from the paper.

9. The authors should consistently indicate in the legends for the figures whether they are using the full-length or deleted Orc1, Cdc6 and Cdt1 proteins in the assays shown. For example, this is not indicated in ED Figs. 1 or 2 legend and should be.

10. The evidence that the benzonase assay detects sliding strongly depends on experiments showing that it is the time in benzonase rather than the time of the reaction that leads to loss of signal. For

Fig. 2g the authors include the appropriate control to demonstrate this is the case (last two lanes). This same control is equally important for the experiments suggesting the "OCCM" slides in Fig. 3h. Without the control the change in sensitivity is equally likely to be due to instability and the interpretation would need to be changed.

Author Rebuttals to Initial Comments:

Response to the referees:

Reviewer 1:

We are delighted that the reviewer recognises that “the reconstitution of human MCM loading is a major advance”, that our data are “generally of high quality” and that the structural and mechanistic insights provided are “important”. It is also pleasing that the reviewer considers most of our conclusions “sound based on data”. We thank the reviewer for stating that they “support publication in Nature [...] subject to [their] comments being addressed”.

The clarity of the manuscript and organization of some data could be improved. Sometimes the narrative is not so intuitive and, in several places, additional experimental detail should be provided. Some of the structural descriptions are longer than necessary. We appreciate that a lot of data is presented and developing a model consistent with all observations is challenging. However, we think the manuscript would benefit from greater discussion of the model and certain aspects of the data (see below).

We have now improved the narrative and the description of the experimental procedure. We have cut down on the description of the DH structure and expanded on the discussion of our data and mechanistic models.

Although the model is mostly consistent with the data, no direct evidence is provided showing that DH can be formed from SH or how MO* leads to DH. There are also areas of the model that we think need to be reconsidered.

The reviewer makes a fair point. In our first submission several models for helicase loading were presented, but we did not elaborate on which mechanisms are more likely to support DH formation based on our data. This was mainly because key intermediate assemblies could be hypothesised but not observed directly, or otherwise could be observed, but in conditions (ATP- γ -S) that support helicase recruitment but not full DH loading. As described below, we have now collected multiple repeats of large negative stain EM datasets (ED Fig. 3a). We provide evidence that, in the ATP hydrolysis-powered MCM loading reaction, double OCCM assemblies can be observed, alongside OCCM encountering a single loaded MCM. Our observations provide new support for two of the DH loading mechanisms proposed. Other models such as the diffusion of two single hexamers that meet via their dimerising N-termini, or DHs forming via a mechanism akin to the yeast MO are possible and deserve further investigation. We note that it took more than 15 years from the first structural evidence that the DNA-loaded yeast MCM forms DHs, until we reached a complete understanding of helicase loading in that model system. Likewise, it will take time to establish whether all the protein assemblies observed in our first EM experiments with the human system are bona fide intermediates of DH loading.

- We think the order of results at the beginning of the manuscript should be reconsidered. The first section describes attempts to perform MCM loading on bead-bound DNA and is shown in ED Fig. 1. These experiments do not find evidence for salt stable MCM. The authors then write: “Nonetheless, negative-stain EM imaging of a human MCM loading reaction on a short synthetic yeast origin flanked by nucleosomes yielded 2D averages similar to yeast DH, suggesting that hDH was being assembled“. We presume these EM experiments were done in solution and not on beads? The description in the results makes this ambiguous. If they were done in solution, this should be made clear. We find it strange that these EM experiments are included as an ED figure, rather than in Fig. 1 because they demonstrate the first reconstitution of human DH with purified factors.

We agree with the reviewer that this passage should have been clearer and the data better presented. We now decided to omit our attempt to perform DH loading on beads. Also, we now explain that the negative-stain EM imaging of the human MCM loading reaction was performed in solution and not on beads. We previously decided to include all the negative stain EM analysis in an Extended Data Figure for the sake of completeness, given that many conditions were tested resulting in several observation for each condition. We feel that adding a negative stain class average of a human DH in Figure 1 would be redundant, considering that Figure 2 is fully focused on the high resolution cryo-EM structure.

- Some elements of the model in Fig. 5 seem contradictory. In 5a two loaded SH diffuse along DNA and meet to form DH. In 5c, MO* that is formed without CDT1 and CDC6, releases SH along DNA. If this can happen as depicted, two SH loaded in this manner should be able to form DH, just like in 5a. However, CDT1 and CDC6 are shown to be essential for DH formation. The model therefore invokes that the second SH must be loaded via OCCM but I can't see any reason for this. No explanation is provided for why the SH that is released from MO* in 5c is any different to the SH that is released from OCCM in 5a. They are depicted as being exactly the same thing in the model.

Prompted by this comment from the reviewer, we investigated loading factors requirement in DH formation further. We established that not only ORC6 but also CDC6 can be omitted from a human MCM loading reaction and still result in DH formation (as observed by salt stable signature DNA footprint and negative stain EM). CDT1 however is strictly required for DH formation, indicating that at least one of the two MCM rings loaded likely derives from OCCM or OC1M (ORC-CDT1-MCM), whose structure we described to 3.8 and 4.1 Å resolution respectively. We observe MO* in the absence of CDC6 or CDT1, meaning that single MCM hexamers can be topologically closed around DNA without visiting the OCCM or OC1M state. As the reviewer points out, our data imply that SHs released from OCCM/OC1M and from MO* are somehow different. This could possibly be a different nucleotide occupancy, and this is something we now elaborate on in the discussion.

- The ATPase site occupancy for SH and DH is the same and DH but not SH has unwound DNA. It would be helpful if the authors could discuss how they envisage unwinding/melting occurring when two loaded SH meet in pathways a, b and c of the

model? Do they envisage ATP hydrolysis being required? If so it wouldn't seem entirely compatible with the model that two SH undergo linear diffusion to form a DH.

This is a great point of discussion. We changed Figure 5 to suggest the possibility that, after passive diffusion, the locking of two DNA-loaded SHs to form a DH might require ATP hydrolysis. Alternatively, DH locking might happen concomitantly with the ATP hydrolysis driven release of the ORC, Cdc6 and Cdt1 loading factors that comes with the maturation of one (or two) OCCM(s) to loaded MCM, now mentioned in the Discussion (see below).

If ATP hydrolysis is not required, why wouldn't two single hexamers loaded in ATP- γ -S form a DH if the SH from ATP/ATP- γ -S are said to look indistinguishable (line 229/230) (apart from presumably ATPase site occupancy)?

This is a good point and an argument in favour of ATP-hydrolysis dependent locking of the DH, which we now address in the Discussion.

The size of the hDH footprint changes depending on the length of the benzonase digestion. It's larger than yeast DH with a 0.5 min incubation but more similar to yeast DH with a 20 min incubation. This is not commented on. Related, why are different length benzonase treatments used in Fig. 1?

This is true and we agree that this point requires discussion. After longer (20 min) Benzonase digestions, the lengths of the products with yeast and human DHs are similar to each other and are about 55 bp. This is very similar to the average DNA fragment size bound by MCM DH sequenced by Li et al. 2023 (54-56 bp) and we believe represents a 'complete digestion' product. After shorter digestion (e.g. 0.5 min) there is a longer (~ 75 bp) intermediate seen primarily with the human wt DH. It is unlikely that this is due to some other protein (e.g. ORC) still bound to a DH because we see the same intermediate sized products with Benzonase digestion after transient high salt treatment (Fig. 1j). It is possible that one or more of the C-terminal MCM WHDs in human MCM bind DNA better than their yeast counterparts, or in some non-specific way hinder Benzonase from easy access to the channel exit. Alternatively, the yeast DH can slide whilst the human DH does not and this may somehow affect the nuclease patterns. We include a new Extended Data Figure 1b showing a direct comparison of Benzonase products in a time course with human and yeast DH and we have modified the text on p 3 to reflect these differences. Where possible, we used a 20 min Benzonase digestion to show the complete digestion product. For snapshots (e.g. time courses to monitor sliding) we used the shorter (0.5 min) digestions.

- In Figs. 2g and 3h the footprint for WT gets smaller over time and resembles the smaller footprint seen with the mutants in 2f. What might be the reason for this?

It is true that all of the MCM5 mutants have less of the 75 bp product and more of the 55 bp product. This correlates with the fact that the mutants can slide on DNA while the wt DH cannot and suggests that the ability to slide somehow favours the 55 bp product.

We have made this explicit on p. 6.

- The MCM5 mutants that are analyzed in Fig. 2e are done with no ORC6 and truncated proteins. Because a key discovery of this work is the multiple MCM loading pathways the mutants should also be tested under loading conditions where ORC6 is required.

We agree with this point and have included a new experiment (Extended data Fig.6d) showing that the mutants behave very similarly when loading by full length proteins in reactions containing ORC6.

- Some of the structural descriptions are unnecessarily detailed. Apart from ATPase site occupancy differences the DH structure seems to be identical to the published hDH structure so the text could be condensed.

We followed the reviewer's suggestion and condensed the text describing the human DH structure.

- More detail should be provided for the plasmid used to make DNA for EM experiments. It is not clear from the methods how close the Widom sequences are to the ACS so it is difficult to gauge exactly how big the nucleosome free region is.

The Widom sequences map 7 and 5 base pairs away from the inverted ACS sites, making the nucleosome free region 148 base pairs long. Now stated in the Methods section.

Minor points

Presumably the nucleosomes used on the DNA for EM are yeast? This should be made clear to the reader.

The reviewer is correct. We now state in the methods that the nucleosome used in the EM experiments are reconstituted using yeast histones.

Fig. 1a, right: MCM2-7 are not truncated.

Thank you for spotting this issue. We fixed the labelling.

What is the band at the bottom of Fig. 1j?

The band at the bottom of Fig. 1j appears because of incomplete digestion due to the higher salt concentration used. We now explain this in the figure legend.

Fig. 2d legend: It should be MCM5 not Mcm5.

Thank you for spotting this, corrected.

Fig. 2 legend: Some of the marker annotations overlap each other. This is also seen in some ED figures.

Fixed.

Line 198: OC1M can't be described as a subcomplex of OCCM because it doesn't contain CDC6.

The nomenclature we chose might have created confusion. ORC-CDT1-MCM (OC1M) is a subcomplex of ORC-CDC6-CDT1-MCM (OCCM) because it lacks Cdc6.

Line 315: Fig. 1x is referenced.

Thank you for spotting that now Fig. 1c is referenced.

Why was a different DNA construct used for imaging MCM AG?

The DNA sequence used was the same but nucleosome roadblocks were swapped for the more robust methyltransferase (covalent) roadblocks (which can be conveniently aliquoted and stored at -80°). Previous work with yeast proteins showed that both nucleosomes and methyltransferase are equally proficient at retaining MCM on DNA. We expand on these concepts in the methods section.

In Fig. 5 why is one of the MCM subunits in the MO states colored yellow? This is not mentioned in the legend.

MCM5 is colored yellow and MCM2 is colored pink to locate the gate used for DNA loading. This distinction is superfluous in our model figure and we now reverted the color of all MCM subunits to green. Thank you for spotting this issue.

Reviewer 2:

We would like to thank Reviewer 2 for stating that our manuscript “addresses an important research question in a timely fashion”, that we brought “an impressive amount of high-quality biochemical and structural data” to explain origin licensing in human cells”. We were also pleased to read that the reviewer deems biochemical reconstitution of human MCM DH loading “an important step forward in the quest to understand the activation mechanism of the CMG helicase”.

My main critique is that none of the novel observations reported shifts decisively the paradigm that we have learnt from yeast, thanks also to major earlier contributions from the authors' laboratories. In fact, what impresses this reviewer most is the degree of evolutionary conservation of these mechanisms, something we have learned already from the structures of yeast and human replisomes.

We respectfully disagree with the reviewer. We showed in our first submission that DH loading can happen in the absence of Orc6, pointing towards a new mechanism for origin licensing. This is independent of the MO intermediate, which supports DH formation in yeast, according to published evidence (note - we cite an MO-independent pathway in yeast; this manuscript still under review, but is available in bioRxiv, <https://doi.org/10.1101/2024.01.10.575016>). When human ORC6 is present, we also observe MO* formation in CDT1-dropout conditions, indicating that topological loading of a single MCM hexamer around duplex DNA can occur independently of the OCCM intermediate. This was never observed with the yeast reconstituted system.

From the architectural viewpoint human and yeast loading factors show fundamental differences. Unlike with yeast proteins, CDT1 does not interact with loading-competent MCM, although it occupies a very similar position to yeast Cdt1 in the OCCM. Likewise, ORC6 does not interact with free ORC or the OCCM complex. This makes it highly unlikely for the same ORC molecule to flip from interacting with the C- to the N-terminal face of a single-loaded MCM, unlike in yeast DH loading. ORC6 can bridge between MCM and ORC in the MO* complex, but this assembly visits an incompetent configuration that cannot possibly recruit a second MCM hexamer, again unlike yeast.

This is not all. In our revised manuscript we report the discovery that only ORC and CDT1, but not CDC6, are strictly required for origin licensing, though DH loading without CDC6 occurs most efficiently at lower salt concentrations.

Collectively, these observations show that, in the human system, DHs can be loaded by factors previously reported to be essential for origin licensing in yeast, however the identity, function and order of events present fundamental differences.

The observations that some MCM-DH loading can be obtained in the absence of ORC6 is interesting, but it is difficult to assign it its proper biological significance in isolation. Given that human cells perform origin licensing with full-length proteins and in the presence of ORC6, when would this pathway become relevant? Can we be reasonably sure that in the chromatin environment of the cell ORC6 might not be required? I appreciate that this is a biochemical/structural study but for publication in Nature some further proof should be warranted.

Because there were already several publications on the effect of Orc6 depletion on DNA replication in cell lines, we did not feel we needed to repeat them. As we stated in our first submission, depletion of ORC6 from human U2OS cells was shown to have no effect on MCM loading levels in a single cell cycle (<https://doi.org/10.1073/pnas.2121406119>). Also, recently published evidence indicates that ORC6 can be knocked out in primary and immortalised glioma cells. Cell viability and proliferation are decreased but can be observed (<https://doi.org/10.1038/s41419-024-06764-w>). Collectively, these findings support the notion that an ORC6-independent pathway must exist for human origin licensing.

With our biochemical reconstitution work we show that DH loading can occur with or without ORC6. It is likely that the N-terminal ORC1 IDR blocks DH loading without

ORC6, for regulatory purposes. Establishing what molecular event releases the inhibitory function of the ORC1 IDR will take time. The mechanism we discovered is reminiscent of Bruce Stillman's observation that a truncation of the Mcm4 N-terminal tail phenocopies phosphorylation by DDK during replication origin activation reaction (Sheu et al, Nature, 2010). We note that it took us 14 years to discover the molecular mechanism (Puehringer et al. *in preparation*). Similarly, our observation of a function for the ORC1 IDR in selecting the ORC6-dependent or -independent pathway paves the way for the discovery of the regulatory pathway. We note that there is a report indicating that even ORC1 and ORC2 may not be essential in some contexts (Shibata et al. (2016) eLife 5. 10.7554/eLife.19084.) and in Orc1 or 2 depleted cells CDC6 becomes especially important for MCM loading. Though beyond the scope of this current manuscript, this is consistent with the idea that MCM loading in human cells may occur by a variety of mechanisms. It will be fascinating to understand these mechanisms in detail.

The authors have completed an impressive cryoEM tour-de-force and obtained high-resolution information for all of the loading intermediates that have been observed in yeast. The most interesting is the MO*, a putative human yeast-like MO intermediate that does not support a straightforward application of the yeast model for loading of the second MCM copy due to asteric clash.

We are pleased that the reviewer finds our new MO* complex of interest, like we do, and identifies a fundamental difference with the yeast MO intermediate, which we described before.

The interpretation of the role of this intermediate is uncertain; the authors postulate an alternative route whereby the MO* loads single MCM-SH hexamers that can slide towards an OCCM intermediate (or another MCM-SH?) to yield an MCM-DH.

We do observe MO* in the absence of CDT1, implying that topological loading of a single MCM can occur, without ever visiting the OCCM intermediate. That said, CDT1 (which is found associated with MCM only in the OCCM intermediate) is required for DH formation. It does not appear speculative, based on this evidence, to propose a model whereby a single hexamer loaded via MO* can only be converted to DH if it encounters an OCCM (or a SH loaded via OCCM). We now present evidence for the presence of OCCM-SH particles during ATP-dependent loading, which might represent such encounters.

Is it possible that other, more open MO intermediates exist that might be compatible with the 2-step ORC-flip model proposed earlier by the authors? Can the authors confirm that they were unable to observe any MO + MCM species?

Having looked extensively, we have not been able to observe an alternate configuration for the MCM-ORC assembly, which would resemble the yeast MO configuration, or indeed the MO + MCM complex. What appears unlikely based on our data is a yeast-like, ORC-flip model as instead observed in yeast, given that ORC6 is not part of the hOCCM complex. It appears to only interact with the N- and not the C-terminal side of MCM. This is now mentioned in the discussion.

Is it possible that the 2-step ORC flip model evolved uniquely in yeast to exploit the presence of juxtaposed binding sites at the origin?

This is a great observation and the focus of a separate study currently under review and currently available on bioRxiv. doi: <https://doi.org/10.1101/2024.01.10.575016>.

SPECIFIC COMMENTS

The authors state that CDC6 is not required in the MCM loading reaction of ED Fig. 1b. However, the reaction lacking CDC6 shows no MCM loading after 150 mM NaCl wash, unlike the loading reaction with all components. Please clarify.

This was a bead-based experiment which we included in our initial submission as an entry point to the current manuscript and which is now removed for clarity. We include a new experiment (Fig. 3f-h) showing that salt-stable DHs are assembled even in the absence of CDC6 if reactions are performed at slightly lower salt concentrations. Therefore, CDC6 is not strictly needed for DH assembly.

I agree that the ORC1 IDR seems to mediate the ORC6-dependent stimulation. However, removal of the ORC1 IDR stimulates MCM loading in the absence of ORC6 (Fig. 1g, compare 3xFL and truncated ORC1 in the absence of Orc6). So the ORC1 IDR seems to have an intrinsic inhibitory effect on loading. Please comment.

In the results section, we now state specifically that ORC6 may stimulate DH loading by counteracting an inhibitory function of the ORC1 IDR.

Removal of all three IDRs leads to much higher levels of MCM-DH loading relative to truncating ORC1 IDR only, in the absence of ORC6 (Fig. 1g, compare 3xDN). This would suggest further IDR-dependent effects, beyond the effect mediated by ORC1 IDR alone. Please comment.

Thank you for pointing this out. We now mention this in the Results section and comment in the Discussion section.

The authors should make clear in the text that the salt-resistance experiments of Fig. 1j were performed with truncated proteins. The loading reactions in the DNA beads assay of ED Fig. 1a, performed with FL proteins, are sensitive to salt above 150 mM NaCl, whereas the loading reactions in the nuclease footprinting assay of Fig. 1j with truncated proteins are salt-resistant up to 0.5 M NaCl. What is the salt resistance when the assay of Fig. 1j is performed with FL proteins?

We have extended our use of salt resistance as a general assay for stable DH assembly. We now show salt resistant footprint products with truncated proteins +/- ORC6, Full length proteins +/- Orc6 and truncated proteins without Orc6 and Cdc6 in ED Fig. 1g, Fig 3g. We have also repeated the salt stability experiments in Fig. 1j, ED Figs. 1e,f using a more efficient DNA precipitation protocol.

In Fig. 2g, two bands are clearly distinguishable after benzonase digestion of the WT MCM loading reaction, the shorter of which looks similar in size to the shorter product seen with the AG mutant. Further reduction of the MCM footprint on DNA is also noticeable in Fig. 1, comparing panels reporting 0.5 min incubation with benzonase (DNA runs as the 75 marker size) or 20 min incubation (DNA runs just above 50). The authors should interpret this observation.

As explained in our response to reviewer 1, after longer (20 min) Benzonase digestions, the lengths of the products with yeast and human DHs are similar to each other and are about 55 bp. This is very similar to the average DNA fragment size bound by MCM DH sequenced by Li et al. 2023 (54-56 bp) and we believe represents a 'complete digestion' product. After shorter digestion (e.g. 0.5 min) there is a longer (~ 75 bp) intermediate seen primarily with the human wt DH. It is unlikely that this is due to some other protein (e.g. ORC) still bound to a DH because we see the same intermediate sized products with Benzonase digestion after transient high salt treatment (Fig. 1j). It is possible that one or more of the C-terminal MCM WHDs in human MCM bind DNA better than their yeast counterparts, or in some non-specific way hinder Benzonase from accessing the channel exit. Alternatively, the yeast DH can slide whilst the human DH does not and this may somehow affect the nuclease patterns. We include a new Extended Data Figure 1b showing a direct comparison of Benzonase products in a time course with human and yeast DH and we have modified the text on p. 3 to reflect these differences. It is true that all of the Mcm5 mutants have less of the 75 bp product and more of the 55 bp product. This correlates with the fact that the mutants can slide on DNA while the wt DH cannot and suggests that the ability to slide somehow favours the 55 bp product. We have expanded our explanation of these bands in the text on p. 3 and p. 6.

A simple explanation for the increased mobility of the AG MCM-DH double mutant is that R195 and L209 act as pins by protruding into the double helix and reducing the any residual sliding between protein and DNA.

This is a good suggestion and we now include this analogy in the results section. Thank you.

This reviewer is unclear about how it is possible to obtain a fully loaded MCM-SH with ATP γ S, as the authors observe by cryoEM and propose in Fig. 6c. It would be helpful if they commented on it in the Discussion, perhaps at the expense of the text dedicated to the MCM-DH, which can be usefully reduced in size.

Thank you for the useful suggestion. We now elaborate on topological loading of the MCM-SH with ATP γ S in the Discussion section.

MINOR POINTS

Line 187: Please define the MCM6-2 and MCM4-6 "A" domains, for MCM-naive readers.

We now define this as ‘N-terminal alpha-helical (“A”) domain’.

Line 155: Full stop missing.

Thank you for spotting this. Corrected.

Line 256: (ED Fig. 10a) should be ED Fig. 10b?

The reviewer is right. Corrected.

Referee #3:

We would like to thank the reviewer for stating that the “reconstitution of human origin licensing with purified proteins that creates a cryoEM solvable DH (and potential intermediates) is a major advance for the field.” We are pleased that our structural studies of various assemblies observed in DH loading reactions was deemed “well executed” and a strong aspect of the paper.

We would like to thank the reviewer for noting the resemblance of the published DH purified from cells to our DNA-loaded DH reconstituted in a test tube. The reviewer is right in saying that “This correspondence provides strong evidence that the in vitro process is recapitulating human origin licensing.” It is a point we now also make in the results section. Also, we are pleased that the reviewer wrote that “An important contribution of this study is that it demonstrates that the partially unwound DNA observed when DHs are purified from cells is a consequence of loading and not a downstream activation event.” We agree that this a key aspect of our findings. We also appreciate the reviewer’s remark that the human loaded DH is ADP-bound as in the yeast reconstituted structure, “and this observation suggests that Mcm2-7 ATP hydrolysis is involved in human helicase loading as is known for the yeast reaction.” This is a good point, which we now also note in the results section. We appreciate that the reviewer identifies elements of novelty in our structural study, i.e. “Orc6 is not observed in the human OCCM “ and “a structure of single Mcm2-7 hexamer” where “the Mcm2-5 gate [...] is in a closed state.” The reviewer goes on to write that “Such an intermediate has not been characterized in yeast previously, suggesting that this could be an intermediate that is unique to the human reaction.” We agree with the reviewer that the cryoEM shows that “the reconstituted reaction [...] leads to the formation of a relevant Mcm2-7 DH”. Also we concur that our structures leave options open regarding the mechanism of DH loading. We are pleased that the reviewer thinks that the cryo-EM work provides “structures that could facilitate further studies to reveal such mechanism(s).”

We are pleased to read that the reviewer considers our DNA protection assay for helicase loading “a potential step forward” and, following their suggestion, we characterised our assay further.

A major concern with the paper is the discrepancies between the benzonase protection assay and the EM studies. While the benzonase-protection studies focus on reactions without Orc6, the EM studies used to solve structures are performed with Orc6. A particularly confusing aspect of the study arises when the studies are compared. For example, all the EM studies are done with the deleted Orc1, Cdc6, and Cdt1 and mostly with Orc6 (all structural studies included Orc6, but some 2D experiments did not).

To clarify, our cryo-EM studies were performed in the presence of ORC6. Our (negative stain) EM studies (2D averages) were done in the presence or absence of ORC6, to recapitulate the conditions explored with the Benzonase assays.

According to the benzonase protection assay this should lead to very poor DH formation, yet the authors get enough DHs and other potential intermediates using this combination of proteins to determine their structures. This raises questions about the validity of either the benzonase assay or the EM assay to detect DHs.

The usefulness of the Benzonase assay is discussed above. Also, despite using conditions that render loading inefficient, we obtained enough DH particles to obtain a high resolution reconstruction. The large size and symmetric nature of the DH makes this possible. In fact, less than 16,000 DH particles were used to solve a 3.1 Å resolution structure. Instead, for example, more than double the number of OCCM particles are needed to solve a 3.8 Å resolution structure. The DH is just an easier structure to solve.

It is possible that the EM assay detects unstable versions of a DH structure. Alternatively, the benzonase assay may not be an accurate measure of DHs. Another important area of discrepancy is that cryo-EM structural studies are not performed on the DHs formed in the absence of Orc6. Because the DH structure is the strongest evidence that human helicase loading can be reconstituted with purified proteins and is forming a correct (and hopefully active) DH, this absence greatly reduces the relevance of the studies without Orc6.

To address the points made by the reviewer we decided to determine the structure of human DH in the absence of ORC6. We provide evidence that the architecture of the DH complex is the same. Most importantly, the degree of DNA untwisting and opening at the interface between two hexamers is the same. (ED Fig. 6a-c). We conclude that the insights gained by our structural investigation of DH loading in the presence of ORC6 are valid and will be useful to the field as the mechanisms of loading and regulation are explored. Please note - the resolution of the DH obtained in ORC6 dropout conditions is slightly lower because we used a less powerful electron microscope for this control experiment, (operated at 200kV, not 300kV) and a direct electron detector operated in linear, not in counting mode.

Finally, that they chose to study the structures from a condition that is non-productive in their DNA protection assay reduces the value of the structural studies (see below specific point 6 about potential differences between the EM and benzonase assays that could explain such discrepancies).

The new footprinting assays we provide make it clear that even with the truncated proteins, there is detectable DH assembly in the presence of Orc6 and with the FL proteins there is detectable DH assembly in the absence of Orc6. That is, both the Orc6-dependent and -independent pathways are available with either FL or truncated proteins; it is just the balance between the pathways that differs.

The idea of a “double OCCM” as an intermediate in loading becomes a theme for a proposed Orc6-independent alternative mechanism for human origin licensing.

We provide further evidence for double OCCM formation in the ATP-hydrolysis powered DH-loading reaction, as discussed below.

A significant weakness of these studies is the lack of evidence that the benzonase-protection interpreted to be a double-OCCM is in fact due to this complex. For example, instead of being double the size of the OCCM protection (50 bp), it is 2.5 times larger. This suggests a role for other complexes in this larger protection (e.g. an hMO?).

This is an interesting point, which we now address in the Results section. The extent of the double OCCM footprint is likely be influenced by how tightly the N-terminal domains interact. A double OCCM is unlikely to have zinc finger domains as tightly interacting as a DH, which will be likely to result in a more extended footprint. Most importantly, these experiments were done in ATP γ S and we do observe double OCCMs under cryo-EM in the same conditions. No other cryo-EM intermediate could justify such a large DNA footprint. This observation rules out the possibility that the extended footprint in question is due to hMO.

In favor of the double OCCM hypothesis, the authors provide evidence that the “OCCM” has increased benzonase sensitivity which they interpret as sliding on DNA (note: this experiment is missing a critical control, see below). On the other hand, both the lengthy time required for appearance of the “double OCCM” protection (Fig. 3f) relative to DH footprint formation (Fig. 1c) and the apparent lack of prevalence of the “double OCCM” in the EM studies of reactions with ATP suggests that this is not a common mechanism/intermediate.

This is a good point from the reviewer. We now collected larger negative stain datasets of the ATPase powered DH loading reaction (performing several repeats shown in ED Fig 3a) and observed two new species that previously escaped our notice. One is the same double OCCM which was previously observed in ATP γ S. The second is a single OCCM next to a single loaded MCM. Both species were observed reproducibly both in the presence as well as in the absence of ORC6. Notably, OCCM is less frequent in ATP than in ATP γ S, supporting the notion that single loaded MCM can be the product of ATPase powered maturation of OCCM. Likewise, double OCCMs are less frequent than DHs in ATP than in ATP γ S. This observation supports the notion that double OCCMs (or OCCM-SH assemblies) are possibly loading intermediates. The

Overall, the data presented shows that the authors have developed biochemical assays to assemble human Mcm2-7 into DHs on DNA and they have used this to analyze the

structure of potential intermediates in this process. These structures will be valuable in the subsequent determination of the mechanism of this event. The data presented does not allow the authors to strongly support any of the models presented in Fig. 5 and instead they are mostly derived from analogy with the yeast system.

We would like to thank the reviewer for stating that the structures will be “valuable in the subsequent determination of the mechanism of this event.” We want to point out difference in the structural assemblies observed in the DH loading reaction established with human vs yeast proteins, seeking to build the foundation for future studies.

The strongest conclusion of the paper is that there is an Orc6-dependent and a possible Orc6-independent mechanism (but see specific point 2 below). The finding that they only see the Orc6-independent mechanism with mutant forms of Orc1, Cdc6, and Cdt1 that would not be present in cells makes significantly reduces the biological interest of this potential pathway.

As explained above, the balance between the Orc6-dependent and independent pathways is different with the FL and truncated proteins. We believe that our observation of double OCCM as well as single OCCM engaging a SH in the ATPase powered DH loading reaction provide important corroborating evidence that an ORC6-independent reaction for DH formation exists. This is for two reasons: 1. We observed no evidence of ORC6 binding in OCCM. 2. Double OCCM and single OCCM engaging a SH both interact via their N-terminal domains, which is the dimerisation interface observed in the DH. As stated above, the fact that double OCCMs are far less frequent in ATP than in ATP_γS (now shown in ED Fig. 3) invites the reasonable hypothesis that they might be a loading intermediate (as we originally showed in our Figure 5b cartoon).

Specific points:

1. One significant concern with the manuscript is the extensive use of a benzonase protection assay to identify protein complexes associated with DNA. Although an interesting assay, the authors cannot use this assay to demonstrate what proteins are responsible for the observed protections. While they perform a number of experiments that suggest the DH is responsible for the protected DNA, there are important discrepancies with the EM studies (see above). The inability to identify the associated proteins responsible is a significant concern for the double OCCM. Without more direct evidence, the authors need to be more careful in their statements concerning the components of the assemblies responsible for DNA protection in these assays. It would greatly improve the paper to include some assay that identifies the proteins in these complexes. Although the authors frequently refer to the EM studies to support their conclusions, there are many different assemblies observed in these EM grids, any one of which could be responsible.

The reviewer points out that the Benzonase protection assay does not provide information on identity of stable DNA-bound complexes, and acknowledges that we interpret our data by comparing the observed protein footprints with the cryo-EM

results. We point out that much of what we know about the yeast system actually came from early footprinting experiments; by using a variety of mutants and conditions, DNA protein complexes could be inferred. We believe that coupling footprinting with cryo-EM is significantly more informative than any bead-based assay. It tells us what factors are present in what nucleo-protein assembly formed during the DH loading reaction, what conformation they are in and how they protect DNA. We believe that the bead-based assay, informing us on the identity of factors that remain bound to DNA after washing, would not add useful information. We use the Benzonase assay to detect molecular species that accumulate over time (e.g. double OCCM). As stated above, our correlation with double OCCM complexes observed by EM is unambiguous, as no other protein assembly observed would protect such an extensive stretch of DNA.

2. In ED Fig. 1b, the “no Orc6” lane shows an easily detectable band at the position of Orc6. Could this be insect cell Orc6? This is a critical issue that needs to be addressed given that the major take home message of the paper relies on this reaction not having Orc6. While the gels of the purified proteins show no bands at this point in the Orc1-5, Cdc6, or Cdt1 proteins, small amounts of contaminating Orc6 could be present and become concentrated on the DNA during the reaction. It would be much more compelling if the authors included analysis of a mutant in Orc1-5 that could no longer interact with Orc6.

We have now analysed all our protein preps by mass spectrometry and were unable to detect ORC6 (either human or *Spodoptera frugiperda*) in any prep which is now stated in Methods.

3. The EM grids that are presented in Ext. Data Fig. 2 are frequently referred to in the paper with a focus on what assemblies are or are not present. The authors must present this data in a more quantitative manner to allow evaluation of their significance. A good way to do this would be to report the percentage of Mcm complexes observed in each type of assembly under each condition. This would be particularly helpful with regard to the “double OCCM”. An averaged EM image of this is shown (Fig. 3g) and is apparently derived from an ATP γ S experiment, but there is no evidence as to its frequency in any of the EM grids. Without a sense of frequencies, these data are of limited use. Given that the authors indicate that they are using computational analysis to identify the different assemblies this should be easy to do. The authors should also indicate what percentage of particles were left unassigned/uncharacterizable.

This is a fair point. Because we felt that the key concern of the reviewer is whether or not DH loading can happen in the presence or absence of ORC6, we decided to repeat the negative stain EM assays in these two conditions and in both ATP or ATP γ S. Quantitation shown in Extended Data Figure 3a shows that OCCM and dOCCM particles are more abundant in ATP γ S than in ATP and confirm the observation that DHs can only be observed in ATP. Because the side of MCM at the interface between two OCCM particles is the same as the homo-dimerisation interface of the DH, our results suggest that the double OCCM might be a DH loading intermediate.

4. If the authors want to propose that the double OCCM is an important intermediate in

the absence of Orc6, it should be observed in experiments with ATP and it should be inhibited by addition of Orc6. Tests of both of these predictions could and should be included in the manuscript.

As stated in our reply to point 3, double OCCMs are now observed in ATP, having collected large datasets for reactions performed in multiple repeats. We could observe this molecular species both in the presence or absence of ORC6. We could also observe a single OCCM interacting with the N-terminal domain of a single loaded MCM. Although we did observe MO* averages when ORC6 was present, we could never detect MO* particles next to an OCCM or a single loaded MCM, compatible with the notion that ORC6 poses a steric impediment to the dimerisation of two MCM assemblies, in the conditions tested (3xΔN, N-terminal truncation of ORC1, CDC6 and CDT1). We thank the reviewer for suggesting this set of experiments, which we feel strengthen our study.

5. In the discussion, the authors suggest that the hMO* could act to stabilize single hexamers (SHs) on the DNA; however, earlier in the paper they suggest that SHs observed are stably closed around the DNA. The authors should address this discrepancy.

In the Discussion we now refer to our recent studies in yeast, indicating that a single loaded MCM is less stable than the DH, indicating that unloading might occur because of the transient opening of the Mcm2-5 gate. The same might happen with the human single-loaded MCM. Providing one additional DNA grip, the N-terminally interacting ORC observed in MO* might serve a stabilising function.

6. One concerning aspect of the manuscript is the use different protein concentrations in different assays without explanation. For example, they use a two-fold molar ratio of Orc1-5 and Orc6 in the experiments with the deleted proteins but a 1:1 ratio of Orc1-5 and Orc6 in the full-length experiments and cryoEM experiments (using the deleted proteins). This raises the possibility that equimolar Orc6 is beneficial and excess Orc6 is inhibitory. For an accurate comparison, these experiments should be repeated using the same protein concentrations. Similarly, they use lower concentrations of most of the proteins in the bead-based assays than they use in the benzonase assay. Could this explain the lack of signal in the bead-based assays? Finally, although the authors show that lack of the Orc1-IDR reduces the extent of Orc6 stimulation, comparison of Fig. 1g lanes 1&2 vs lanes 5&6 shows that loss of the Cdc6 and Cdt1 IDRs further reduces Orc6 stimulation. It is concerning that there are no repetitions and quantification of any of the benzonase experiments to demonstrate that such differences are significant. It would also improve Fig. 1 if the authors stuck with one time of benzonase treatment (either 0.5 or 20 min).

Some of the smaller differences in conditions are due to the fact that the EM and Benzonase assays were optimised in parallel and therefore diverged somewhat. However the reviewer makes an important point regarding the ratio of ORC6 to ORC1-5 and the possibility that equimolar amounts might be beneficial and excess might be inhibitory. We have now tested this with a new experiment (ED Fig. 1d) with one

Benzonase time (20 min) which shows clearly that the inhibitory effect of ORC6 with 3X Δ N and the stimulatory effect of ORC6 with 3XFL occur at sub-stoichiometric, stoichiometric and super-stoichiometric Orc6 amounts. In general, we have used 0.5 min Benzonase when we were trying to get a 'snapshot' of the DH, for example in sliding experiments. Because of the aggregation caused by the N-terminal IDRs, 0.5 min results in a smeary product with full length proteins, so we also used the 20 min Benzonase as an endpoint of the digestion reactions.

7. One oddity of the paper is that the authors are not able to identify protein associations in the bead pull-down assay that has been a reliable measure of helicase loading protein-DNA association in the yeast assay. Does this mean that the complexes they are observing by benzonase footprinting are short-lived? Is the reaction very inefficient? Why are so many intermediates observed by EM but not by other assays? Could these differences have to do with the different protein concentrations used in the different assays mentioned above?

With our extensive experience, we, of course, started out using bead-based assays. We have removed this data from our revised manuscript. We found that such assays were not reliable primarily because the human proteins have a much higher tendency to aggregate and stick to beads. Once we developed the Benzonase assay, we saw no reason to return to the bead-based approaches. As discussed above, we obtain so much information from the electron microscopy, there is little reason to return to the bead-based assays. We do not think the terminal complexes (DH in ATP; OCCM in ATP γ S) are any less stable than in yeast. Some of the intermediates (e.g. double OCCM in ATP) are probably too short-lived to detect by Benzonase footprinting.

8. The comparison of an ARS1-containing plasmid and a human origin-containing plasmid is not useful since the experiments were not done in the same plasmid backbone. Also, there is a notable difference in the benzonase footprint on different DNA templates suggesting that the ARS1 sequence is enhancing (with the caveat that the plasmids have different backbones and the authors have not established the quantitative nature of this assay). Does this mean that human origins inhibit the reaction they are studying? If the authors want to explore the role of DNA sequence in this reaction, they need to do a better experiment. If not, the current experiment should be removed from the paper.

We have now repeated the origin comparison experiment using equal-sized (2.4 kb) DNA fragments containing the human c-Myc origin, the human Lamin B2 origin, yeast ARS1 and no origin (pET21 backbone). The human and yeast fragments lack any bacterial sequences. We see no significant differences in DH assembly with any sequence.

9. The authors should consistently indicate in the legends for the figures whether they are using the full-length or deleted Orc1, Cdc6 and Cdt1 proteins in the assays shown. For example, this is not indicated in ED Figs. 1 or 2 legend and should be.

Figure legends have all been checked and now explicitly state when full-length or truncated proteins were used.

10. The evidence that the benzonase assay detects sliding strongly depends on experiments showing that it is the time in benzonase rather than the time of the reaction that leads to loss of signal. For Fig. 2g the authors include the appropriate control to demonstrate this is the case (last two lanes). This same control is equally important for the experiments suggesting the “OCCM” slides in Fig. 3h. Without the control the change in sensitivity is equally likely to be due to instability and the interpretation would need to be changed.

We agree and have performed this experiment with the appropriate control, located in Extended Data Fig. 9e.

Reviewer Reports on the First Revision:

Referee #1:

In their revised manuscript the authors have reasonably addressed all of our specific points, including adding a new experiment testing MCM5 mutants. I think the changes have improved what was already a very strong manuscript and I support publication.

Line 186: 50bp

For a general audience it might be worth mentioning that MCM is a heterohexamer in the intro or start of results - currently different MCM subunits are mentioned in the text for the first time in the description of the hDH structure.

Referee #2:

I find that the manuscript revisions by Weissman et al. have markedly improved an already strong manuscript. The additional biochemical data especially strengthen the case for the multiple postulated mechanisms of MCM DH loading. The description of the cryoEM intermediates is also sharper and the discussion of possible mechanism of hDH loading clearer and easier to follow, although it could be reduced in size if required. Please find below a list of minor points that should be addressed before publication.

Lines 46-47: Please include brief experimental details in the text for 'a human MCM loading reaction'. Given the importance of salt concentration in past experiments on yeast DH loading, the readers will wonder how was this reaction performed. The authors should mention briefly at least the salt concentration used here and whether FL or truncated proteins were used, and point to the section of the Methods where details of the reaction are provided. At the moment, the Methods section 'Sample preparation and data collection for negative stain EM' refers to the dropout experiments in ED Fig. 2 only. Were the same conditions used for the negative-stain EM experiment of ED Fig. 1a?

Line 82: Rather than 'the preferred pathway', it would be better to say 'a more efficient pathway'. The presence of ORC6 does decrease DH yield with truncated proteins, which means they don't have a choice not to use it.

Line 97: The reader might wonder why dilution is explicitly mentioned here. If it is a routine step of the digestion protocol, it doesn't need mentioning; otherwise an explanation should be added. I suggest to replace the sentence with: 'before Benzonase treatment'.

Line 112: Better to use 'The nucleotide-binding site' rather than the more generic 'The active site'.

Line 122, '45 bp could be clearly resolved inside the MCM hexamer': Should this be the double MCM hexamer?

Line 131: The structure shows that 5 bp of DNA become underwound at the MCM DH interface. However, the abstract uses the word 'unwound', which to me implies full strand separation (not the case here). Please correct the abstract. Also check the out use of 'unwound' in the Discussion (line 359).

Line 141: Better to use 'MCM2-7 proteins' or 'MCM2-7 heterohexamers' rather than 'MCM complexes', which can mean MCM bound to some other proteins.

Fig. 3a lacks FL/trunc annotation in either figure panel or legend. Please make sure throughout that it is always clear whether FL or truncated proteins are used.

Line 206: 'which cannot be averaged when CDC6 is absent' is technical jargon and should be clarified.

TYPOS

Line 166: Add space in '50bp'.

Line 209: 'independently' instead of 'idependently'.

Line 243: 'glutaraldehyde' instead of 'gluraraldehyde'.

Line 262: Comma missing between 'body' and 'ED'.

Line 348: 'capitalises' instead of 'captitalises'.

Line 783: 'Preparation DHs...' (missing 'of')

LANGUAGE

The title is too technical at the moment and impenetrable to the general readership. Please consider expanding it to something like: "DNA loading of the Replicative MCM Helicase Double Hexamer visualised with human proteins."

Please consider revising the following colloquial sentences:

Line 45: Replace 'unlike yeast' with 'unlike in yeast'.

Line 48: Replace 'yielded 2D averages similar to yeast DH' with 'yielded 2D averages of a particle similar to yeast DH'.

Lines 55-57: Replace the sentence 'all that remains after digestion ... polyacrylamide gels' with 'the size of DNA protected by proteins after digestion is analysed by polyacrylamide gel electrophoresis'.

Line 68: Replace 'the C-terminal exit' with 'the MCM C-terminal exit'.

Line 69: Replace 'very similar' with 'in a very similar fashion'.

Line 140: Please use 'initiation of dsDNA melting' rather than 'bubble formation'.

Line 184: Use 'contact' instead of 'touch'.

Lines 359-61: The sentence 'stabilised by R195 and L209 of MCM5 that pin the double' should refer to 'the double helix'. Also avoid repetition of 'stabilise'.

In the annotation of figure lanes, it might be preferable to use 'all' to refer to samples with all origin licensing proteins added, rather than 'full'.

Referee #3:

The revised manuscript from Costa and colleagues addresses the mechanisms of human origin licensing using a powerful combination of biochemistry and structural biology. The authors have nicely addressed the major concerns with the initial manuscript. While there are still questions concerning the mechanisms of origin licensing used in vivo (it still seems most likely that a MO-dependent mechanism will be most common), the data presented makes a strong case that other mechanisms will occur in some situations. The final manuscript will be of interest to those studying DNA replication and cell cycle control.

Author Rebuttals to First Revision:

Referee #1:

We are delighted that the reviewer supports publication of our work.

Line 186: 50bp

Changed to 50bp

For a general audience it might be worth mentioning that MCM is a heterohexamer in the intro or start of results - currently different MCM subunits are mentioned in the text for the first time in the description of the hDH structure.

We changed the first sentence of the results to: “We expressed human ORC, CDC6, CDT1 and MCM2-7 (hereafter MCM) using the biGBac baculovirus expression system¹⁰ (Fig. 1a).”

Referee #2:

We would like to thank the reviewer for stating that we have “markedly improved an already strong manuscript” and that “the additional biochemical data especially strengthen the case of multiple mechanisms for MCM DH loading”. Below we address the minor points raised.

Lines 46-47: Please include brief experimental details in the text for ‘a human MCM loading reaction’. Given the importance of salt concentration in past experiments on yeast DH loading, the readers will wonder how was this reaction performed. The authors should mention briefly at least the salt concentration used here and whether FL or truncated proteins were used, and point to the section of the Methods where details of the reaction are provided. At the moment, the Methods section ‘Sample preparation and data collection for negative stain EM’ refers to the dropout experiments in ED Fig. 2 only. Were the same conditions used for the negative-stain EM experiment of ED Fig. 1a?

We provide now more details for the experiment shown in ED Fig. 1a in the main text, legend and methods. The sentence now reads " Negative-stain EM imaging performed in solution for a human MCM loading reaction using reaction conditions similar to established yeast reactions (100 mM potassium glutamate) on a short synthetic yeast origin flanked by nucleosomes yielded 2D averages of a particle similar to yeast DH (ED Fig. 1a), suggesting that hDH was being assembled." The legend now states that ΔN proteins were used and details specific to this experiment are now given in the methods sections "Preparation of DNA templates for EM experiments" and "Sample preparation and data collection for negative stain EM".

Line 82: Rather than ‘the preferred pathway’, it would be better to say ‘a more efficient pathway’. The presence of ORC6 does decrease DH yield with truncated proteins, which means they don’t have a choice not to use it.

The revised sentence now reads “From this we conclude that there is a pathway for MCM loading that does not require ORC6 and that this is a more efficient pathway when truncated proteins are used.”

Line 97: The reader might wonder why dilution is explicitly mentioned here. If it is a routine step of the digestion protocol, it doesn't need mentioning; otherwise an explanation should be added. I suggest to replace the sentence with: ‘before Benzonase treatment’.

For efficient Benzonase activity, dilution to a lower salt concentration is required in these experiments. We followed the suggestion and replaced the sentence with "before Benzonase treatment". The legend to Fig. 1j now reads "... followed by dilution to lower salt and Benzonase treatment."

Line 112: Better to use 'The nucleotide-binding site' rather than the more generic 'The active site'.

We feel "The nucleotide-binding site" would create a repetition (ATP-bound). We changed "active site" to "catalytic site".

Line 122, '45 bp could be clearly resolved inside the MCM hexamer': Should this be the double MCM hexamer?

Thank you for catching this mistake. The sentence now reads: "45 bp could be clearly resolved inside the two MCM rings, though the length of the central channel is compatible with a 75 bp protection seen by DNA footprinting."

Line 131: The structure shows that 5 bp of DNA become underwound at the MCM DH interface. However, the abstract uses the word 'unwound', which to me implies full strand separation (not the case here). Please correct the abstract. Also check the out use of 'unwound' in the Discussion (line 359).

As suggested, we swapped unwound to underwound whenever discussing the untwisted DNA at the MCM homo-dimerisation interface.

Line 141: Better to use 'MCM2-7 proteins' or 'MCM2-7 heterohexamers' rather than 'MCM complexes', which can mean MCM bound to some other proteins.

We changed 'MCM complexes' to 'MCM hexamers'.

Fig. 3a lacks FL/trunc annotation in either figure panel or legend. Please make sure throughout that it is always clear whether FL or truncated proteins are used.

We indicate now in both Fig. 3a and legend that ΔN proteins were used.

Line 206: 'which cannot be averaged when CDC6 is absent' is technical jargon and should be clarified.

We changed this to 'which becomes too flexible to be seen when CDC6 is absent.'

TYPOS

Line 166: Add space in '50bp'.

Changed to 50 bp.

Line 209: 'independently' instead of 'idependently'.

Fixed.

Line 243: 'glutaraldehyde' instead of 'glurraldehyde'.

Fixed.

Line 262: Comma missing between 'body' and 'ED'.

Fixed.

Line 348: 'capitalises' instead of 'captitalises'.

Fixed.

Line 783: 'Preparation DHs...' (missing 'of')

Fixed.

LANGUAGE

The title is too technical at the moment and impenetrable to the general readership. Please consider expanding it to something like: “DNA loading of the Replicative MCM Helicase Double Hexamer visualised with human proteins.”

The suggested title is great but exceeds the limits on title length by 14 characters. We decided to stick to our original title. Thank you for the suggestion.

Please consider revising the following colloquial sentences:

Line 45: Replace ‘unlike yeast’ with ‘unlike in yeast’.

Done.

Line 48: Replace ‘yielded 2D averages similar to yeast DH’ with ‘yielded 2D averages of a particle similar to yeast DH’.

Thank you for the suggestion. The sentence now reads: “Negative-stain EM imaging performed in solution for a human MCM loading reaction on a short synthetic yeast origin flanked by nucleosomes yielded 2D averages of a particle similar to yeast DH”.

Lines 55-57: Replace the sentence ‘all that remains after digestion ... polyacrylamide gels’ with ‘the size of DNA protected by proteins after digestion is analysed by polyacrylamide gel electrophoresis’.

Thank you for the suggestion. We decided to keep the original sentence.

Line 68: Replace ‘the C-terminal exit’ with ‘the MCM C-terminal exit’.

We changed the sentence to “The 75 bp intermediate suggests additional contacts with DNA at the C-terminal exit from the MCM central channel.”

Line 69: Replace 'very similar' with 'in a very similar fashion'.

Thank you for the suggestion. We believe the sentence is clear and concise as written.

Line 140: Please use 'initiation of dsDNA melting' rather than 'bubble formation'.

Changed to "To assess the role of R195 and L209 of MCM5 in hDH assembly and nucleation of DNA melting,"

Line 184: Use 'contact' instead of 'touch'.

Change implemented as suggested.

Lines 359-61: The sentence 'stabilised by R195 and L209 of MCM5 that pin the double' should refer to 'the double helix'. Also avoid repetition of 'stabilise'.

The sentence now reads "Firstly, hDH has a 5bp stretch of underwound DNA between the hexamers, with one broken base pair stabilised by R195 and L209 of MCM5 that pin the double helix and are not required for untwisting."

In the annotation of figure lanes, it might be preferable to use 'all' to refer to samples with all origin licensing proteins added, rather than 'full'.

Changed as suggested.